# Sheared turbulent flows and wake dynamics of an idled floating tidal turbine

Lilian Lieber [1,2] ✉, Shaun Fraser [3], Daniel Coles [4] & W. Alex M. Nimmo-Smith [2]

Ocean energy extraction is on the rise. While tides are the most predictable amongst marine renewable resources, turbulent and complex flows still challenge reliable tidal stream energy extraction and there is also uncertainty in how devices change the natural environment. To ensure the long-term integrity of emergent floating tidal turbine technologies, advances in field measurements are required to capture multiscale, real-world flow interactions. Here we use aerial drones and acoustic profiling transects to quantify the site- and scale-dependent complexities of actual turbulent flows around an idled, utility-scale floating tidal turbine (20 m rotor diameter, D). The combined spatial resolution of our baseline measurements is sufficiently high to quantify sheared, turbulent inflow conditions (reversed shear profiles, turbulence intensity >20%, and turbulence length scales > 0.4D). We also detect downstream velocity deficits (approaching 20% at 4D) and trace the far-wake propagation using acoustic backscattering techniques in excess of 30D. Addressing the energy-environment nexus, our oceanographic lens on flow characterisation will help to validate multiscale flow physics around offshore energy platforms that have thus far only been simulated.

Ocean energy has great potential to help decarbonise global energy demands, with tides offering the most predictable marine renewable resource[1]. Tidal stream energy generation, which mostly uses underwater horizontal axis turbines conceptually similar to wind turbines, is estimated to meet up to 11% of the UK's current annual electricity demand (34 TWh/year)[2]. However, tidal stream sites are highly energetic environments, where high loading from fast (current speeds >2 ms⁻¹) and turbulent flows (turbulence intensities >10%) challenge reliable energy extraction, also requiring an increase in the conversion efficiency[3,4]. While average tidal flow velocities are largely predictable, more constrained flow passages that provide the densest tidal energy resource[2], such as tidal channels, basins or headlands generate highly dynamic flow environments[4]. Where tidal currents accelerate in areas of complex topography and bathymetry, such sites can experience strong temporal and spatial variability in mean velocities and associated turbulence. For instance, fast currents past headlands and islands can generate back-eddies (local flow reversals) and leeward wakes bounded by strong horizontal shear (cross-stream gradient in streamwise velocity, or 'shear layer') hereafter referred to as 'shear lines'[5–7]. Vortical structures, upwelling (surface divergence), and associated downwelling are characteristic of shear lines[7,8] and their often kilometre-scale streamwise extent and cross-stream location will change with underlying flow velocities, as well as wind and wave action. On finer scales, one cannot assume that vertical profiles of velocity are either spatially uniform or follow a logarithmic profile in the vertical distribution (where current velocities increase towards the surface)[9–11], with highly variable vertical shear (vertical gradient in horizontal velocity). Further, the ubiquitous presence of macroscale turbulent coherent structures, known as 'boils' when their turbulent signature impinges on the free-surface, will result in high-velocity fluctuations through the water column[12–15]. Together, the interaction of shear (vertical and horizontal) and intermittent, yet highly dynamic

[1]Marine Biological Association of the United Kingdom, The Laboratory, Citadel Hill, Plymouth PL1 2PB, UK. [2]School of Biological and Marine Sciences, University of Plymouth, Plymouth PL4 8AA, UK. [3]UHI Shetland, University of the Highlands and Islands (UHI), Scalloway Campus, Shetland ZE1 0UN, UK. [4]School of Engineering, Computing and Mathematics, University of Plymouth, Plymouth PL4 8AA, UK. ✉e-mail: lilian.lieber@mba.ac.uk

turbulent motion from coherent structures can result in unsteady forcings to tidal turbine blades, rotors, support structures and foundations[16,17], compromising the structural integrity and, therefore, the reliability of tidal stream turbines[18–23]. While turbulence intensity (TI), which relates the root-mean-square of velocity fluctuations to the mean velocity, is often considered as the key variable affecting turbine performance, TI can be a poor predictor of water column turbulence in the presence of coherent structures[15]. Instead, turbulence length scale, defined as the size of the largest energy-containing turbulent eddies, may have the greatest impact on turbines[18,24].

Thus, detailed characterisation of highly complex tidal flows is critical to inform suitable device placing to ensure turbine loading is maintained within design limits. However, the collection of field measurements is not trivial given the diverse designs of tidal stream energy devices, especially with the emergence of floating turbine technologies. Unlike more conventional tidal turbines, which are generally fixed to the seabed and fully submerged, surface-floating devices are anchored to the seabed using mooring systems. Designed to make use of stronger current magnitudes near the sea surface, they also allow easier access for installation, operation and maintenance. Generally wider in the cross-stream dimension, some floating platforms have several horizontally-spaced rotors (2–4) with a total cross-stream extent in excess of 30 m (e.g. the PLAT-I by Sustainable Marine Power or the O2 by Orbital Marine Power). Being secured by dynamic mooring systems, floating turbines experience platform motion, excursion from their centre point during the tidal cycle in excess of tens of metres, and changes in turbine heading. Associated inflow measurements for power performance assessments, therefore, cannot be achieved using conventional acoustic Doppler current profiler (ADCP) point measurements (e.g. from a seabed-mounted ADCP) if flows vary in the cross-stream.

Apart from the inflow, it is important to assess the wake dynamics of floating platforms to quantify hydrodynamic effects on the environment (i.e. their 'physical footprint' on the environment including infrastructure-induced turbulence[25]), energy dissipation[26], as well as to inform array planning[27]. As the flow interacts with the rotors and the floating platform itself, far-wake effects can be observed downstream, with velocity deficits and turbulence fluctuations influencing the performance of downstream turbines and energy dissipation due to wake mixing[4,25,27–30]. It is still unknown how wakes expand, dissipate and recover in complex natural flows, and how combined effects of free-stream turbulence, shear and waves may affect wake dynamics[31,32].

To capture the fine-scale spatial heterogeneity in flows across sites, including inflow and wake dynamics, both cross-stream vessel-mounted or streamwise drifting ADCP surveys provide valuable means for flow field characterisation[23,28,33–37], and it may be that such mobile measurements provide the best approach to capture the flows around dynamic floating platforms. Complementary spatial measurements of surface currents and turbulence structures can be achieved using image orthorectification[38] or more novel approaches, such as radar[39], satellites[40], or aerial drone measurements[41,42] using image-based velocity estimates. Specifically, aerial drone hovers can be used to track surface flows to extract mean velocities and turbulent coherent structures[43], highly complementary to ADCP transects which omit the very near-surface. Aerial drone measurements also allow safe access to measure surface flows in the direct vicinity of platforms (e.g. upstream or even over platforms) in strong flows (>3 ms⁻¹) where conventional mobile instrumentation would be unsafe to use.

While wake signatures are usually detected using ADCP-derived velocity quantifications, including measures of velocity deficit and turbulence intensity[28], wakes can also be visualised using acoustic backscattering techniques[25]. Active acoustics provides a powerful tool to rapidly and remotely investigate both biological and physical properties of the water column. Whilst acoustic backscattering techniques are more commonly used for biological studies (e.g. fish

detection[44–46]), they are also widely used in physical oceanography to map physical scattering sources[47–51] and processes, specifically wave-breaking and the formation of entrained bubble clouds and subsequent advection by near-surface flows[14]. As surface wakes contain entrained air in the turbulent surface boundary layer (bubble entrainment)[52], using acoustic scattering to trace the wake propagation of floating platforms with downstream distance may provide a high-resolution, synoptic approach to traditional velocity-related wake measures.

Field measurements still present costly and difficult endeavours, however validation at all scales is key to better understand complex flow environments and multiscale flow physics that have thus far only been simulated[4,31]. Inflow and wake measurements alongside advances in analysis techniques are thus vital to ensure the effective design, operation and long-term integrity of tidal stream turbines[4]. Our present study was motivated by the large, existing gap between theoretical predictions of flow dynamics and the fine-scale complexity of real-world flows[20].

Here we set out to develop new combined survey methodologies and analytical approaches for floating platforms that can capture all three points raised above, namely, (1) the spatio-temporal variability in flows across a site, (2) the immediate inflow to the rotors and (3), downstream far-wake dynamics. Using a combination of aerial drones and acoustic sensors, we characterise the flow fields around the world's most powerful tidal stream turbine, the floating dual-rotor O2 (Orbital Marine Power, 2 MW capacity; D = 20 m rotor diameter). The O2 is installed at Europe's largest tidal energy test site; the European Marine Energy Centre (EMEC) in the highly energetic tidal stream of the Fall of Warness, Orkney, Scotland, UK (Fig. 1); a site which has never previously been characterised using spatial ADCP transects. Our objective is to provide a more holistic, physical oceanographic lens to multiscale flow interactions at the device-scale (excluding blade hydrodynamics), as opposed to idealised fluid dynamic conditions currently resolved in either numerical models or in the laboratory. We use aerial drone hovers to capture the surface velocity fields up-and downstream of the O2, as well as cross-stream line transects using a vessel-mounted ADCP and an EK80 echosounder to map the flows, measure wake velocity deficits and trace the far-wake propagation associated with the O2 in its idled configuration.

## Results
### Broad-scale spatial flow variability
Broad-scale ADCP transects across the Fall of Warness tidal channel (Fig. 1C) reveal the spatial and temporal variability across the site throughout our tidal measurements. As an example, we show vertical profiles of horizontal velocities (Fig. 2A), ADCP-derived back-scatter (Fig. 2B) as well as depth-averaged horizontal velocity vectors (Fig. 2C) along each of the three lines comprising one transect during peak ebb (>3 ms⁻¹) tidal flows (see Supplementary Fig. 1 for other tidal states and Supplementary Table 1 for environmental context). Clearly shown are the sharp horizontal gradients towards the edges of the channel (indicated by the arrows in Fig. 2C), resulting from the island wake effects of Eday to the Northeast and Muckle Green Holm to the Southwest, leading to weaker flows and flow reversals beyond the shear lines. The back-eddy system in the embayment on the Eday side is particularly prominent. The aerial drone overview image visualises the intensity and spatial extent of the surface signatures of the shear line (Fig. 2F, with matching arrow colours to Fig. 2C), dominated by upwelling boils (smooth in appearance), associated downwelling and vortical structures. Figure 2B, D capture the main areas of pronounced ADCP backscatter, a proxy for macro-turbulence entraining bubbles into the water column, mainly associated with the two shear lines. The O2 floating tidal turbine (magenta circles in all figures) is situated at the boundary between the stronger flows in the main

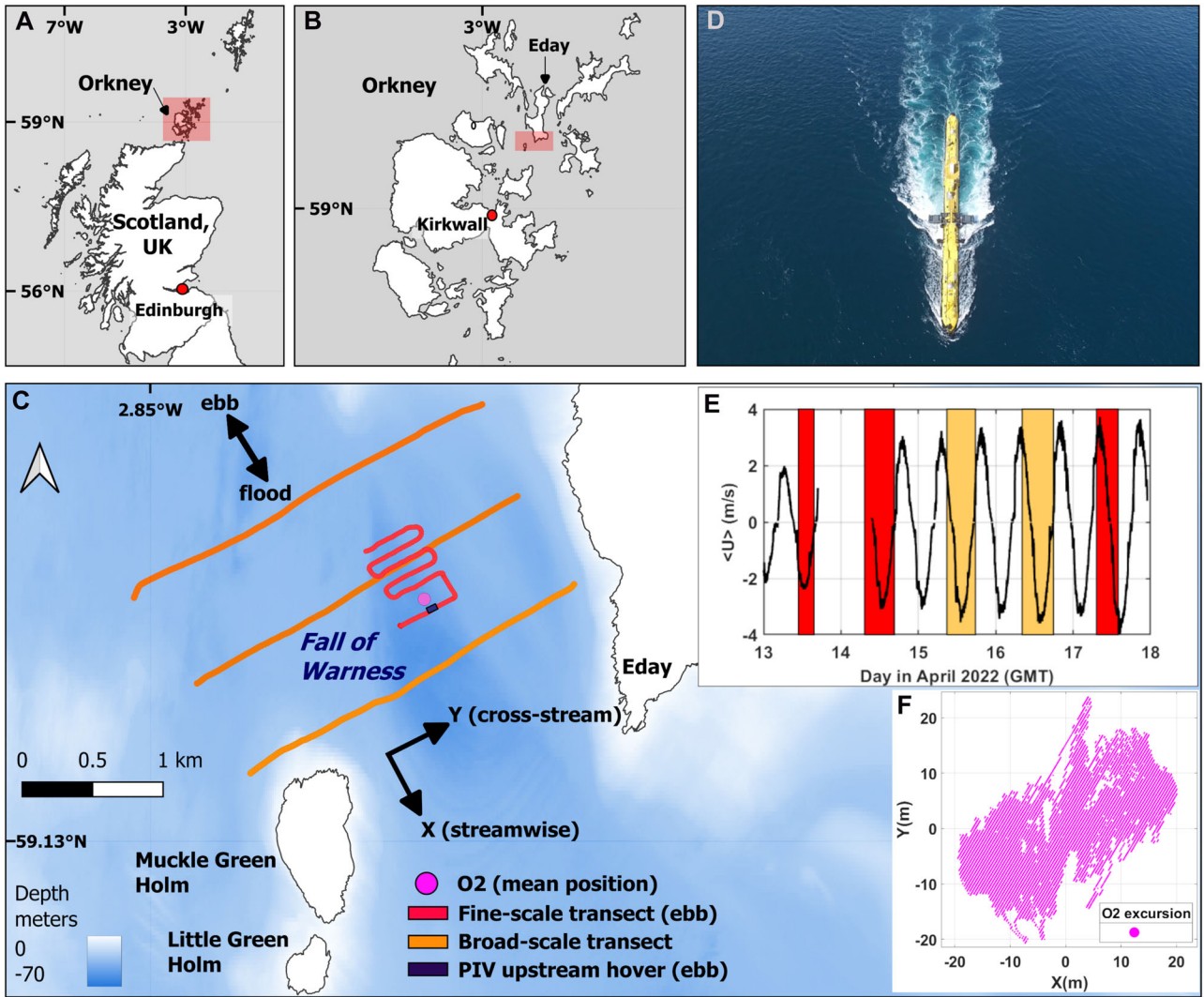

**Fig. 1 | Map of the study site in the Fall of Warness, Orkney, Scotland, UK.**
**A** Overview map showing the location of Orkney off Scotland's mainland, high-lighted by a red box. **B** Location of the Fall of Warness managed by EMEC, high-lighted by a red box. **C** Map of the Fall of Warness coloured by bathymetry showing the mean location of the O2 (59°8.664′N, 2°48.935′W), representative ebb tide fine-scale (14/04/2022, T6) and broad-scale (15/04/2022, T7) transect line data, respectively, as well as an upstream aerial drone hover used in particle image velocimetry (PIV), to scale. The (x,y) axis corresponds to the local coordinate sys-tem used in this study. **D** Aerial drone shot of the idled O2 floating tidal turbine (hull/body overall length = 74 m) deployed in the Fall of Warness taken on 14/04/2022 during peak ebb tidal flows (note, while the turbine was not generating, the rotors were submerged in the operating position). **E** Streamwise depth-averaged velocities extracted from O2 hull-mounted ADCPs highlighting survey periods during the fine-scale (red) and the broad-scale (orange) transects, respectively. Velocities are aligned with the local coordinate system, with positive values on the flood tide. Instruments were not recording during the night of the 13th/14th. **F** Overall excursion (in metres) of the O2 during the sampling period. Bathymetry:© British Crown and OceanWise, 2024. All rights reserved. Licence No. EK001-20180802. Not to be used for Navigation. Boundary-Line™ shape files updated in 2022 using EDINA Digimap Ordnance Survey Service. Source data for E&F are provided.

channel and the weaker flows influenced by the Eday shear line (note the moored location of the O2 platform was determined by available EMEC berth locations). The strong cross-stream shear in the vicinity of the O2 is further demonstrated in the gridded velo-cities shown in Fig. 2E. The streamlines, shown in red with start points distributed at 5D (where D = 20 m rotor diameter) intervals cross-stream of the O2, capture the behaviour of the flow cross- and downstream of the O2 with the flow direction being more variable downstream and towards Eday. There is some indication from the streamlines originating either side of the O2 overlaying increased ADCP backscatter (Fig. 2D), that a wake extends downstream of the platform. However, this is better captured during the fine-scale transects consisting of more narrowly spaced lines set out below. On the flood tide (Supplementary Fig. 1H), the flow direction in the main channel is reversed so that the Muckle Green Holm shear line is

no longer present. However, the Eday shear line and associated flow reversal, induced by the headlands and bathymetry at either end of the EMEC site, persist throughout all tidal stages except slack water.

## Fine-scale flow variability
The fine-scale transects, consisting of one upstream and multiple downstream lines (5D line spacing), provide a more detailed view of the variability in the flows in the direct vicinity of the O2.

As shown in the lefthand-side panels of Fig. 3 (rotated into local coordinate system with the O2 at the origin and Eday towards the positive Y-axis), the large-scale pattern of strong, more undisturbed flow towards the mid-channel (negative Y-axis) and weaker flow mag-nitudes towards Eday can be observed for both the ebb (Fig. 3A, B) and flood (Fig. 3C) transects during strong flows (>2 ms⁻¹). The overlaid vectors illustrate fine-scale horizontal shear, showing the difference in

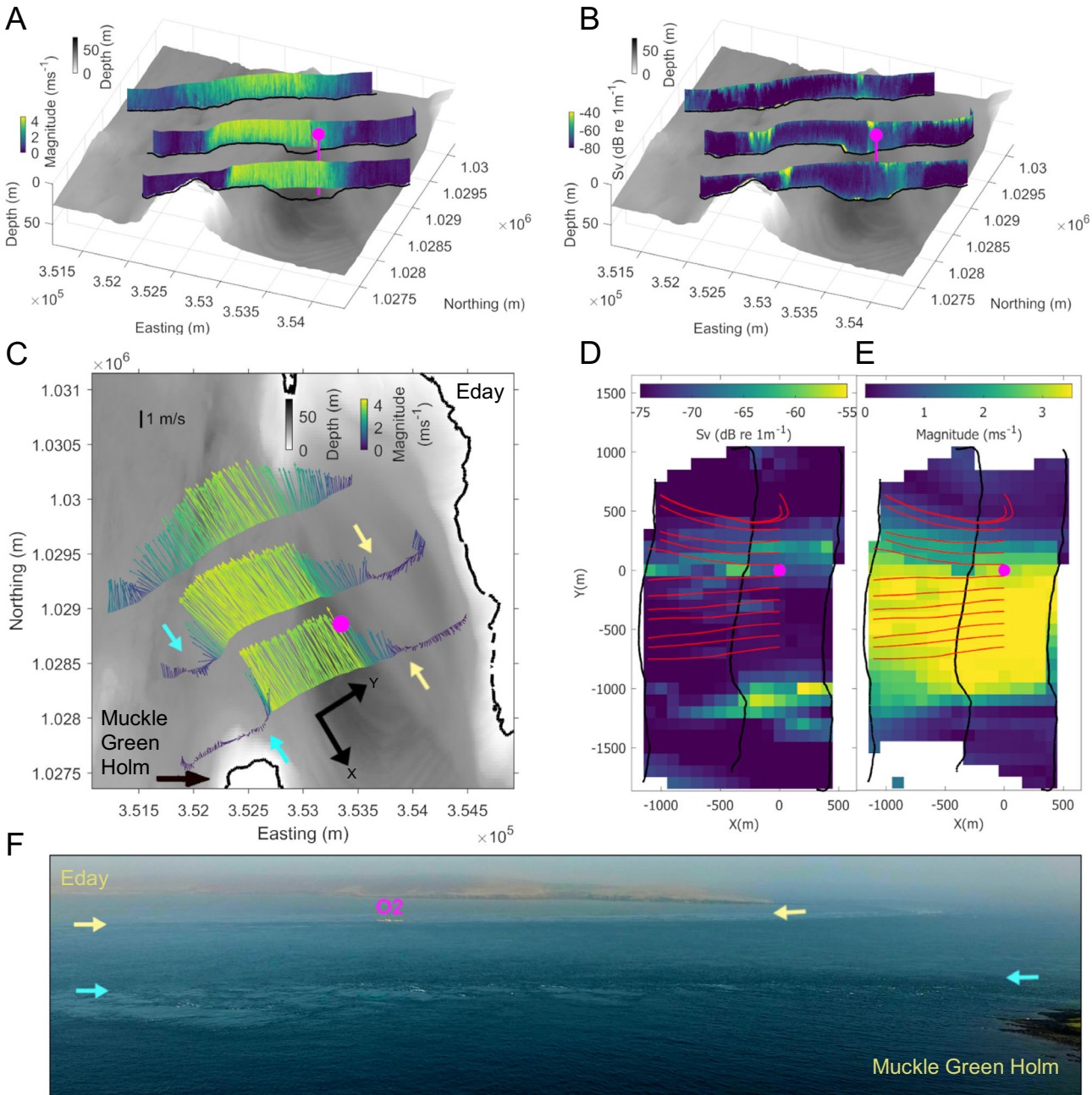

**Fig. 2 | Current velocities and acoustic backscattering across the Fall of Warness from broad-scale ADCP transects. A** Vertical distributions of horizontal velocity coloured by magnitude (ms⁻¹) and, (**B**) ADCP-derived backscatter, a proxy for surface-connected turbulence coloured by backscattering strength (Sv; dB re 1 m⁻¹) during ebb tide (16/04/2022, T5, 3.4 ms⁻¹). **C** Corresponding plot of depth-averaged horizontal velocity with vectors coloured by magnitude.
**D**, **E** Corresponding depth-averaged ADCP-derived backscatter coloured by backscattering strength (Sv; dB re 1 m⁻¹) and depth-averaged horizontal velocity coloured by magnitude rotated to the local coordinate system and gridded at 5D.

Streamlines (red) show the behaviour of the flow cross- and downstream of the O2. For all plots (**A**–**E**), the mean location of the O2 tidal turbine is marked with a circle (magenta). Note, the O2 was not generating with the rotors left idle. **F** Aerial drone overview image visualising the intensity and spatial extent of the surface signatures (lighter linear features) of the shear lines associated with Eday and Muckle Green Holm, with arrows in (**C**, **F**) indicating shear line signatures. Bathymetry:© British Crown and OceanWise, 2024. All rights reserved. Licence No. EK001-20180802. Not to be used for Navigation.

local velocities relative to the upstream reference region (orange area in upstream lines). In all cases, the vectors closest to Eday point upstream of the mean flow direction, indicating relative flow reversal. The streamlines (shown in red; released 2D down- and cross-stream) appear relatively straight and parallel towards the main channel (negative Y-axis), indicating more homogenous flow, while they curve and diverge towards the Eday side (positive y-axis).

## Inflow characterisation

Examining the upstream line of the fine-scale ADCP transects provides detailed insight into the inflow characteristics of the O2 at 5D (100 m). This was the closest, permissible approach that the survey vessel could make upstream of the O2 platform during peak tidal flows. An example during ebb flows is shown in Fig. 4, with other tidal states shown in Supplementary Fig. 2. The inflow region extending ±200 m cross-

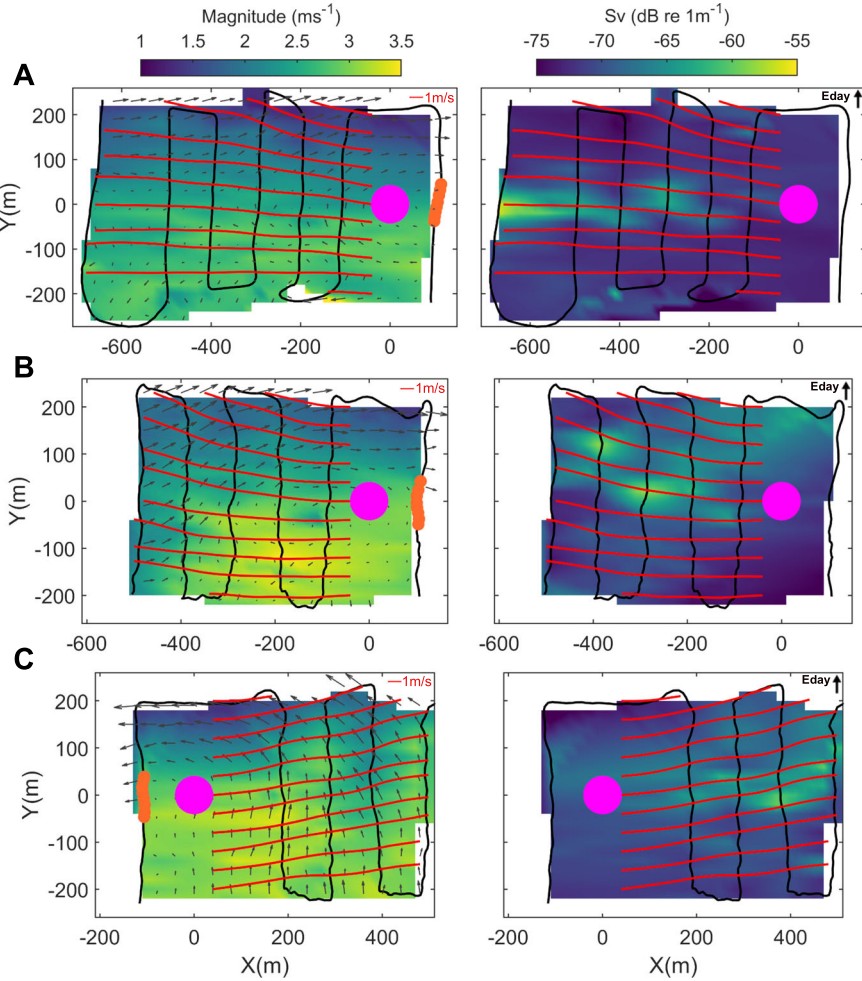

**Fig. 3 | Current velocities and wake tracing around the O2 from fine-scale ADCP transects.** Fine-scale ADCP transect lines consisting of one upstream line and several lines downstream of the O2 marked by continuous black lines. All plots show ADCP-derived depth-averaged horizontal velocity coloured by magnitude (ms$^{-1}$) (left) and backscatter coloured by backscattering strength (Sv; dB re 1 m$^{-1}$) (right) rotated to the local coordinate system with the origin at the O2 location (X,Y) = (0,0), aligned with the mean flow direction and gridded at 1D. The stream-lines (red), released at intervals of 2D cross-stream and 2D downstream of the O2, show the behaviour of the actual flow cross- and downstream of the O2 while the vectors (scale of 1 ms$^{-1}$ shown by red vector on the top right of each plot) show the difference in local flows relative to the average of the inflow region (orange in upstream lines). The O2 location is marked with a circle (magenta) and Eday is located towards the Y positive direction. **A** Ebb flow transect (14/04/2022, T5, mean ebb velocity = 2.6 ms$^{-1}$), (**B**) Ebb flow transect (17/04/2022, T5, mean ebb velocity = 2.8 ms$^{-1}$), and (**C**) Flood flow transect (17/04/2022, T1, mean flood velocity = 3.2 ms$^{-1}$).

stream of the O2 shows a strong cross-stream gradient in streamwise velocities (Fig. 4A) with stronger flows in the freestream (negative Y values) and weaker flows towards Eday (positive Y values). Figure 4B shows the difference in streamwise velocities compared to the mean velocity experienced upstream of the rotor-swept area, highlighting the cross-stream and vertical shear upstream of the rotors in the context of the overall velocity gradients. The Eday shear line is vertically inclined (as indicated by the diagonal dashed line in 4B) and impinges directly upstream of the rotor-swept area of the O2, resulting in both vertical and cross-stream shear affecting the two rotors on either side of the platform differently, potentially even affecting different blades of a single rotor both vertically and horizontally. The depth-averaged streamwise velocity (Fig. 4C) highlights the large variability in flows across the upstream line, with flows of 3 ms$^{-1}$ within the freestream of the main channel and the edge of the O2 inflow region (Y = −200 to −25 m), subsequently (Y > −25 m) dropping to below 2 ms$^{-1}$ towards Eday.

When dividing the upstream ADCP transect line into three sections, as indicated by the white regions in Fig. 4C, the freestream (FS) velocity in the main channel (unaffected by the Eday shear line), the direct inflow region (O2) and the adjacent (S) region (affected by the

Eday shear line), there are marked differences in the spatially averaged vertical profiles of streamwise velocity (Fig. 4D). There is strong vertical shear within the FS region, with higher velocities near the surface. In comparison, there are weaker flows near the surface (<10 m depth) in the O2 region than at depth (reversed shear, corroborated by the O2-mounted ADCP, in magenta), whilst the S region experiences weaker and more variable flows down through the water column. Overall, on the ebb tide, the impingement of the Eday shear line into the upstream area thus results in both inclined, cross-stream shear and variations in vertical shear within the inflow. During peak flood tide (>3 ms$^{-1}$), the O2 inflow is unaffected by the Eday shear line, however the shear line intrudes during the decelerating flood tide (Supplementary Fig. 2M, N).

## PIV on aerial drone imagery

The effects of the Eday shear line impingement in the area immediately surrounding the O2 can be directly quantified using LSPIV (Large-scale Particle Image Velocimetry) techniques applied to the aerial drone imagery. This gives a unique insight into the inflow directly upstream of each rotor simultaneously. The oblique drone shot (Fig. 5A) viewing the O2 from the main channel shows the Eday

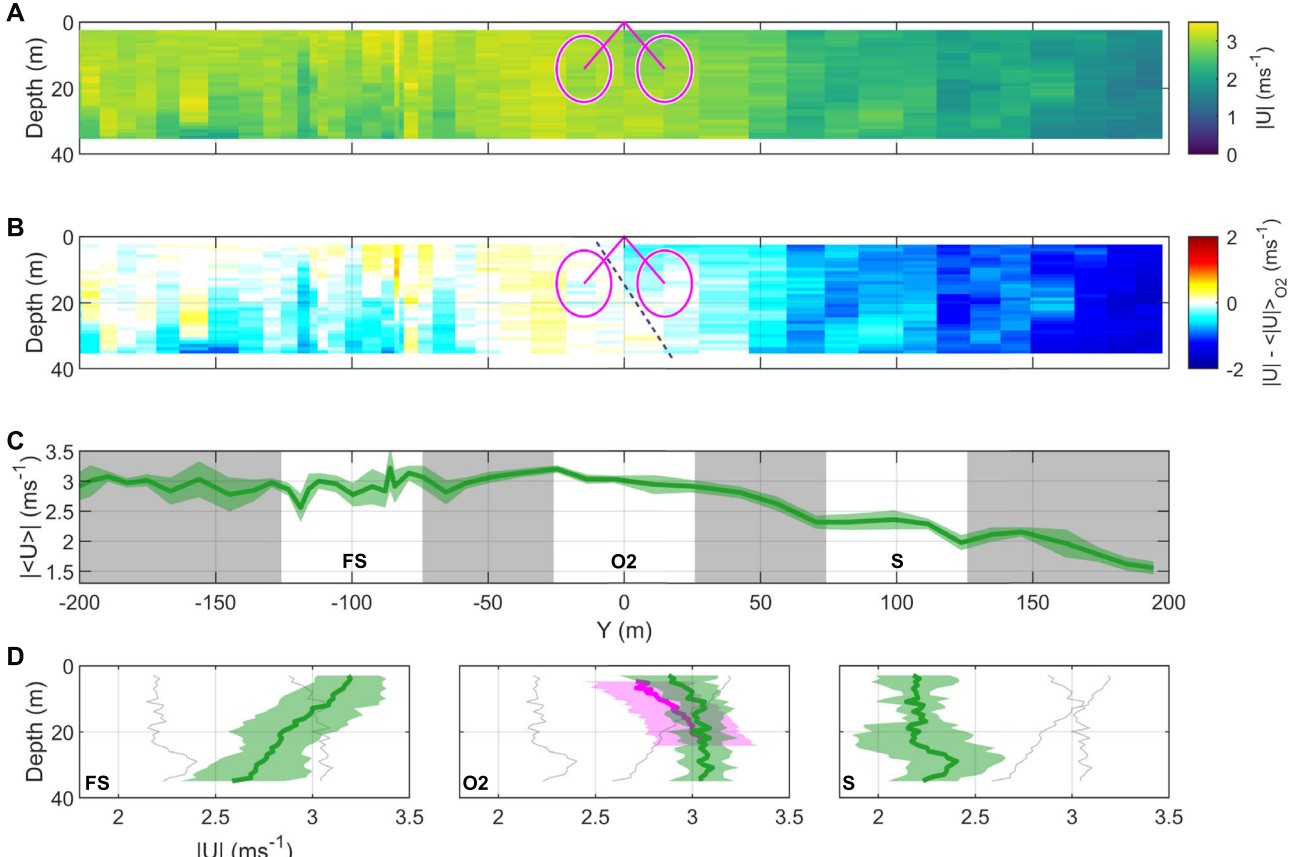

**Fig. 4 | Inflow characterisation from upstream ADCP transect line. A** Ebb flow transect line (17/04/2022, T5, mean ebb velocity magnitude = 2.8 ms$^{-1}$) showing absolute streamwise horizontal velocity (|U| ms$^{-1}$) upstream (100 m or 5D) of the O2. For reference, the downstream location of the O2 is superimposed (hull structure, rotor arms and rotor-swept area in magenta). **B** Corresponding plot showing the difference in streamwise velocity compared to the velocities experienced in the rotor-swept area of the O2 (|U|-<|U|>$_{O2}$ ms$^{-1}$). The dashed line indicates the inclination of the shear line. **C** Depth-averaged absolute streamwise velocity with shading indicating ±1 standard deviation. Three sections are indicated (in white, each section is 45 m in cross-stream extent, representing the extremities of the two rotors), the freestream (FS; centred at Y = -100 m), the direct inflow area (O2; centred at Y = 0), as well as the adjacent shear line region (S; centred at Y = 100 m). **D** Spatially averaged depth profiles of horizontal velocity magnitude (green, shading indicating ±1 standard deviation) of the three sections in (**C**) with grey lines showing mean profiles from the other sections for reference. The middle panel in (**D**) also shows the corresponding mean velocity magnitude profile of the turbine-mounted, downward-facing ADCP during the time of the transect line (in magenta). Source data are provided.

shear line in close proximity beyond the platform, characterised by highly dynamic coherent flow structures (dominated by vortices and boils, defined and visualised by their vorticity and divergence, respectively, in Fig. 5G, H). In Fig. 5B, we show a single video frame presenting the drone's vertically downward-facing field of view during an ebb tide PIV hover over the O2. Incorporating all of the instantaneous (4 Hz) velocity fields over a two-minute hover, we show the mean horizontal surface velocity magnitude and turbulence intensity (TI) in Fig. 5C, D, respectively. The flow magnitude is reduced on the side towards Eday (positive Y) as well as directly adjacent to the (masked) platform. Consistent with the ADCP transects, the flows are strongest in the freestream on the main channel side of the O2 (negative Y). The turbulence intensity (TI) increases towards Eday as well as directly downstream of the rotor arms. Looking at the cross-stream distribution of spatial mean properties, it can be seen that the velocity magnitude (Fig. 5E) decreases across the O2 from about 3.5 ms$^{-1}$ in the freestream to below 3 ms$^{-1}$ in the shear line, consistent with the cross-stream gradient observed in the upstream ADCP line (Fig. 4C). Turbulence intensity (Fig. 5F) increases from 10% in the freestream to more than 30% towards Eday with an associated increase in turbulence length scales (L$_u$/D) from 0.3 to more than 0.6. The increase in turbulence intensity and length scales in the shear line is due to the prevalence of intermittent, intense turbulent coherent structures. An example

of these is shown in the instantaneous vorticity and divergence fields in Fig. 5G, H, respectively. The corresponding figure during the ebb inflow at 5.5D upstream of the O2 is shown in Supplementary Fig. 3. The variability in mean magnitude across the O2 is similarly visible during peak flood tides (Supplementary Fig. 4), however, the prevalence and intensity of turbulent coherent structures is more equal on either side of the turbine and of similar value in TI and length scales in the freestream.

## Wake dynamics

The O2 far-wake dynamics are characterised using water column measurements of ADCP-derived velocity and backscatter data, EK80-derived backscatter and PIV-derived surface velocities. In what follows, all presented measurements are taken from when the O2 turbine was not generating but with the rotors submerged in their idled, operational configuration. Despite this, the wake signal from the overall platform is noticeable and quantifiable in all estimated flow and scattering metrics. The ADCP-derived backscatter data are visualised in the right-hand side panels of Fig. 3. All three transects during both the ebb and flood tide show a strong wake signature downstream of the O2, with bubble clouds entrained into the wake generally intensifying during the second or third downstream line (~200–300 m downstream distance or 10D). The propagation of the wake downstream can be traced using the overlaid streamlines, which show the behaviour of the

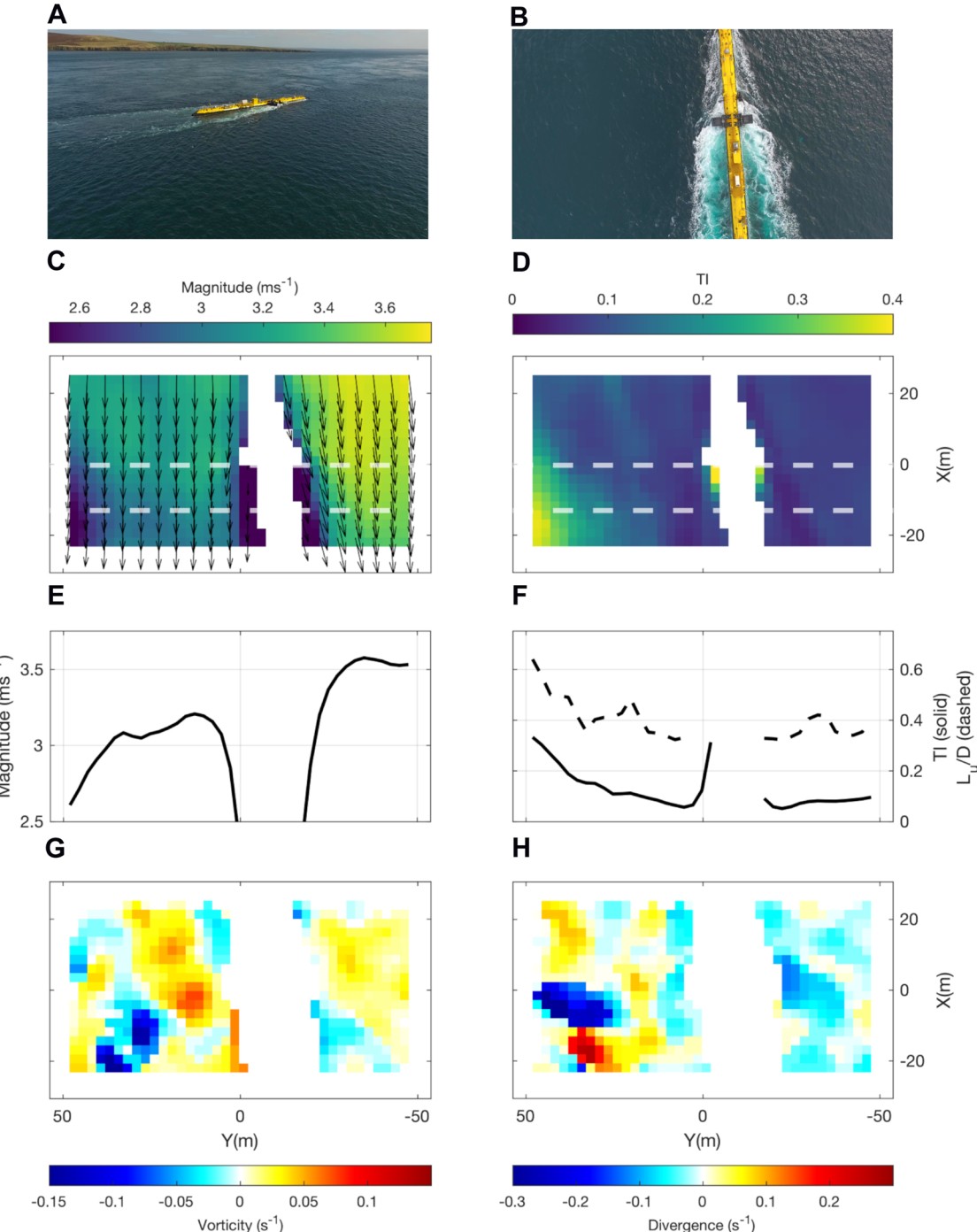

**Fig. 5 | PIV-derived surface current magnitude and turbulence experienced by the O2. A** Oblique aerial drone image approaching the O2 platform during ebb flow (17/04/2022, mean ebb velocity = 3.2 ms⁻¹). **B** Aerial image of drone hover field of view (T6, hover 2, altitude = 65 m) over the O2, with boils visible on the left side of the structure (towards Eday). **C** Mean flow field coloured by horizontal velocity magnitude with velocity vectors overlaid (sub-sampled for clarity) and (**D**), turbulence intensity (TI), as calculated from the 2-min hover. Note, that the O2 has been masked. **E** Spatial and temporal mean horizontal velocity magnitude and (**F**), turbulence intensity both calculated across the region bounded by dashed, white lines in (**C**, **D**), highlighting the

difference in horizontal velocity magnitude and TI on either side of the platform. The dashed lines in (**F**) are the turbulence length scales $L_u(\Delta X)$ normalised by the rotor diameter D. Regions of (**G**) vorticity (positive = anti-clockwise) and (**H**), divergence (positive = upwelling) and convergence (negative = downwelling) on either side of the O2 (frame = 237, same as in **B**). The local coordinate system is centred around the mean O2 location. The left side of the O2 (Y = 0–50 m, closer to the Eday shear line) is characterised by lower velocity magnitudes, higher turbulence intensity, larger turbulence length scales and more intense regions of vorticity and divergence. Source data are provided.

flow cross- and downstream of the O2 location. Notably, wake propagation between the two ebb flows differs and may be associated with flow variability at larger scales as there is no clear difference in wind speed and direction or wave height between the two transects.

Figure 6 shows a representative ebb flow transect with water-column ADCP-derived velocities in 6A and corresponding EK80-derived backscatter in 6B. While the wake is not clearly visible in the velocity data, the backscatter clearly visualises the wake propagation

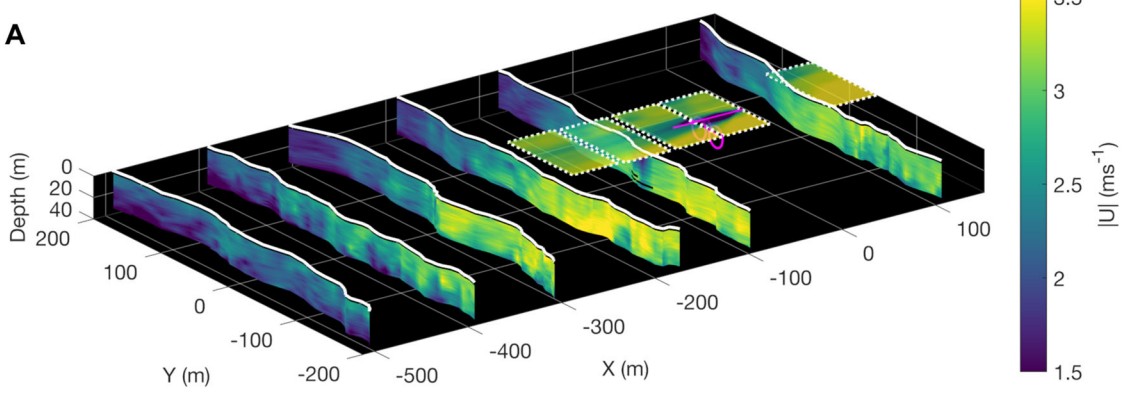

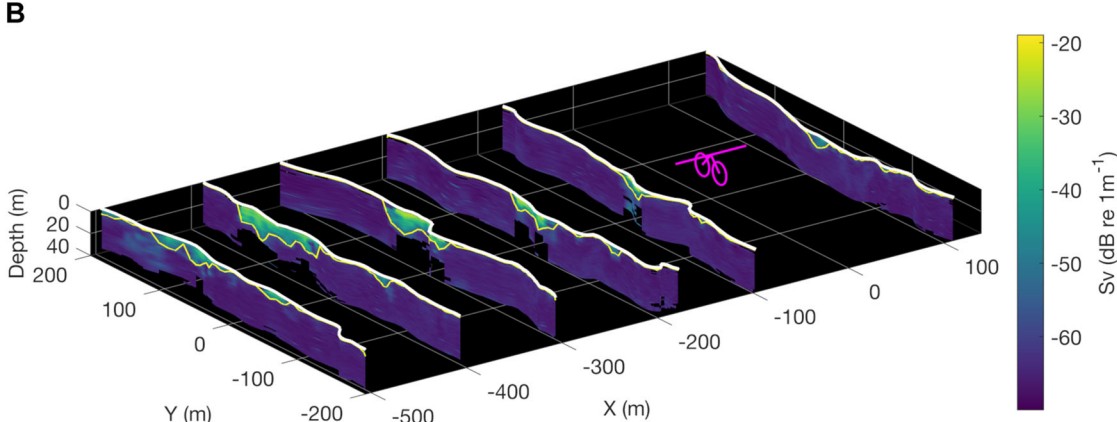

**Fig. 6 | ADCP-derived water column streamwise velocity with PIV hover locations and concurrent EK80-derived wake tracing. A** Curtain plot of ADCP-derived absolute streamwise velocity (|U| ms⁻¹) along fine-scale transect lines during ebb tide (17/04/2022, T5, mean ebb velocity = 2.8 ms⁻¹) up- and downstream of the O2 located at X = Y = 0 (showing hull structure, rotor arms and rotor-swept area in magenta). PIV hover locations (upstream, over and downstream of the O2) are visualised as rectangles filled by mean surface current magnitude demarcated with white-dashed boundaries. PIV hovers started 17 min following the upstream ADCP line. **B** Concurrent EK80-derived backscatter coloured by backscattering strength ($S_v$; dB re 1 m⁻¹) visualising wake-induced bubble entrainment, demarcated with yellow lines depicting the observed depth of surface-connected acoustic scattering. For both plots, the white solid lines demarcate the surface track followed by the survey vessel.

through bubble entrainment. The wake during this ebb transect propagates towards Eday, which is aligned with the ADCP streamlines presented in Fig. 3B. In this transect, the wake first expands and deepens (-5D–20D downstream; Fig. 6B), before starting to dissipate during the last line (>25D downstream).

We subsequently analysed the concurrent EK80 backscatter data, isolated surface-connected bubble entrainment through macro-turbulence and were thus able to trace the cross-stream dynamics in the Y-dimension (mean cross-stream location in m), the spread (cross-sectional area in m²) and backscattering strength ($S_v$; dB re 1 m⁻¹) of the wake as a function of downstream distance from the O2 at a higher resolution compared to the ADCP-derived backscatter (Supplementary Fig. 5). As a result, the wake dynamics for all transects where flow speeds exceeded 0.75 ms⁻¹ (leading to generation of a wake) are summarised in Fig. 7. Downstream of the O2, there is an increase in the variation in cross-stream location of the wake with increasing downstream distance (Fig. 7A), with this 'meandering' being independent of the mean velocity magnitude but with more variation on the ebb than the flood tide. The area (Fig. 7B) does not notably change during the first 100–200 m downstream (5–10D), while it almost doubles in extent further downstream (>200 m downstream or >10D) when the mean velocity magnitude exceeds 1.5 ms⁻¹ with no discernible difference between the ebb and flood tides. Further, during these higher flow speeds, the backscattering strength (Fig. 7C) of the wake is generally

highest immediately downstream of the O2 (5D) and only decreases as the wake dissipates further downstream. Finally, to assess full wake recovery, we performed one longer ebb tide transect, extending the downstream lines to 1600 m (80D) downstream of the O2 (Supplementary Fig. 6). Here, the wake's physical scattering could be traced up to 1300 m downstream (65D).

Overall, the wake forms immediately downstream of the turbine (see drone image in Fig. 1D) and can be characterised by shallow, intense bubble entrainment at the first downstream line (-100 m downstream or 5D). The bubbles then start to disperse in the cross-stream and vertically (as deep as 20 m) as they are advected downstream.

### Wake velocity deficit from PIV

Following analyses of all three PIV downstream hovers (2.75, 5.5 and 8.25D) during both peak ebb and flood tidal flows, the wake's velocity deficit ($U_D$) is discernible across all downstream distances, despite the spatial gradient in flow velocities and in the absence of the O2 generating (Fig. 8A). However, the most notable velocity deficit (>20%) is measurable at 2.75D, extending ±15 m in the cross-stream. At larger downstream distances, the peak in $U_D$ is less pronounced but tends towards Eday (positive Y), consistent with Fig. 3. Figure 8B shows the corresponding variation of turbulence intensity (TI) and turbulence length scales ($L_u$/5D). On the ebb tide, TI increases towards the Eday

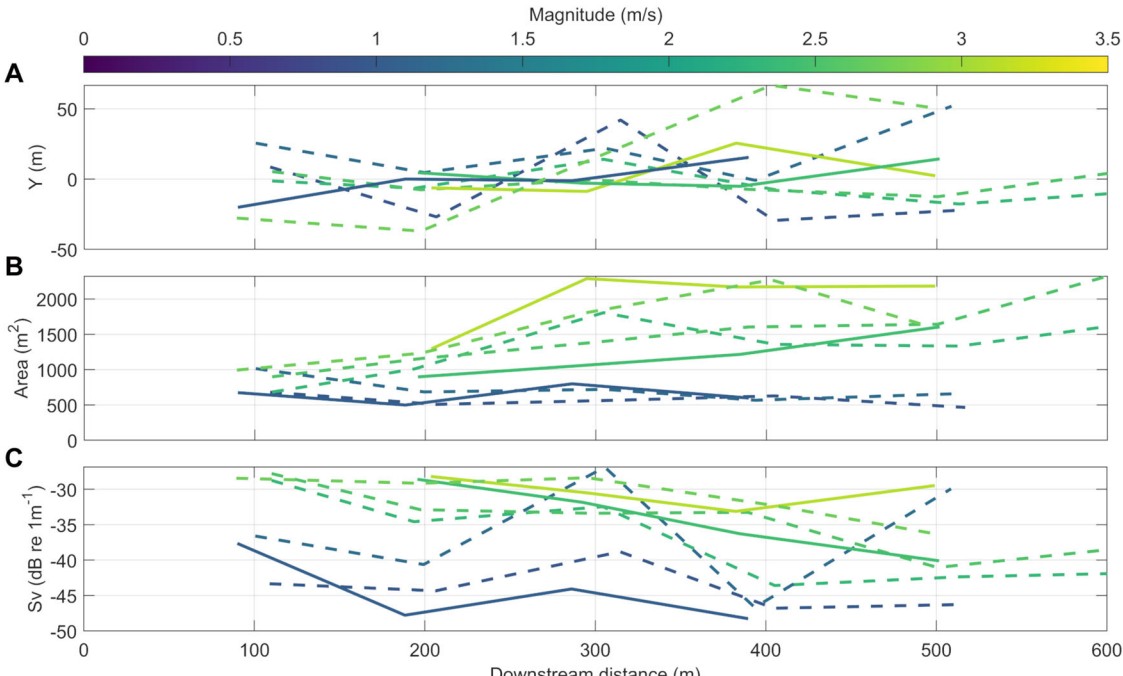

**Fig. 7 | Wake propagation, spread and backscattering strength among fine-scale transects. A** Mean cross-stream (Y) location as a function of downstream distance from the O2 platform. **B** Wake cross-sectional area (spread) as a function of downstream distance. **C** EK80-derived backscatter strength (Sv; dB re 1 m⁻¹), a proxy for surface bubble entrainment through macro-turbulence. Transects are coloured by underlying mean horizontal velocity magnitude and include those from the 14th and the 17th of April 2022, where mean magnitudes range from 0.75 to 3.2 ms⁻¹. In all plots, solid lines indicate flood transects and dashed lines indicate ebb transects. Source data are provided.

shear line (Y > 30 m) but is relatively constant across the O2 wake area and in the freestream. $L_u/5D$ is elevated within the downstream wake area of the O2. As previously noted, on the flood tide, the distributions of both TI and $L_u/5D$ are more equal across the extent of the sample area, with turbulent coherent structures prevalent near the shear line, the wake and the freestream area. The corresponding spatial distributions of the PIV data at 5.5D as per Fig. 5, are provided in Supplementary Figs. 7, 8 and show that TI is relatively constant and weak (10%) within the wake, especially when compared to near the shear line where TI more than doubles. However, turbulence length scales increase in the wake area due to the larger scales of spatial coherence found here. We subsequently assessed the variation of fluctuations in the velocity components over time at three single locations: (1) within the O2 wake, (2) near the shear line and (3) the freestream area (Fig. 8C). Near the shear line (S), the streamwise component (u') shows strong intermittent reductions in velocity (-2 ms⁻¹ positive fluctuations of the ebb-direction of mean flow) which are indicative of large and intense turbulent coherent structures being advected through the sample area. The freestream (FS) area shows notable velocity fluctuations (-0.5 ms⁻¹), however, the wake area (O2) is characterised by much smaller amplitude and higher frequency fluctuations. The cross-stream component (v') shows similar scale fluctuations (-0.5 ms⁻¹) near the shear line and freestream area, but again smaller amplitude and higher frequency fluctuations within the wake. Therefore, although the wake signature is highly discernible visually (Supplementary Figs. 7, 8), the velocity fluctuations within it are less intense and of similar amplitude in both horizontal components (resulting in small TI but higher $L_u$) compared to the natural turbulent coherent structures in the adjacent areas.

**Wake velocity deficit from ADCP**
To compare our PIV-derived surface velocity deficits to the ADCP water column data, we extracted the nearest up- and downstream lines of the corresponding ebb and flood tide transect, respectively. Due to the O2's excursion from its mean centre point across tidal states (Fig. 1F) and the requirement to maintain a safe operational distance, the first downstream ebb tide transect line was at ~4D, and the flood tide transect line was at ~9D. Figure 9 shows a sharp decrease in along-channel velocity at rotor depth (left upper panel), even though the turbine was not generating. During the ebb tide, the velocity deficit at 4D reaches nearly 20% at rotor depth, while it is less pronounced closer to the surface (10%), which is in accordance with the PIV-derived surface velocity deficit of ~10% at 5D. The flood tide deficit (lower panels) is lower (12%), consistent with the increased downstream distance of the transect line (approx. 9D). The scattering from entrained bubbles (Fig. 9, right panels) is most intense near the surface, up to 10 m depth.

## Discussion
Our study presents field measurements of inflow and wake dynamics around an idled, utility-scale floating tidal stream turbine set within Europe's largest tidal turbine test site. Our combination of conventional and innovative measurement approaches (e.g. Fig. 6) helps to bridge between idealised simulations and the real-world complexities of natural turbulent flows, allowing us to make several contributions set out below. We also discuss our findings in relation to the energy-environment nexus with implications for other floating platforms.

First, the O2 is located within relatively strong cross-stream gradients in mean streamwise velocities, highly influenced by the proximity of the Eday island wake and associated shear line (Fig. 2). This heterogeneity in flows across the Fall of Warness was already noted in previous modelling and field campaigns using ADCP point measurements[10,11,53,54]. However, unlike tidal stream sites with similarly complex flow dynamics, including Ramsey Sound[20] (Wales, UK), Alderney Race[23] (English Channel) or the Grand Passage[28] in the Bay of Fundy (Canada), the Fall of Warness has previously not been characterised using mobile ADCP transects. Our spatial transects, therefore, not only help to validate existing coarse hydrodynamic

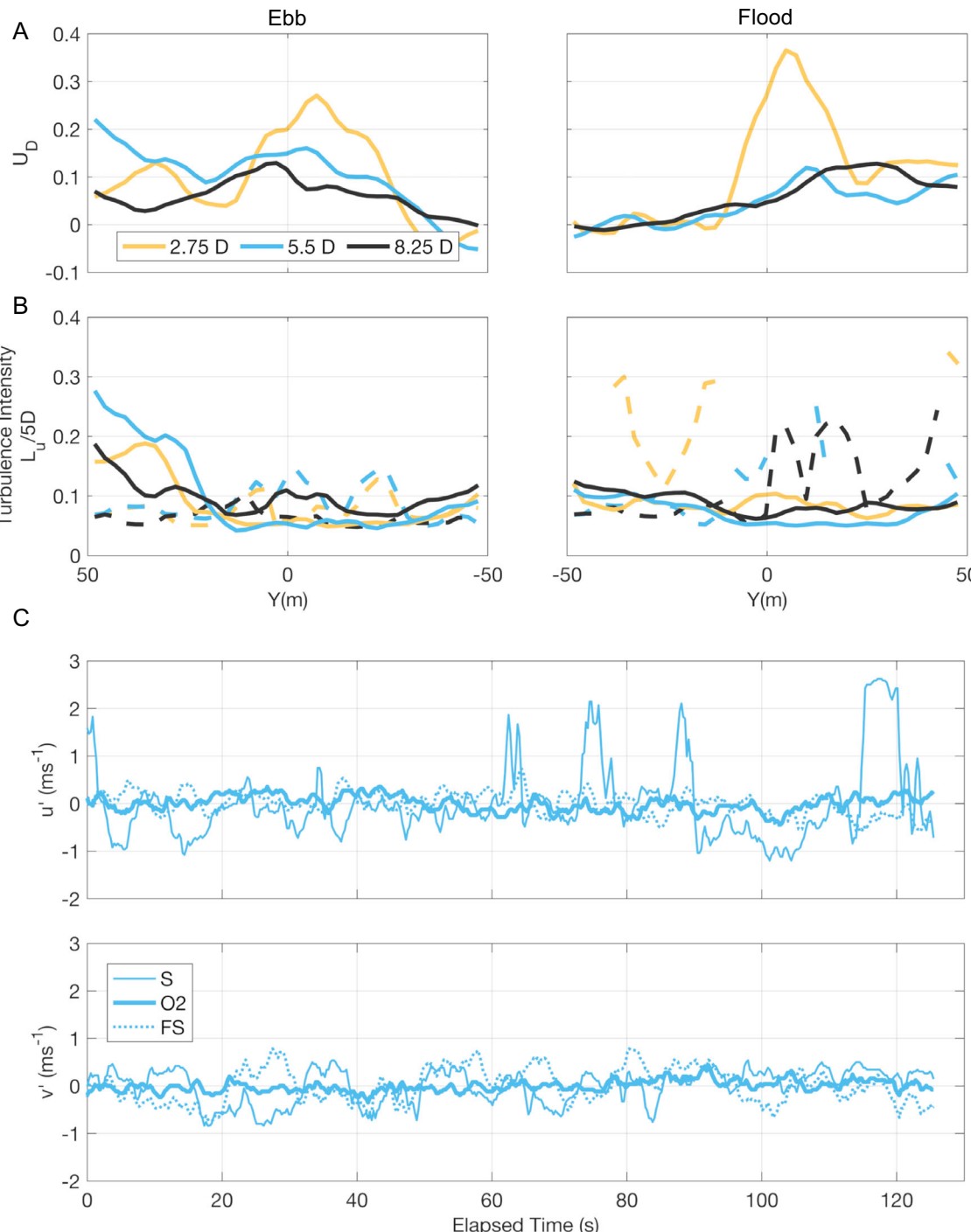

**Fig. 8 | PIV-derived surface wake velocity deficit and turbulence characteristics as a function of downstream distance and surface velocities comparing the O2's wake with the adjacent natural flows. A, B** Peak ebb (left, 17/04/2022, T6, 3.2 ms⁻¹) and flood (right, 17/04/2022, T1, 3.2 ms⁻¹) hovers at different downstream distances indicated by colours. **A** Streamwise spatially averaged wake velocity deficits ($U_D$). **B** Streamwise spatially averaged turbulence intensity (lines) and turbulence length scales (dashed lines, $L_u/5D$). **C** Velocity fluctuations at 5.5D during peak ebb showing u' (upper panel) and v' (lower panel) components over time at single PIV vector locations within the O2 wake (O2, Y = 0.3 m), near the shear line (S, Y = 48.2 m) and the freestream area (FS, Y = -47.6 m), respectively. Source data are provided.

models, but also provide a baseline ahead of planned arrays with details in fine-scale flow complexities (Figs. 3, 4). For example, upstream bathymetry-generated turbulence is known to lead to vertical distributions of velocity[16,55] that cannot always be approximated to a logarithmic profile through the entire water column[56,57], including homogenised[20,23] or even reversed[53] vertical profiles. Here, the cross-stream impingement of the shear line into the inflow region of the O2 generates a reversed vertical shear profile with

reduced current speeds in the upper water column (at hub height of the O2), most pronounced during strong (>2.5 ms⁻¹) ebb tidal flows (Fig. 4) and consistent with previous ADCP point measurements from this area[53]. Our data, therefore, demonstrate that a cross-stream impingement of a topologically-generated shear line provides an additional route to 'shaping' vertical shear profiles, an important consideration for geographically similar sites. Our spatial measurements thus provide further evidence that vertical shear

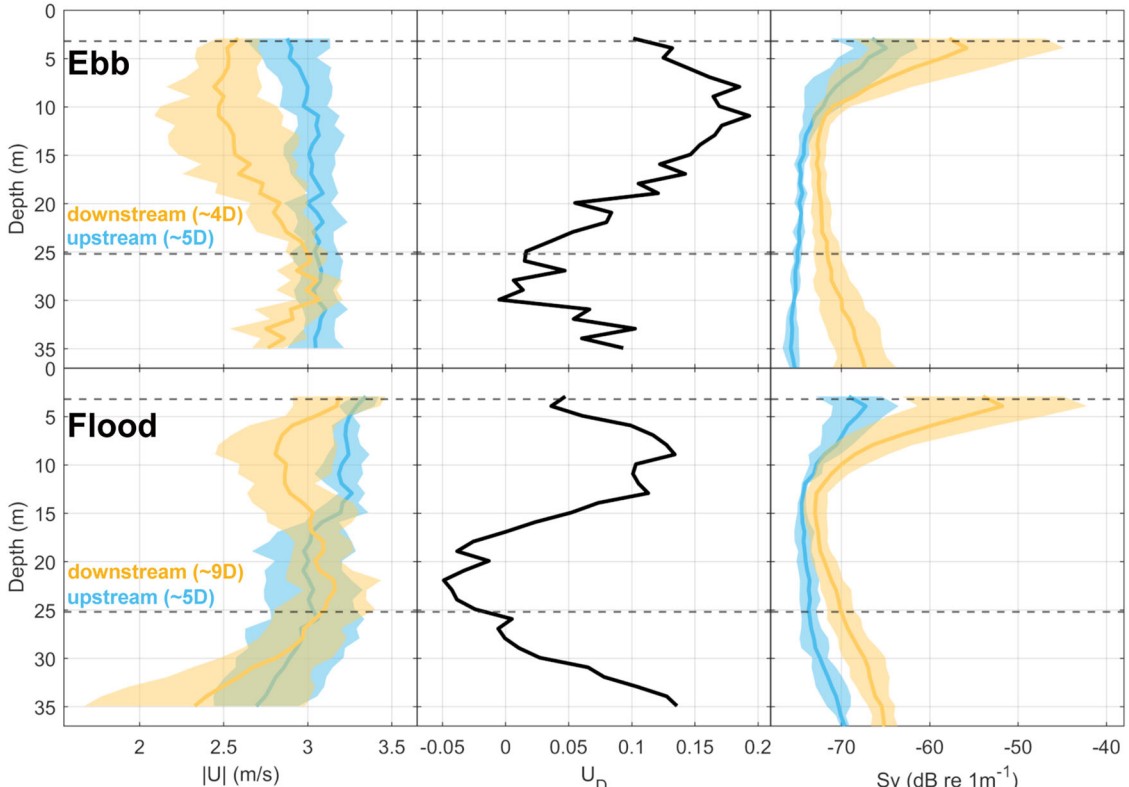

**Fig. 9 | ADCP-derived wake velocity deficit during ebb and flood tide, respectively.** Ebb tide (upper panels, 17/04/2022, T5, mean ebb velocity = 2.8 ms⁻¹) and flood tide (lower panels, 17/04/2022, T1, mean flood velocity = 3.2 ms⁻¹) averaged vertical profiles of absolute streamwise velocity (|U|, left) upstream (blue) and downstream across the O2 wake area (orange), the calculated velocity deficit $U_D$ (middle), and the ADCP-derived backscatter (right), a proxy for bubble entrainment by macro-turbulence. The profiles have been spatially averaged over a cross-stream extent of 45 m, centred at the O2 location. Note the difference in up- and downstream distances during each tidal state. Shading indicates ±1 standard deviation. Dashed lines indicate the O2's rotor-swept area. Source data are provided.

profiles in actual, complex flows may deviate from conventional power-law distributions[9,11]. As our method of using moving vessel surveys is limited by compromises between spatial coverage and accurate determination of velocity profiles at single locations, it could be improved by utilising multiple instrumented vessels simultaneously (e.g. performing up- and downstream lines simultaneously).

Second, our surface flow spatial measurements using PIV techniques on aerial drone hovers provide evidence of the prevalence of turbulent coherent structures in the inflow region near the shear line, resulting in reduced velocity magnitudes and corresponding increased turbulence intensity and length scales (Fig. 5), indicative of the spatial variation in flow parameters. Our novel PIV-derived TI measurements are consistent with other sites[10,58,59] and are in the order of 10% in the freestream, while TI increases to >20% in the area characterised by coherent structures (Fig. 5F). This is consistent with our measurements of turbulence length scales that are highest ($L_u/D > 0.5$) near the shear line. Given our findings on vertical shear variability in the cross-stream (Fig. 4D), it is rather unlikely that aerial drone measurements at this site could be reliably translated to currents at depth[42], however, aerial imagery is useful in highlighting regions of flow variability that will require further investigation using profiling instrumentation. Whilst we were limited to using a single drone, our method could be expanded by using multiple stabilised (RTK-GPS) drones flown simultaneously, which would provide temporal synchronisation between inflow and downstream regions, increase spatial coverage, improve on-surface resolution and enable additional insight for the modelling of multiscale, multiphysic regimes. All inflow and site characterisation in this study are independent of

the turbine generating, and together, our findings are relevant to help inform the design, placing and testing of tidal stream turbines. The cross-stream (or lateral) velocity gradient shown in Fig. 4 could result in a 50% difference in thrust between the freestream and shear line (100 m cross-stream distance), with an 18% difference in thrust experienced by each individual rotor at the present O2 location. With new generations of tidal stream turbines incorporating larger rotors, and, consequently, larger cross-stream extents, this will result in even higher levels of thrust differences between rotors. Rotor loading imbalance, due to cross-stream shear, and turbulent coherent structures are all aspects that should inform device design[18] as these fine-scale flow features will impact device loading whether generating or not. Although not considered here, platform motion under the combined effects of currents and waves can further amplify rotor load fluctuations, potentially leading to damaging effects[60]. The variability in onset conditions combined with extensive platform excursion (exceeding 40 m in both the cross- and along-stream directions; see Fig. 1F and Supplementary Table 1) also has implications for power curve testing. Power curve standards following the International Electrotechnical Commission Technical Standard 62600-200[61] currently recommend two independently located ADCPs deployed up- and downstream of the device ('in-line', preferred option) or else 'adjacent' to the device. This approach is designed for mono-rotor, seabed-mounted (i.e. static) devices, which is understandable given their prevalence to date. We show that this approach is incapable of capturing the spatially varying flow conditions across a dual-rotor floating platform (cross-stream extent of 45 m) within the bounds of the varying device position. Our measurements thus provide valuable insight for power performance assessments given the added

complexity presented by dual-rotor, floating tidal stream turbines. The data also demonstrate that over-reliance on simplified, broad-scale modelling may lead to poor decision-making in the placement of moored ADCPs.

Third, we demonstrate that our small-scale transects with resulting streamlines and wake isolation provide a synoptic approach to map out wake dynamics. Despite the O2's idled status, our methods provide first insight into wake signatures which would be intensified if the turbines were operational (Figs. 3, 6, 7). Wake propagation in the cross-stream is highly variable (particularly on the ebb tide with cross-stream variation of ±50 m), an important consideration when evaluating idealised wake modelling for array planning. Simulations of offshore floating wind platforms have recently shown that the side-to-side motion of platforms can be a novel origin for the onset of wind turbine wake meandering[62] which may also be applicable for the O2 platform given its cross-stream movement (e.g. up to 17 m in the cross-stream within 20 min, Supplementary Table 1). Generally, the position of the centre of the wake tends towards Eday, consistent with our streamlines indicating advection, with enhanced recovery likely associated with turbulent coherent structures and associated increases in turbulence intensity and length scales near the shear line. We further show that wake spread (Fig. 7B) increases with flow speed as it is advected downstream, consistent with numerical simulations[63]. While not considered here, as waves were negligible during the small-scale transects, the inclusion of waves would likely further impact wake propagation and recovery[32]. The ADCP-derived backscatter visualises entrained bubbles sufficiently to provide a highly complementary measure for wake tracing. There are, of course, inherent trade-offs in selecting instrumentation for tracing acoustic backscatter. For instance, multi-channel broadband echosounders, in combination with scattering models, would provide improved sampling resolutions and the determination of additional acoustic scattering sources[64]. However, such approaches would require multiple transducers to cover a broad frequency range, instrument calibration, physical sampling and scattering model expertise. In contrast, ADCPs present low-cost, off-the-shelf systems readily deployable making them more suitable for more routine monitoring.

We show that the wake's velocity deficit is measurable at hub height in the ADCP transect data (Fig. 9) and at the surface through PIV measurements (Fig. 8 and Supplementary Figs. 7, 8) despite the sheared flow and in the absence of the O2 generating. For operating submerged tidal turbines, wake effects are generally still noticeable at 10 rotor diameters (D)[30], with some effects, such as velocity fluctuations and flow skewness still present beyond 10D[65]. Our ADCP-derived velocity deficits (18% at 4D and 13% at 9D) are comparable to those estimated downstream of the floating PLAT-I platform in the Bay of Fundy, which showed a 10% velocity deficit at 10D during the non-operational state[28]. Wake measurements presented in this study are relevant to the future build-out of tidal stream turbine arrays and our methods can be repeated during periods with power generation. There is currently a lack of wake measurements downstream of operating floating tidal stream turbines (but see ref. 28), preventing an in-depth understanding of turbine–turbine interaction from being developed. This is also important when considering the planned 30 MW tidal array in Westray Firth (8 km northwest of the Fall of Warness), also fringed by small islands. Our PIV measurements highlight that turbulence intensity measures are less informative compared to turbulence length scales for determining wake characteristics, especially when adjacent natural flows contain large coherent turbulent structures. As above, our method could be improved using multiple drones with increased resolution.

Finally, anthropogenic turbulence and associated vertical mixing has been documented previously in flows past other energy platforms, such as a tidal turbine monopile structure set in a tidal channel[25], offshore wind turbines with seabed foundations in well-mixed shallow water[66] and was most recently suggested for floating offshore wind platforms set in seasonally stratified deeper waters[67]. Our combined approach of wake tracing and drone hovers can be directly applied to investigate wake propagation (spread and velocity deficit) in all these cases. Even floating wind turbines will have a minimum draft of 20 m (similar to the O2's operational draft of 24 m) and albeit exposed to weaker tidal flows (e.g. in the Celtic Sea where currents speed exceed 0.5 ms$^{-1}$), wake signatures are likely still measurable[67]. Essentially increasing the physical footprint of floating tidal devices, downstream wake effects may not only pose a challenge for realised energy capacities, but will also lead to changes to the physical environment (e.g. wake-induced bubble clouds and velocity deficits as seen here or changes in tidal elevation, velocities, mixing or stratification for very large-scale arrays of several GWs of installed capacity[68–71]) which could result in altered marine fauna habitat use, distributions and population dynamics[25,72–75]. Energy extraction and environmental effects are coupled, and ecological considerations may restrict how much of the tidal resource can be exploited in a constrained flow passage[4] such as the Fall of Warness. However, it is also important to consider that comparatively, the projected ecological effects of climate change on marine habitats could be an order of magnitude larger compared to the effects of even very large-scale energy extraction[71,73].

## Methods
### Study site
Surveys were performed in the Fall of Warness (FoW), a tidal stream situated between the islands of Eday and Muckle Green Holm, in between Westray Firth to the north and Stronsay Firth in the south, Orkney, Scotland, UK (Fig. 1). The FoW site is characterised by semidiurnal tides with current magnitudes exceeding 3.5 ms$^{-1}$ during spring tides[76]. The mean tidal flow is orientated from the northwest (315–330°) to the southeast (150°) during the flood, with the reverse during the ebb tidal phase. We define the axis 330 to 150 degrees as 'streamwise' (X) for the purposes of our analysis, with 'cross-steam' (Y) directed towards 60 degrees (Fig. 1C). The peripheries of the FoW are dominated by large, kilometre-scale eddies most prominent to the west of Eday and to the east of Muckle Green Holm[77,78]. While the tidal flow is largely ebb-dominant in Westray Firth and flood-dominant in Stronsay Firth, this tidal asymmetry (variations between the flood and ebb phases of the tidal cycle) is less pronounced in the FoW[76].

### Floating tidal turbine
The European Marine Energy Centre (EMEC) manages the test site within the FoW and provides eight grid-connected tidal energy test berths, ranging from 12 to 50 m in overall depth. The O2 is moored at Berth 5 (59°08.712'N 002°48.999'W) in water depths of 40–45 m below chart datum. The O2 has a 74 m long hull with twin 1 MW power-generating nacelles (contra-rotating; cut-in current speed is 1 ms$^{-1}$) at the end of retractable leg structures. The rotor diameter (D) on each nacelle is 20 m, with a maximum draft of 24 m depth during operation. When the legs and rotors are parked horizontally in operational mode, the cross-stream extent of the platform (tip to tip) is 45 m. The O2 platform experiences platform excursion exceeding 40 m in both the cross- and along-stream directions (movement during tidal cycle, see Supplementary Fig. 8) and experiences more than 5° difference in turbine heading from flood to ebb tide (Supplementary Table 1). During the time of the surveys, the O2 was not generating due to maintenance. However, the leg structures were submerged in operational mode, with the rotors left idle, but the platform itself generated a visible surface wake (see Fig. 1D). Therefore, even with the turbine not generating, surveys were designed to quantify both the inflow and wake dynamics as a function of distance to the O2 platform.

## Transect design

Data were collected in the FoW during spring tides (April 16th = new moon) on five consecutive survey days between April 13th and 17th 2022 (Fig. 1E). Repeat parallel-line transects perpendicular to the dominant flow direction (cross-stream) were performed onboard an 18-m-long catamaran vessel (The Green Quest, Green Marine UK Ltd). Travelling at a constant vessel speed of 5-7 knots through the water, the boat provided a stable platform for acoustic data collection and a large forward unobstructed deck area (24 m²) for aerial drone operations. Two types of transect surveys were performed, smaller-scale transects around the O2 turbine (hereafter 'fine-scale' transects), and 'broad-scale' transects across the FoW (Fig. 1C). Transects were designed to best capture variation in hydrodynamics across all tidal states.

On the more fair-weather days (April 13, 14 and 17), fine-scale parallel-line transects were run up-and downstream of the O2. On the 13 and 14, each transect started on the upstream side (1 line), followed by consecutive transect lines (4–11) downstream of the O2 to characterise both the inflow and the downstream wake and associated flow fields. On the 17th, this was reversed, and transects started downstream, with the final line upstream. Line spacing was ~100 m (5D), and each line was ~400 m in the cross-stream designated by waypoints and followed manually by the vessel skipper (see Supplementary Fig. 9 for all fine-scale vessel tracks). Transects were repeated approximately every hour to capture flow variation across tidal states. Each transect was preceded by aerial drone surveys (see section below). Broad-scale transects were conducted on the more inclement weather days (April 15 and 16) and consisted of three lines per transect (~600 m line spacing, ~2500 m in the cross-stream). With higher winds and associated sea states more than 2 on the Beaufort scale, these days were not suitable for aerial drone surveys, however conditions were less restrictive for acoustic instrument operation. These broad-scale transects provided context on the wider flow interactions with the O2 as well as the characterisation of the spatial variation in flows across the FoW. A summary of all transects is provided in Supplementary Table 1. Most tidal states were covered during the broad-scale transects, while the fine-scale transects missed the onset of the flood tide (Fig. 1E).

## ADCP data collection and processing

A downward-facing Teledyne RDI Workhorse Monitor ADCP (600 kHz) operating in Mode 1 with bottom-tracking (Mode 5) enabled was mounted on a side pole (submerged 1.35 m) on the starboard side of the vessel. The ADCP was configured to sample ensembles at an interval of 4.22 s (each ensemble comprised 5 water column pings and two bottom-track pings) with 55 vertical bins of 1 m (ambiguity velocity, the maximum observable along-beam velocity, was set to 4.62 ms⁻¹). A USB-connected GPS was linked to the incoming ADCP data stream acquired with VMDas software to provide navigational information during transects. The ADCP's internal compass was calibrated (5.08° error) on the morning of April 13 following mobilisation in Kirkwall. As part of the standard quality control procedures, ADCP data were post-processed in WinADCP (v. 1.14; RD Instruments, Inc.) using default parameters, and data were checked for anomalous pitch and roll. True water velocities were computed by subtracting the bottom-tracked boat velocity. A threshold for good velocity data was set to 35 m depth to omit dubious velocity readings near the maximum range of the ADCP. ADCP velocities were rotated to the local coordinate system. Acoustic scattering in the water column was extracted using volume-backscattering strength ($S_v$, measured in decibels, dB). This was calculated over a finite volume (maximum of 40 bins) from the ADCP's recorded raw echo intensity data using a working version of the sonar equation as described in ref. 79 and updated in ref. 80 using a Kc coefficient of 0.45 dB/count for all four beams. $S_v$ was evaluated separately for each bin along each of the four beams of the ADCP. For each range bin, the maximum of the four beams was taken to create depth profiles of the maximum level of scattering through the water column.

For each transect, a frozen-field[81] approximation was made to allow the spatial visualisation of the depth-averaged data. Broad-scale transects were gridded at 5D, while the fine-scale transects were gridded at 1D in both streamwise and cross-stream directions. Gridding was undertaken using two-dimensional linear interpolation. The non-dimensional ADCP-derived wake velocity deficit ($U_D$) is defined as:

$$U_D = \frac{U_\infty - U_W}{U_\infty} \tag{1}$$

where $U_\infty$ and $U_W$ are the upstream and within-wake velocities, respectively. $U_\infty$ was extracted from the direct inflow region of the upstream line with a cross-stream extent of 45 m centred around the O2 location.

Additionally, near-continuous data were collected by ADCPs installed on the O2 platform. These were two downward-facing RDI Workhorse Monitor ADCPs (600 kHz) recording water column velocities at 1 Hz and a vertical bin size of 0.5 m with a range of 20 m (40 depth bins). These were mounted one at each end of the main O2 hull, such that data uncontaminated by the platform's wake was recorded on both flood and ebb tidal cycles by the upstream ADCP. This uncontaminated data were merged into a single time series and bin-averaged at 15-minute intervals, to provide depth-mean velocities, as shown in Fig. 1E.

## Echosounder data collection and processing

A downward-facing Simrad split-beam EK80 echosounder operating a 38/200 kHz single beam combi transducer (ES38-200-18C, 18° beam-width) was mounted on a moonpool pole located on the aft deck (nominal depth of transducer face = 1.3 m below sea surface). The EK80 was configured for 'normal' operation (continuous wave single frequency pulse at full power; 250 W for 200 kHz, 500 W for 38 kHz) with a ping interval of 500 ms for both frequencies, and a pulse duration of 0.256 ms for the 200 kHz, and 1.024 ms for the 38 kHz frequency, respectively. Acoustic data were collected using the standard EK80 software (Simrad). Analyses focused on capturing near-surface scattering from surface-entrained bubble clouds by macro-turbulence, e.g. turbulent coherent structures, the wake of the O2 platform as well as island wakes. Data sets were initially processed in Esp3, an open-source software for visualising and processing active acoustics data[82]. Inclement weather and turbulent conditions dominating high-flow environments can cause attenuation of the transmitted and/or received acoustic backscatter due to entrained air bubbles and increased noise from a variety of sources[83]. After visual inspection of both frequencies, analysis was focused on the 200 kHz frequency as it provided useful data closer to the sea surface and showed less signal attenuation (due to strong scattering from surface bubbles) compared to the 38 kHz data. The in-built algorithms in Esp3 were applied for seabed removal (bottom detection), signal attenuation (bad transmits, spikes and dropouts) due to impulse or transient noise, and near-surface data were removed to exclude echosounder transmit pulse in the transducer nearfield (~1 m). The time varied gain (TVG) noise correction following ref. 84 was applied in Esp3 to reduce the effects of signal degradation and TVG-amplified background noise apparent below intense surface wake signatures. Factory calibration settings were used as the characterisation of physical scattering was predominantly intended to give a relative measure of bubble entrainment by macro-turbulence in this high-flow environment. Data were manually edited where needed, e.g. to re-define the seabed around abrupt changes in bathymetry or in areas of potential sediment resuspension. All data were gridded at a resolution of 5 pings along the

track by 0.5 m depth bins, and integrated mean volume-backscattering strength data (hereafter $S_v$; dB re 1 m$^{-1}$; logarithmic, see ref. 85 for definitions of acoustic variables) were exported applying an $S_v$ threshold of $-70$ dB re 1 m$^{-1}$. Data were then further processed in Matlab (Mathworks, R2021b) and the processing steps involved in wake isolation and tracing are outlined in Supplementary Fig. 5. To delineate surface bubble entrainment, image processing was applied to the $S_v$ data, retaining only surface-connected scattering in the water column using a $-50$ dB threshold (Supplementary Fig. 5C). The applied de-noising and thresholding sufficiently reduced the impacts of excess attenuation and multiple scattering to isolate and trace the O2 wake (Supplementary Fig. 10). The wake propagation was then traced by calculating the mean cross-stream location in the Y-dimension (m), the cross-sectional area (spread in m$^2$) and the maximum backscattering strength ($S_v$; dB re 1 m$^{-1}$) of the surface-connected scattering within cross-stream sections (Y = ±100 m) downstream of the O2 platform (Supplementary Fig. 5D–F).

### Aerial drone surveys and processing

Aerial drone surveys allowed flow measurements close to, and over the O2, thereby complementing boat transects which had to maintain a safe distance to the platform, especially upstream during strong flows. Aerial drone surveys consisted of multiple hovers up- and downstream of the O2 to characterise the inflow and its wake signature across tidal states (Fig. 6). All surveys were performed by the same qualified (UK Civil Aviation Authority) pilot, assisted by an observer who was in control of releasing and catching the drone during take-off and landing. Drone missions were performed using a DJI Phantom 3 Advanced quadcopter recording 2 K video at 30 Hz. Drone hovers (holding station at a given altitude with a vertically downward-facing camera) were performed upstream (5.5D), over (0D) and downstream (2.75, 5.5 and 8.25 D) of the O2, aligned with the mean flow direction. The hover locations were pre-planned in Matlab, using the mean O2 location, and then exported into the Litchi flight application. The hovers were performed at 65 m altitude, holding station for 130 s at each location (we refer to these as 'two-minute hovers' throughout). The hover duration was a compromise between reliably characterising the turbulent flow at each location and sampling all locations within a short-enough total period to allow direct inter-comparison under the assumption of Taylor's frozen-field approximation[81]. We assessed the convergence of statistics (mean velocity and turbulence intensity)—see Supplementary Fig. 11—showing that they converge in the freestream regions but show less good convergence near the shear line due to the prevalence of large-scale turbulent coherent structures there. The drone's altitude resulted in a footprint of 107.97 m × 61.40 m on the water surface with a resolution of 4.02 cm per pixel. Instantaneous velocity fields (with a spatial resolution of about 2.6 m), calibrated using the drone's altitude, were extracted at 0.266 s (8 frames) intervals through each drone hover using the particle image velocimetry (PIV, also commonly referred to as 'large-scale PIV'; LSPIV[38]) method described in ref. 41 and fully detailed in the Supplementary Methods 1. These were then used to derive the instantaneous vorticity and divergence fields using the standard MATLAB functions (curl and divergence, respectively), as well as calculate cross-stream distributions of spatial and temporal mean properties (velocity magnitude and turbulence intensity; TI). TI was calculated as

$$TI = \frac{\sqrt{\overline{u'^2}}}{U} \qquad (2)$$

where u' is the turbulent fluctuations and U is the location-specific time-mean velocity. Turbulence length scales ($L_u(\Delta X)$) were estimated by calculating the mean correlation as a function of streamwise separation ($R_{u,x}(\Delta X)$) prior to integration up to the separation needed

for the correlation curve to decrease to $1/e$. This method follows ref. 86 and is examined in more detail in the Supplementary Methods 2.

An assessment of drone stability is given in Supplementary Methods 3 and Supplementary Table 2 which gives a conservative upper bound for a mean error in PIV-derived velocity measurements of 0.149 ms$^{-1}$ which for our typical flow speeds (order 2 ms$^{-1}$) results in a TI sensitivity threshold of 0.075. Finally, the PIV-derived wake velocity deficit was derived in a similar manner to the ADCP $U_D$, using the immediate inflow region (45 m cross-stream extent) of the upstream hover.

## Data availability

All processed ADCP data are available here: https://doi.org/10.24382/244fae5d-2d16-4219-98f6-7aa96757ae49. Source data are provided with this paper.

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

## Acknowledgements

L.L. acknowledges the Supergen ORE Hub ECR Research Fund and the EPSRC for funding this project. The authors wish to thank the European Marine Energy Centre (EMEC) for their guidance and financial support for this project and gratefully acknowledge Catherine Tait and Ana Couto for joining the surveys. Thanks to James Waggitt as the dedicated seabird observer. The authors thank Orbital Marine Power for their in-kind contributions. L.L. acknowledges additional support provided by the Bryden Centre (2017-2022), which was supported by the European Union's INTERREG VA Programme, managed by the Special EU Programmes Body (SEUPB). D.C. acknowledges the financial support of the Tidal Stream Industry Energiser project (TIGER), which is co-financed by the European Regional Development Fund through the Interreg France (Channel) England Programme.

## Author contributions

L.L. and W.A.M.N.-S. conceived the ideas and L.L., S.F. and W.A.M.N.-S. designed the methodology. W.A.M.N.-S. collected the aerial drone data (CAA-approved pilot), L.L. and W.A.M.N.-S. collected the ADCP data, and S.F. collected the EK80 data. L.L., S.F. and W.A.M.N.-S. performed analysis and LL., S.F., D.C. and W.A.M.N.-S. interpreted the results. L.L. drafted the manuscript, and all authors contributed critically to the drafts and gave the final approval for publication.

## Competing interests

The authors declare no competing interests.
