## [Peer Review File · Nature Communications]

Sheared turbulent flows and wake dynamics of an idled floating tidal turbineREVIEWER COMMENTS

Reviewer #1 (Remarks to the Author):

Review of "Troubled waters: Revealing sheared turbulent flows and wake dynamics of a floating tidal turbine"

This investigation presents new flow measurements around the Orbital O2 tidal turbine located in the Fall of Warness, UK. The paper presents measurements collected with a mobile acoustic current Doppler profiler, echo sounder, and with a drone following a PIV technique. Results show new information about the turbine's inflow conditions and about the wake generated by the floating tidal energy platform and its turbines.

Field measurements around tidal turbines are scarce, mainly because there are only a few operating full-scale tidal turbines in the field and because collecting data in energetic tidal sites is difficult and expensive. I find the data presented in the paper to be relevant to the tidal energy industry and necessary for evaluating the hydrodynamic environmental effects of tidal energy extraction. However, there are aspects of the paper that need to be greatly improved before publication.

To ease this review, I will answer the following questions provided in this journal review guidelines.

What are the noteworthy results?

The data itself is relevant and unique, especially the acoustic wake measurements and PIV velocity estimations from drone images, which is a novel technique barely applied in the field.

The wake description, although simple, is noteworthy because most wake velocity deficit estimates come from numerical simulations, and there are not many field data sets about it.

Will the work be of significance to the field and related fields? How does it compare to the established literature? If the work is not original, please provide relevant references.

Yes, it will be relevant for the tidal energy field and for the field of fluid dynamics.

The work is original because it is new data in a different site, and around a turbine that to the best of my knowledge very few studies have been published. Prior work regarding spatial flow measurements has been cited. However, the paper does not include in the introduction nor in the discussion comparisons with other sites and with other turbine wakes either from models or from the field data.

There is vast literature about turbulence measurements at tidal energy sites that is not included in the paper, and I think is important to include it to provide context and to better discuss the obtained results.

Does the work support the conclusions and claims, or is additional evidence needed?

The flow description is mostly qualitative and includes snapshots of what is happening in this variable unsteady environment. Perhaps authors could include or summarize the results for different tidal conditions.

Another main concern is that the authors describe the turbine wake and claim to have measured or detected wake meandering. Turbine wake meandering has been previously observed in laboratory experiments and in detailed computer fluid dynamic simulations of flow passing through a rotating turbine. This feature has been linked to the interaction between the rotor shed vortex and the counter-rotating the tip-vortex shed by the rotating turbine blades.

In this paper, the authors indicate that the turbines were not operating, meaning the turbine blades were not rotating, so the authors need to justify how they observed the wake meandering feature and what mechanism is producing the meandering.

Are there any flaws in the data analysis, interpretation and conclusions? Do these prohibit publication or require revision?

Yes, the publication requires revision.

Another important concern is about the estimation of turbulence metrics from the PIV data. Authors use only 2 minutes of data to obtain turbulence statistics, in previous works it has been established that to get robust turbulence statistics in unsteady tidal flows, at least 5 minutes of data are needed, in order to have statistical stationarity and to capture a wide range of turbulence length-scales. The authors need to justify their choice of time-series length and discuss errors in their calculations.

Is the methodology sound? Does the work meet the expected standards in your field?

The field measurements techniques are sound and are state of the art. ADCPs are the instrument of choice in oceanography for measuring velocities, and post processing seems to be PIV is a widely used technique in the lab, not common in the field.

Authors should specify in the methods how is the PIV data processed, include sources of error and mention encountered difficulties. Also, need to specify how other quantities such as divergence and vorticity were estimated.

ADCP processing seems to be standard, but authors should specify how the gridded products were obtained.

Is there enough detail provided in the methods for the work to be reproduced?

As mentioned before, more details need to be included in the methods section to reproduce all the data post processing.

Specific comments:

The draft does not have page numbers nor line numbers, which would be helpful for conducting the review. Here I detailed some questions and suggestions regarding figures, results, and methods.

Figure 2:

Are the top panels instantaneous data? It is very hard to see the details in Panels A and B

Figure 3:

How were these data gridded and interpolated? Is not fully specified in the methods section. Is the data collected along the black lines used here for interpolation? How many transects are included in the interpolation?

Authors need to specify at which depth were these data collected or if it is depth averaged.

Arrows in the left-hand panels are misleading, arrows typically show flow direction rather than a difference with upstream velocity. Perhaps the authors can include a velocity deficit color plot.

Also, there should not be data over the platform location, since the platform is supposed to be there, hence no data can be collected on top of it.

Figure 4:

Panel D, figure 5: how do you explain the reversed shear near the bottom?

Also in the text: what do you mean by "the shear line intrudes" and by inclined later shear?

Supplemental materials Figures S2 are impossible to read.

Figure 5:

Vorticity and divergence: Authors should explain how those quantities were estimated in the methods. Vorticity is a vector, I assume this is vertical vorticity. There is only one snapshot of vorticity in Fig. 5, but there is no description or explanation on the sign of vorticity and if the distribution of vorticity makes sense in the text. Same with divergence, why is divergence relevant for wake dynamics?

Figure 6: It would be really helpful if these data were presented in space rather than in time, as are the rest of the data presented in the paper.

Figure 7: In panel A is very hard to distinguish the reduced wake velocities.

Why does the acoustic wake signal deepen if bubbles are buoyant? What could be driving the bubbles to move downward?

Figure 9 and 10:

I find this to be one of the most important or more relevant result of the paper.

Regarding, TI, what is the error of TI estimation with PIV?

Discussion

The discussion section would be greatly improved if comparisons were made with other available datasets and models. Discussion mostly focuses on their own data and how their data is relevant.

For example: How do the wake velocities compared to previously estimated wake velocities obtained from models (if available). How does turbulence intensity compare to other estimates? How does the wake vary with different tidal conditions? What can you expect for a operating turbine wake?

Supplemental Figures S1 and S2 are not readable.

Reviewer #2 (Remarks to the Author):

The authors present measurements and discussions about ocean currents at a floating tidal turbine in the Fall of Warness, UK. The measurements were conducted via multiple instruments, e.g., ADCP, PIV, beam combi transducer, and aerial drone surveys, and different data were recorded, such as ocean current speed and turbulence fluctuation.

I. Intellectual merit

1) I would say such work., e.g., ocean measurements, is interesting and beneficial to the community, given that it is lacking and desired for ocean energy development, especially their multiscale features (flows at and away from the turbine). Also, the work may promote understanding and modeling (in its validation) of ocean flow problems other than floating tidal turbines, such as floating offshore wind energy. Additionally, the submission is well-written, although a little too long.

2) Basically, this submission is a case study, and its results are location-dependent. As a result, the understanding presented in this paper is mainly restricted to the location of this study. The work may be further enhanced so its results may bear insight for other sites, if more analysis (as mentioned in the following paragraph) is provided, such as info to show the complete evolution/swinging of the wake over full cycles of tides. Others are comparison of the results to those without turbine/floating device and those at turbine operation conditions, via modeling if not measurements.

3) The work primarily describes the measurement results and corresponding flows, and it lacks insight and analysis on the flows patterns and mechanisms. For instance, there is lack of discussion on correlations between speeds/turbulence/flows at different locations from the turbine, to reflect multiscale features of the flows. Additionally, it would be interesting if results/measurement are provided for flows at the turbine and those further away, e.g., further away in the southeast and northwest directions.

II. Presentation problems

4) Fig. 3 --- for the velocity field on the left, the color bars shows that dark and light color mean low and high speed, respectively. But the length of vectors in them seem to be in the opposite way. For instance, the color shows high speed in the lower right corner of the left figure in Fig. 3, but the vector length there indicates low speed there. Do I miss something?

5) Some results/analysis may be better presented. For instance, the discussion about Fig. 8, i.e., the paragraph right before the figure is unclear. I don't see an explanation of each of the lines in the figure.

6) From Fig. 9, visually, it seems that U_d and U have certain correlation at $5D$ and $7.5D$. Moreover, V has such a relation at $2.5D$ and $7.5D$, and it is about zero, exhibiting no correlation at $5D$. Are those true, and if yes, why?

7) This work is valuable for ocean modeling modeling (e.g., its validation). Therefore, it helps if more details are provided with this regard, such as adding time and date of the measurements in figure captions.

8) I cannot judge much on measurement techniques but wonder if the measured data by ADCP mounted on the vessel are affected by the blocking/wake from the vessel itself. An discussion on this helps.

Overall, I think this work has impact on understanding of the influence of the turbine on the flows and engineering problems at its location. However, it lacks intellectual merit, particularly science impact/insight on sciences and flow mechanisms. After revision, the submission may fit a domain science journal, such as Ocean Eng., Coastal Eng.

Reviewer #3 (Remarks to the Author):

Key Results

The Authors present results of an experimental campaign at the Orbital Marine Power tidal turbine platform in Scotland UK. Through acoustic profiling and analyzing aerial drone imagery, they attempt to reconstruct the flow conditions around and in the wake of the turbine. To the best of my knowledge, this is the first experiment of flow characterization around a tidal turbine. Experimental results show that the presence of areas of high turbulence intensity in the inflow region of the turbine may reduce the current speed at the rotors. This may have implications for the turbine's structural stability and power generation. Such high flow variability is attributed to the particular configuration of the site (presence of islands).

Validity

The Authors do not support experimental findings with data on the power generated at the turbine. Even if the turbine was not functioning during the experiments, power data generated during similar tidal events are not included/mentioned in the paper. It is then impossible to ascertain whether flow variability has an effect on the turbine performance. Data gathered during aerial surveys is not provided and a few details are included on their processing pipeline. It is difficult to assess whether aerial imagery was of enough quality to be processed. Given the extremely challenging environmental settings during the experiment (I expect wind to be rather severe at the Fall of Warness), the drone vibrations may have significantly impacted images and velocity estimations. Also, velocities estimated from PIV tend to be rather variable. Typically, longer time intervals help to mitigate this aspect. However, I could not find any reference on the velocity temporal standard deviations in the paper.

Significance

Since experimental results are very much connected to the specific site where the turbine is installed, I do not expect the paper's conclusions to highly impact the field of ocean energy extraction. The paper may suggest that the performance of the O2 turbine is not optimal, but it does not introduce any new findings on power generation from tides.

The Authors claim that they develop new survey methodologies, however, I do not find their observational techniques particularly novel. Acoustic current profiling is a well-established technique. PIV was first introduced in the 1990s and aerial PIV from drones has also been adopted in many settings, mostly in riverine but also in coastal areas.

Data and methodology

Regarding data acquisition, I think the Authors should have better commented on how they ensured that transects were parallel and, therefore, data profiles were consistently parallel. I

expect navigation at the experimental site not to be easy and, thus, some corrections to the data could have been made.

ADCP data were collected from an instrument mounted on a pole submerged 1.35m. Did the Authors apply any correction to get data up to the sea surface?

More details should have been provided regarding aerial drone surveys and processing:

- Wind conditions during the hovers, wave conditions, and seasonality at the site should be included.
- Appearance of the sea surface: the presence of features is fundamental to acceptable PIV data
- PIV parameters: the Authors mention Ref. Lieber et al., 2021. However, experimental settings may have been diverse and PIV parameters should have been properly adapted.
- Drone stability is not discussed. The Authors used a commercial drone platform, which is not capable of compensating for high wind conditions. In Lieber et al., 2021, the Authors mention that "There were no static reference points within the camera's field of view so that whole-field contamination from the relative motion of the UAV cannot be removed". However, this is not the case for the O2 experiment, where the turbine (or part of it) could have been regarded as static during the 2 minutes-long hovers.
- PIV only provides a velocity characterization of the sea surface. I believe the Authors should have better clarified and discussed this.
- Standard deviations of the PIV velocity fields are not provided.

Suggested improvements

I suggest the Authors to take into account information on the power generation performance of the turbine. Such data could help in truly assessing whether the O2 is performing well or is negatively affected by the island wakes.

PIV data should be stabilized before processing.

Clarity and context

Generally, I had difficulties in identifying the mean streamwise and cross-stream direction in Figures. I think such pieces of information should be properly included in all Figures.

Figure 3 is of poor quality.

I found it difficult to identify the O2 in Figure 6.

Visualization of the ebb and flood tides is also not easy. For instance, this sentence: "The ebb and flood tide upstream distance to the O2 is 103 m and 108 m, respectively, while the ebb and flood tide downstream distance to the O2 is 86 m and 189 m, respectively." would be clearer through a Figure sketch.

References

I think the Authors should consider similar experimental studies that make use of aerial drone surveys for non-invasive flow characterization.

Reviewer #4 (Remarks to the Author):

Review of the manuscript: "Troubled waters: revealing sheared turbulent flows and wake dynamics of a floating tidal turbine" by Lieber et al. submitted to Nature Communications.

Overview:

This manuscript presents the results of a resource characterisation at one of the berths at the EMEC site in Scotland where an idled floating tidal stream turbine is deployed, using vessel-mounted ADCP and drone imagery to measure the velocity field. Results capture the three dimensional nature of the tidal flow at this location which is highly influenced by the local bathymetry and nearby headland, which induced flow curvature and large levels of shear. Such findings will be of use for the tidal turbine developer to characterise loading when device is in operation.

General comments:

Whilst the analysis of the time-averaged velocity field from ADCP and drone is good, there are not enough insights into the turbulence characteristics that are developed during the different tidal

phases measured. Unfortunately, the turbine is in parked position which does not generate a significant wake throughout the water column as the results evidence. Moreover, the use of free-surface velocity measurements through drone imagery is very recent as is the need for further understanding and/or correlation of such measurements with the velocities through the water column, which the authors did not analyse. In many instances the authors link ADCP signal to bubbles as a signature of turbulence, but this seems quite speculative, e.g. the idled turbine blades would not undergo cavitation and thus they will not generate air bubbles but fluctuations to the velocity field.

Another key fluid-mechanics aspect that has received little attention is the wave field, acknowledging this would be difficult but aiming for a publication in Nature Communications would require such extensive analysis. Authors claim that during two ebb cycles the results differ notably. Thus, the question is what drove those changes, were waves very different in period, direction or height?

The manuscript has several flaws regarding the description of the flow dynamics:

- ♣ In figure 5 you present "divergence" field, but you have not defined what that parameter is and how is calculated.

- ♣ Figure 9 B. This plots the time evolution of the velocity series in absolute magnitude whilst authors indicate these are velocity fluctuation, which is an erroneous concept.

- ♣ Authors refer many times to "Shear line", but this is not a concept used in fluid mechanics. Perhaps you aimed to refer to the shear layer but this is unclear.

Further analysis of the PIV field would be required, e.g. authors refer to those shear lines but without quantifying Reynolds stresses, time-averaged vorticity, etc.

Finally, authors claim that IEC standards do not consider deploying two measuring devices on the lateral sides of the turbine. Please review the standards because they do present this very clearly in Figure 4 of the IEC 62600-200 2013.

Recommendation:

The reviewer considers that the manuscript does not have a rigorous analysis of the fluid dynamics of the flow at this site, instead it provides a good resource characterisation in terms of mean velocity distribution and time series. Unfortunately, the turbine is not operating, reducing the impact of this study in the community. The recommendation is to reject this paper. Hopefully the comments provided here are useful for the authors to improve this paper and resubmit it to another journal.

Minor comments:

Please provide page number and line numbers as it is good practice that eases reviewers' task.

- ♣ Abstract: "utility-scale floating..." need to indicate that the turbine was not in operation during the measurements.

- ♣ Abstract: "sufficiently high to provide... and downstream velocity deficit". what are the key elements that you have unveiled of such flows?

- ♣ Introduction, first sentence: OE is capable of partly contributing to the decarbonisation but not to absorb all of it. Needs update.

- ♣ Remove "underwater" as it is intrinsic of tidal turbines.

- ♣ "shear lines", what do you mean by shear lines? is this referring to the developed flow?

- ♣ "spread laterally", flow would spread normal to the coastline rather than laterally.

- ♣ "km-scale spatial extent of shear lines". still unclear what this is.

- ♣ "horizontal gradient", do you mean in the transverse direction? horizontal involves x and y.

- ♣ Boils. Boils are only one type of coherent flow structure depending on the bathymetry features.

- ♣ "floating multi-rotor turbines". Floating turbines can also be with a single rotor

- ♣ Introduction, second page. The standard actually recommends that two ADCPs are deployed on the sides of the turbines to capture this spatial variability, which has even been discussed in previous studies looking at tidal turbine loading.

- ♣ "It is still unknown... various spatial scales". It is known that turbulence intensity and lengthscale will drive the wake recovery, they just need to be characterised at the site. This statement needs to be corrected.

- ♣ Introduction, page 3, last paragraph. Need to highlight that the device was not in operation.

- ♣ Figure 1. It would be helpful to include flow directions of ebb and flood tides.

- ♣ Results, page 6. "and vortices". What kind of vortices and how did you determined these?

- ♣ Page 11. "divergence distributions". Can you define this quantity?

- ♣ Page 13. "Notably...". would this be attributed to wind-induced shear or waves? have you

measured the wave conditions at the site?

♣ Page 16. "the wake's intensity and area". This is not clear. What do you mean by wake area and how is this quantified?

♣ Page 16. "of flow speed and downstream distance", and more parameters such as TI or lengthscales now only speed here.

♣ Page 16. "The bubbles then start to disperse laterally...". This read very speculative. It is very hard to characterise the flow based on these "bubbles". Why did you not look into the analysis of turbulence intensities in their vertical components? what about vertical velocity distribution? Bubbles are an idealisation as they would only be introduced by the blades when cavitating and as the device is not operating they are not in the flow.

♣ Page 18. "velocity fluctuations of the U and V components over time". These are not fluctuations but instantaneous variation.

♣ Page 18. "In the shear line area... through the sample area". Again, this is speculative because if this is a shear layer there would be vortices induced due to lateral shear that travel intermittently downstream.

♣ Page 18, "more isotropic compared". More isotropic based on what criterion? To establish this you would need to look into the three components of turbulence intensity and is not the case.

♣ Page 18. "The negative velocity deficit... leading to vertical blockage effects". this are likely to be due to bathymetry effects as an idled turbine will not have a large thrust that can block the flow. However, changes in bathymetry (both roughness and slope) can contribute to this, as shown in your figure 2.

♣ Page 20. "tank experiments.". Do you mean water tanks?

♣ Page 20. "there are no studies to date... shear during the inflow". This is the very similar to studying a device with a given yaw angle, as it encounters a given shear across the rotor.

♣ Page 20. "in the cross-stream 'meandering' is... idealised wake modelling.". This is going to be confusing to the readers. Wake meandering occurs only if the turbine is operating as the rotor rotates, it induces wake meandering downstream. However, here the turbine is not operating, thus there is no wake meandering as such. Flow can be oscillating laterally as it goes through a porous body. This needs to be amended.

♣ Page 21. "Current power curve standards... both tidal states". See previous comment. Figure 5 of the IEC 62600:200 show how to locate two ADCPs.

♣ Page 21. "There is currently a lack of measurements... turbine-turbine interaction from being developed". That is right. More needs to be done here.

Responses to Reviewers

Below is a point-by-point response to all four reviewers. Reviewer comments are highlighted in bold using indentation with responses provided below. We sincerely apologise for erroneously omitting the line numbers during first submission. This has now been amended and we refer to Line numbers in the new ('clean') document version. We also show all changes in the manuscript text file with track changes.

Reviewer #1 (Remarks to the Author):

Review of “Troubled waters: Revealing sheared turbulent flows and wake dynamics of a floating tidal turbine”

This investigation presents new flow measurements around the Orbital O2 tidal turbine located in the Fall of Warness, UK. The paper presents measurements collected with a mobile acoustic current Doppler profiler, echo sounder, and with a drone following a PIV technique. Results show new information about the turbine's inflow conditions and about the wake generated by the floating tidal energy platform and its turbines.

Field measurements around tidal turbines are scarce, mainly because there are only a few operating full-scale tidal turbines in the field and because collecting data in energetic tidal sites is difficult and expensive. I find the data presented in the paper to be relevant to the tidal energy industry and necessary for evaluating the hydrodynamic environmental effects of tidal energy extraction. However, there are aspects of the paper that need to be greatly improved before publication.

To ease this review, I will answer the following questions provided in this journal review guidelines.

What are the noteworthy results?

The data itself is relevant and unique, especially the acoustic wake measurements and PIV velocity estimations from drone images, which is a novel technique barely applied in the field. The wake description, although simple, is noteworthy because most wake velocity deficit estimates come from numerical simulations, and there are not many field data sets about it.

We thank the reviewer for their thoughtful consideration of our manuscript and positive evaluation of our novel data collection approaches. The PIV analyses in this context is indeed relatively novel, and in fact, we have pioneered this technique in our 2021 study for tidal, highly dynamic environments. There has since been one further study by Fairley et al who have assessed aerial drones and PIV techniques for mapping tidal flows which we originally cited in the introduction (L103) and which we now also refer to in the discussion (L 486) and the methods (L714). As part of the revision, we have further developed the inflow and wake analyses incorporating turbulent length scales extracted from the aerial drone imagery of surface flows, please refer to updated Figures 5F and 8B, the added section in

the Methods (L711 -714) and the accompanying figure in the Supplementary Information (Fig. S10).

Finally, we would go further to say that this is the first study that presents wake data obtained downstream of a multi-MW, floating tidal stream turbine platform.

Lieber, L., Langrock, R. & Nimmo-Smith, A. A bird's eye view on turbulence: Seabird foraging associations with evolving surface flow features. *Proceedings of the Royal Society B* (2021).

Fairley, I. et al. Drone-based large-scale particle image velocimetry applied to tidal stream energy resource assessment. *Renew Energy* 196, 839–855 (2022).

Will the work be of significance to the field and related fields? How does it compare to the established literature? If the work is not original, please provide relevant references.

Yes, it will be relevant for the tidal energy field and for the field of fluid dynamics.

The work is original because it is new data in a different site, and around a turbine that to the best of my knowledge very few studies have been published. Prior work regarding spatial flow measurements has been cited. However, the paper does not include in the introduction nor in the discussion comparisons with other sites and with other turbine wakes either from models or from the field data.

There is vast literature about turbulence measurements at tidal energy sites that is not included in the paper, and I think is important to include it to provide context and to better discuss the obtained results.

We thank the reviewer for their positive feedback and appreciate the comments regarding comparison with other sites and turbulence measurements. We have originally cited other field and numerical studies mostly in the introduction, however, we have now provided more direct comparisons and concrete examples in the Discussion, including comparisons of vertical shear profiles from other sites, such as the Alderney Race (English Channel), Ramsey Sound (Wales, UK), the Grand Passage in the Bay of Fundy (Canada) as well as from moored ADCP deployments in the Fall of Warness (L467-469), as well as turbulence intensities (L481) and wake measurements (L532), where applicable. Unfortunately, no other study on the O2 platform has been published to date and the PLAT-I platform (assessed by Guerra et al 2021) is the only other floating platform considered to-date. However, we have now added new measurements on turbulence length scales (see updated Figures 5 and 8) and discuss our results with theoretical predictions of inflow conditions, wake propagation and recovery. We have also added a section in the Introduction to define measures of turbulence (turbulence intensity and length scales; L69-73), thereby expanding our previous text which was more focused on turbulent coherent structures.

Guerra, M., Hay, A. E., Karsten, R., Trowse, G. & Cheel, R. A. Turbulent flow mapping in a high-flow tidal channel using mobile acoustic Doppler current profilers. *Renew Energy* 177, 759–772 (2021).

Does the work support the conclusions and claims, or is additional evidence needed?

The flow description is mostly qualitative and includes snapshots of what is happening in this variable unsteady environment. Perhaps authors could include or summarize the results for different tidal conditions.

We have originally provided figures focussing mainly on the stronger flows during the ebb and flood throughout, but we now provide a comprehensive list of additional figures during other tidal states in the Supplementary Information as well as comparisons in the main text, e.g. in L193-L196, L280-282. The Supplementary Information now includes comprehensive flow visualisations of all broad-scale transects (Fig. S1) as well as detailed inflow transect lines for every fine-scale transect (Fig. S2). We now also provide a Summary Table (Supplementary Table S1) on mean tidal velocities for the duration of each transect (as measured by the O2 ADCPs) and concurrently measured environmental data (e.g. wind and waves) as well as information on O2 platform heading and displacement, drone flight times and in-flight wind conditions. We'd like to note that we have originally summarised the behaviour of the wake as a function of flow velocity in a summary plot (originally Figure 8, now Figure 7) and have now added labels to distinguish the flood and ebb tidal cycle which we trust helps to interpret the variable nature of wake propagation.

Another main concern is that the authors describe the turbine wake and claim to have measured or detected wake meandering. Turbine wake meandering has been previously observed in laboratory experiments and in detailed computer fluid dynamic simulations of flow passing through a rotating turbine. This feature has been linked to the interaction between the rotor shed vortex and the counter-rotating the tip-vortex shed by the rotating turbine blades. In this paper, the authors indicate that the turbines were not operating, meaning the turbine blades were not rotating, so the authors need to justify how they observed the wake meandering feature and what mechanism is producing the meandering.

We have originally used the term “meandering” to refer to the variation in cross-stream (Y-dimension) location as the wake is advected downstream, best visualised in our wake summary plot (now Figure 7). However, we appreciate the reviewer’s concern that this might be confusing in light of the traditional use of the term “meandering”. We have amended the manuscript throughout and now discuss the variation as the mean “cross-stream” location of the wake in the Y-dimension which shows large-scale cross-stream variation (± 50 m in the mean cross-stream location during a single transect of order 30 min duration), please refer to L513. It should also be noted that the O2 platform is not static but is fixed to a dynamic mooring. It is thereby in constant motion (e.g. side-to-side movement) which may influence the behaviour of the wake. We have now added the cross-stream movement (in meters) of the O2 platform in the new Summary Table (Supplementary Table S1). We have also referred to this in the Discussion (L518) and draw attention to new findings from the offshore floating wind literature which suggest that side-to-side motion of a floating offshore wind platform can be a novel origin for the onset of wind wake meandering (L515):

Li, Z., Dong, G., & Yang, X. (2022). Onset of wake meandering for a floating offshore wind turbine under side-to-side motion. *Journal of Fluid Mechanics*, 934. <https://doi.org/10.1017/jfm.2021.1147>

Are there any flaws in the data analysis, interpretation and conclusions? Do these prohibit publication or require revision?

Yes, the publication requires revision.

Another important concern is about the estimation of turbulence metrics from the PIV data. Authors use only 2 minutes of data to obtain turbulence statistics, in previous works it has been established that to get robust turbulence statistics in unsteady tidal flows, at least 5 minutes of data are needed, in order to have statistical stationarity and to capture a wide range of turbulence length-scales. The authors need to justify their choice of time-series length and discuss errors in their calculations.

Yes, this is always a challenge but in reality, there are compromises between overall flight time and being able to capture various sites (upstream and downstream hovers) under the assumption of Talyor's frozen field. We have expanded on this in the Methods in L695-L697. We have now assessed convergence given our hover time (please refer to Supplementary Fig. S9). In short, TI and velocity magnitude measurements show convergence over time in the freestream, however, do not fully converge in the shear line region due to the presence of large intermittent, turbulent coherent structures. This has now been added to the Methods section in L697-L700. Further we have included an assessment of the expected error in TI due to drone stability (L714-717).

Is the methodology sound? Does the work meet the expected standards in your field?

The field measurements techniques are sound and are state of the art. ADCPs are the instrument of choice in oceanography for measuring velocities, and post processing seems to be PIV is a widely used technique in the lab, not common in the field.

Authors should specify in the methods how is the PIV data processed, include sources of error and mention encountered difficulties. Also, need to specify how other quantities such as divergence and vorticity were estimated.

ADCP processing seems to be standard, but authors should specify how the gridded products were obtained.

We thank the reviewer for this comment and have now updated our Methods section to include these details. The gridding is now specified in L634 and the calculation of divergence and vorticity is specified in L706 in the Methods. Further we have added an assessment of the expected error in TI due to drone stability (L714-717).

Is there enough detail provided in the methods for the work to be reproduced?

As mentioned before, more details need to be included in the methods section to reproduce all the data post processing.

We have now updated our Methods section to include these details.

Specific comments:

The draft does not have page numbers nor line numbers, which would be helpful for conducting the review. Here I detailed some questions and suggestions regarding figures, results, and methods.

We sincerely apologise for this and have now included line numbers.

Figure 2:

Are the top panels instantaneous data? It is very hard to see the details in Panels A and B

Yes, these are instantons. We agree and have now amended the figure caption and adjusted the layout for Figure 2 to improve clarity and readability.

Figure 3:

How were these data gridded and interpolated? Is not fully specified in the methods section. Is the data collected along the black lines used here for interpolation? How many transects are included in the interpolation?

We have further clarified the gridding and interpolation process in the Methods sections in L634, specifying that we used 2-dimensional linear interpolation.

Authors need to specify at which depth were these data collected or if it is depth averaged.

We have amended the caption to specify that these data (Fig. 3) were depth-averaged.

Arrows in the left-hand panels are misleading, arrows typically show flow direction rather than a difference with upstream velocity. Perhaps the authors can include a velocity deficit color plot.

This plot is aimed to visualise the complex flows around the O2 platform, including relative variation in flows, the shear line impingement as well as the behaviour of the flows (using streamlines). We have clarified the display by reducing the density and increasing the size of the vectors.

Also, there should not be data over the platform location, since the platform is supposed to be there, hence no data can be collected on top of it.

We acknowledge that there would be no data at this point and we have now amended this (however, it is in the nature of the interpolation that they cover the O2 location).

Figure 4:

**Panel D, figure S: how do you explain the reversed shear near the bottom?
Also in the text: what do you mean by “the shear line intrudes” and by
inclined later shear?**

The Eday shear line is vertically inclined, and we have now indicated this by the diagonal dashed line in Figure 4B. The shear line impinges directly upstream of the rotor-swept area of the O2, resulting in both vertical and cross-stream shear affecting the two rotors on either side of the platform differently, potentially even affecting different blades of a single rotor both vertically and horizontally. We have modified the text to describe this as ‘impingement’ rather than an intrusion, e.g. in L264, L279 and L283, L469 and L473.

Supplemental materials Figures S2 are impossible to read.

We agree and have improved all figures in the Supplementary information.

Figure 5:

Vorticity and divergence: Authors should explain how those quantities were estimated in the methods. Vorticity is a vector, I assume this is vertical vorticity. There is only one snapshot of vorticity in Fig. 5, but there is no description or explanation on the sign of vorticity and if the distribution of vorticity makes sense in the text. Same with divergence, why is divergence relevant for wake dynamics?

We have now clarified how these are calculated and utilised in the Methods section (L706) and have clarified the sign in the caption for Figure 5. Surface divergence is a useful measure of turbulent coherent structures that lead to surface upwelling (“boils” – positive divergence) and associated downwelling. This has also been clarified in the Introduction (L63-L65).

Figure 6: It would be really helpful if these data were presented in space rather than in time, as are the rest of the data presented in the paper.

Please note, this plot has now been moved to the Supplementary Information (Fig. S5). Time was deliberately retained on the x-axis to emphasise the duration of a single sampling transect and the corresponding data with meters on the X axis is displayed in the curtain plot (now Figure 6).

Figure 7: In panel A is very hard to distinguish the reduced wake velocities.

Yes – which is why we then include the acoustic backscatter that acts as a clearer indicator of the location of the wake in this complex and variable natural flow environment and in the absence of the O2 generating. Please note, ADCP-derived velocity deficits are specifically addressed in what is now Figure 9.

Why does the acoustic wake signal deepen if bubbles are buoyant? What could be driving the bubbles to move downward?

Small bubbles have a very low rise-velocity (order mm/s) and so are readily advected vertically downwards when vertical velocities exceed this. Hence, they have been widely used to map the distribution of near-surface physical flow features. We have updated the Introduction (L114-L119) to explain this phenomenon and included some helpful example references, e.g.:

Nimmo-Smith, W. A. M., Thorpe, S. A. & Graham, A. Surface effects of bottom-generated turbulence in a shallow tidal sea. *Nature* 400, 251–254 (1999).

Thorpe, S. A., Green, J. A. M., Simpson, J. H., Osborn, T. R. & Nimmo Smith, W. A. M. Boils and turbulence in a weakly stratified shallow tidal sea. *J Phys Oceanogr* 38, 1711–1730 (2008).

We have also included references on how active acoustics are used in physical oceanography to map physical processes (L114), including turbulence and microstructure, gas seeps, or mixed layer depth.

Figure 9 and 10:

I find this to be one of the most important or more relevant result of the paper.

Thank you. Please note, we have further enhanced these plots with additional analyses on turbulence intensity and length scales within the O2 wake area, the shear line and the freestream in Figure 8B (previously Fig. 9) and shaded errors in Figure 9 (previously Fig. 10).

Regarding, TI, what is the error of TI estimation with PIV?

We have now included as assessment of the error of TI estimation with PIV in our Method section (L714-L717).

Discussion

The discussion section would be greatly improved if comparisons were made with other available datasets and models. Discussion mostly focuses on their own data and how their data is relevant.

For example: How do the wake velocities compared to previously estimated wake velocities obtained from models (if available). How does turbulence intensity compare to other estimates? How does the wake vary with different tidal conditions? What can you expect for a operating turbine wake?

Supplemental Figures S1 and S2 are not readable.

We thank the reviewer for this helpful observation and have now entirely restructured our Discussion to include comparison with other studies (submerged turbines and another floating turbine) and simulations, e.g.:

Blackmore, T., Batten, W. M. J. & Bahaj, A. S. Influence of turbulence on the wake of a marine current turbine simulator. Proceedings of the Royal Society A: Mathematical, Physical and Engineering Sciences 470, (2014).

We now also compare our turbulence intensities and wake deficits with other studies previously only mentioned in the introduction. Wake propagation as a function of tidal state is provided in Figure 7. We have also improved all figures in the Supplementary Information.

Reviewer #2 (Remarks to the Author):

The authors present measurements and discussions about ocean currents at a floating tidal turbine in the Fall of Warness, UK. The measurements were conducted via multiple instruments, e.g., ADCP, PIV, beam combi transducer, and aerial drone surveys, and different data were recorded, such as ocean current speed and turbulence fluctuation.

I. Intellectual merit

1) I would say such work., e.g., ocean measurements, is interesting and beneficial to the community, given that it is lacking and desired for ocean energy development, especially their multiscale features (flows at and away from the turbine). Also, the work may promote understanding and modeling (in its validation) of ocean flow problems other than floating tidal turbines, such as floating offshore wind energy. Additionally, the submission is well-written, although a little too long.

We thank the referee for their support and agree that our rare field observations bridge a gap in transferring idealised numerical and laboratory findings to the real world.

2) Basically, this submission is a case study, and its results are location-dependent. As a result, the understanding presented in this paper is mainly restricted to the location of this study. The work may be further enhanced so its results may bear insight for other sites, if more analysis (as mentioned in the following paragraph) is provided, such as info to show the complete evolution/swinging of the wake over full cycles of tides. Others are comparison of the results to those without turbine/floating device and those at turbine operation conditions, via modeling if not measurements.

Within the scope of the data that were feasible to collect, we have further developed our analysis and related discussion to broaden the relevance of our work beyond the specific site – highlighting the large-scale variability that will be encountered as these technologies are deployed in equally complex natural environments that are not readily simulated numerically or at laboratory scales. As part of the revision, we have further developed the inflow and wake analyses incorporating turbulent length scales extracted from the aerial drone imagery of surface flows, please refer to updated Figures 5F and 8B, the added

section in the Methods (L711 -714) and the accompanying figure in the Supplementary Information (Fig. S10). We have extensively reworked the Discussion to address these points.

3) The work primarily describes the measurement results and corresponding flows, and it lacks insight and analysis on the flows patterns and mechanisms. For instance, there is lack of discussion on correlations between speeds/turbulence/flows at different locations from the turbine, to reflect multiscale features of the flows. Additionally, it would be interesting if results/measurement are provided for flows at the turbine and those further away, e.g., further away in the southeast and northwest directions.

Figure 3 specifically shows the spatial variability in the speeds and relative directions of the flows in the vicinity of the O2, while the mechanisms leading to the observed patterns are described and discussed within the relevant sections. Our sampling was determined by the as-deployed siting of the O2 platform, that following our measurements presented here, might have been better placed elsewhere on the EMEC site had our insights be available before berth locations were set. We now discuss our new analyses on turbulence intensities and length scales in relation to inflow measurements as well as wake propagation and recovery.

II. Presentation problems

4) Fig. 3 --- for the velocity field on the left, the color bars shows that dark and light color mean low and high speed, respectively. But the length of vectors in them seem to be in the opposite way. For instance, the color shows high speed in the lower right corner of the left figure in Fig. 3, but the vector length there indicates low speed there. Do I miss something?

As explained in both the caption and accompanying Results text, the vectors show the distribution of the flows _relative_ to the O2 inflow region. The specific intention is to highlight the local scale flow variations as suggested above (in direct vicinity and away from the turbine). We have now modified the figure by reducing the density of displayed vectors to improve the clarity in presentation.

5) Some results/analysis may be better presented. For instance, the discussion about Fig. 8, i.e., the paragraph right before the figure is unclear. I don't see an explanation of each of the lines in the figure.

We have modified this figure (now Figure 7) to separate out the Ebb and Flood for each of the transects represented by the individual lines and have updated the caption and associated text accordingly. We have also modified the figure to only include downstream wake characteristics.

6) From Fig. 9, visually, it seems that U_d and U have certain correlation at 5D and 7.5D. Moreover, V has such a relation at 2.5D and 7.5D, and it is about zero, exhibiting no correlation at 5D. Are those true, and if yes, why?

We apologise for the poor choice of colour-coding within the original plot that led to confusion between the panels, we have now amended the plot to show the distinction between the Ud (A) and U & V (C) parts. We have also included turbulence intensity and length scales to this figure in (B).

7) This work is valuable for ocean modeling (e.g., its validation). Therefore, it helps if more details are provided with this regard, such as adding time and date of the measurements in figure captions.

We thank the reviewer for this comment and have now provided a detailed Summary table in the Supplementary Information (Table S1) of all measurement transects and associated mean tidal velocities for the duration of each transect (as measured by the O2 ADCPs) and concurrently measured environmental data (e.g. wind and waves) as well as information on O2 platform heading and displacement, drone flight times and in-flight wind conditions. We have also updated all figure captions to include the date and transect number.

8) I cannot judge much on measurement techniques but wonder if the measured data by ADCP mounted on the vessel are affected by the blocking/wake from the vessel itself. An discussion on this helps.

We have amended the Methods section to highlight that the ADCP was mounted below the hull depth and thus be unaffected by the vessel.

Overall, I think this work has impact on understanding of the influence of the turbine on the flows and engineering problems at its location. However, it lacks intellectual merit, particularly science impact/insight on sciences and flow mechanisms. After revision, the submission may fit a domain science journal, such as Ocean Eng., Coastal Eng.

Our work provides a bridge between simulations and reality that is fundamental to allow the wider adoption of this (and similar) technology and analyses approaches to measure flow interactions with floating infrastructure, which includes their impact on the natural environment. The latter is not only important for realised energy capacity but also for environmental and ecological considerations – we have now expanded on this highly topical issue in the Discussion (in L539 onwards, and L549 onwards). We have extensively modified the manuscript to better highlight these points and emphasize our objective on providing a more holistic, oceanographic lens on flow characterisation (L134 in the Introduction).

Our baseline data are highly valuable for a number of reasons as they will: 1) inform the developer in its current location, 2) help EMEC to re-consider berth locations, 3) inform site characterization approaches at the nearby planned tidal array site (Westray Firth; refer to L538) and further afield and 4) increase the understanding of complex, natural tidal stream sites by providing data for modelling data sets that have yet to be validated (e.g. the RealTide project).

Finally, we have substantially advanced conventional and novel instrument-specific analysis approaches, including ADCP-related wake tracing and PIV-derived turbulence intensity and length scales (please refer to comments below on newly added analyses) as well as velocity deficits. Therefore, we believe that this is a highly valuable contribution, with methods that can be extended to other offshore platforms, including floating wind (please refer to changes in the Discussion in L539 onwards).

Reviewer #3 (Remarks to the Author):

Key Results

The Authors present results of an experimental campaign at the Orbital Marine Power tidal turbine platform in Scotland UK. Through acoustic profiling and analyzing aerial drone imagery, they attempt to reconstruct the flow conditions around and in the wake of the turbine. To the best of my knowledge, this is the first experiment of flow characterization around a tidal turbine.

Experimental results show that the presence of areas of high turbulence intensity in the inflow region of the turbine may reduce the current speed at the rotors. This may have implications for the turbine's structural stability and power generation. Such high flow variability is attributed to the particular configuration of the site (presence of islands).

We thank the reviewer for the consideration of our manuscript.

Validity

The Authors do not support experimental findings with data on the power generated at the turbine. Even if the turbine was not functioning during the experiments, power data generated during similar tidal events are not included/mentioned in the paper. It is then impossible to ascertain whether flow variability has an effect on the turbine performance.

We thank the reviewer for this comment and can confirm that the turbine was not functioning during the time of our measurements. However, our study was not aimed to assess power performance of the turbine. Instead, our measures provide a baseline for the non-operational status and characterise the highly variable inflow region as well as the variable wake propagation which was still measurable despite the turbine not operating. Additionally, obtaining power performance data from tidal energy developers is not easy as such data is commercially sensitive— particularly in cases where turbines are prone to functional difficulties for prolonged periods.

Data gathered during aerial surveys is not provided and a few details are included on their processing pipeline. It is difficult to assess whether aerial imagery was of enough quality to be processed. Given the extremely challenging environmental settings during the experiment (I expect wind to be rather severe at the Fall of Warness), the drone vibrations may have significantly impacted images and velocity estimations. Also, velocities estimated from PIV tend to be rather variable. Typically, longer time intervals

help to mitigate this aspect. However, I could not find any reference on the velocity temporal standard deviations in the paper.

We appreciate the reviewer's concerns. To clarify, each flight consisting of multiple hovers up- and downstream of the platform approaches 4 GB in file size and will thus not be deposited online at this stage but the aerial drone data can be made available upon request.

We have now expanded the detail in the Methods section with emphasis on the processing. We have also provided a summary table in the Supplementary information (Table S1) with information extracted from the O2 weather station, including wind speed and direction and wave height. This table also provides in-flight wind readings at altitude which show that wind conditions were benign (averaging $6.53 \pm 2.00 \text{ ms}^{-1}$) on the days the fine-scale transects and associated drone hovers were performed. During the days with higher wind speeds (15th & 16th), we undertook the broad-scale transects only and did not attempt to conduct drone hovers given higher winds. Based on Fairley et al. 2022 who assessed drone stability using a similar sized drone and wind speed range (up to 5.3 ms^{-1}), frame to frame movement was converted to error in ms^{-1} , giving a mean value of 0.07 ms^{-1} with an average geo-referencing error of 2.1 meters (reference provided below). For our flow speeds (order 2 ms^{-1}), this equates to a contamination of turbulence intensity of order 0.035 – this has now been added to the Methods in L714-717. In addition to providing further details on our PIV analyses that reduces the inherent variability in these data (including the use of correlation thresholds, see L705), we have now assessed convergence of statistics given our hover time (please refer to Supplementary Fig. S9). In short, TI and velocity magnitude measurements show convergence over time in the freestream, however, do not fully converge in the shear line region due to the presence of large intermittent, turbulent coherent structures. This has now been added to the Methods section in L697-L700. Further we have added an assessment of the expected error in TI due to drone stability (L714-717). Extending the sampling duration to fully account for these would have then limited our spatial coverage under the assumption of Taylor's frozen field.

Fairley, I. et al. Drone-based large-scale particle image velocimetry applied to tidal stream energy resource assessment. *Renew Energy* 196, 839–855 (2022).

Significance

Since experimental results are very much connected to the specific site where the turbine is installed, I do not expect the paper's conclusions to highly impact the field of ocean energy extraction.

Thank you for your comment. Our work provides a bridge between simulations and reality that is fundamental to allow the wider adoption of this (and similar) technology and analyses approaches to measure flow interactions with floating infrastructure, which includes their impact on the natural environment. The latter is not only important for realised energy capacity but also for environmental and ecological considerations (energy-environment nexus) – we have now expanded on this highly topical issue in the Discussion (in L539 onwards, and L549 onwards). We have extensively modified the manuscript to better highlight these points and emphasize our objective on providing a more holistic, oceanographic lens on flow characterisation (L134 in the Introduction).

Our baseline data are highly valuable for a number of reasons as they will: 1) inform the developer in its current location, 2) help EMEC to re-consider berth locations, 3) inform site characterization approaches at the nearby planned tidal array site (Westray Firth; refer to L538) and further afield and 4) increase the understanding of complex, natural tidal stream sites by providing data for modelling data sets that have yet to be validated (e.g. the RealTide project). Finally, our study provides important baseline data to address changes in our most dynamic coastal marine habitats.

We have substantially advanced conventional and novel instrument-specific analysis approaches, including ADCP-related wake tracing and PIV-derived turbulence intensity and length scales (please refer to comments below on newly added analyses) as well as velocity deficits. Therefore, we believe that this is a highly valuable contribution, with methods that can be extended to other offshore platforms, including floating wind (please refer to changes in the Discussion in L539 onwards).

The paper may suggest that the performance of the O2 turbine is not optimal, but it does not introduce any new findings on power generation from tides. The Authors claim that they develop new survey methodologies, however, I do not find their observational techniques particularly novel. Acoustic current profiling is a well-established technique. PIV was first introduced in the 1990s and aerial PIV from drones has also been adopted in many settings, mostly in riverine but also in coastal areas.

Thank you for your comment. Although ADCPs are a standard oceanographic tool, they have not until this study been applied to map out the fine-scale details of flows in the vicinity of a tidal turbine structure drawing together both the standard velocity information as well as our novel adoption of wake tracing using acoustic backscattering. Yes, PIV is also a standard tool and we and other authors have previously applied it to coastal environments, however, we have demonstrated for the first time, the extraction of turbulence metrics (including turbulent coherent structures, turbulence intensity and length scale), particularly with regard to a utility-scale platform and associated inflow and wake dynamics (including wake velocity fluctuations, wake velocity deficits and wake recovery). We are not aware of any other work that has combined and reported on a combination of these two complementary approaches around a utility-scale turbine.

Data and methodology

Regarding data acquisition, I think the Authors should have better commented on how they ensured that transects were parallel and, therefore, data profiles were consistently parallel. I expect navigation at the experimental site not to be easy and, thus, some corrections to the data could have been made.

We thank the reviewer for this comment as yes, undertaking boat-based transects in tidal flows that approach 8 knots whilst maintaining safe navigation is challenging. No corrections to the data have been applied as it is not possible to do so. After initial scoping of the site (first transect), the boat skippers were provided with waypoints setting out parallel-line

transects aligned perpendicular with the dominant flow direction. They then manually navigated the survey vessel to the best of their abilities given the highly variable flows across the site as well as wind and wave conditions. We always included our boat transects within our data and plots prior to any gridding to demonstrate the minor variability that is inherent with this kind of sampling. We have now included all boat tracks in the Supplementary Information (Fig. S8) to show the consistency in our transect lines (with the exception of our very first scoping transects). The skippers have extensive experience in navigating tidal stream environments, further improved by the suitability of the survey vessel in terms of power (top speed of 27kn) and stability (catamaran). We have now expanded on this in the Methods (L601).

ADCP data were collected from an instrument mounted on a pole submerged 1.35m. Did the Authors apply any correction to get data up to the sea surface?

No, wherever profiles have been used, they have been shown with the separation from the surface (including the blanking distance and mounting depth). Where we show spatial plots, these are representing depth-means, we have included this now in all captions as well.

More details should have been provided regarding aerial drone surveys and processing:

- **Wind conditions during the hovers, wave conditions, and seasonality at the site should be included.**
- **Appearance of the sea surface: the presence of features is fundamental to acceptable PIV data**
- **PIV parameters: the Authors mention Ref. Lieber et al., 2021. However, experimental settings may have been diverse and PIV parameters should have been properly adapted.**

Thank you for this suggestion. We have now included a full summary table of environmental conditions covering the entire survey periods (Supplementary Table S1). This includes mean tidal velocities for the duration of each transect (as measured by the O2 ADCPs) and concurrently measured environmental data (e.g. wind and waves) as well as information on the O2 platform heading and displacement, drone flight times and in-flight wind conditions. We have also extended our Methods to include more details on the PIV processing including correlation thresholds, a key indicator of data quality (L702 onwards). We have also assessed convergence of statistics (mean velocity and turbulence intensity) in L697-L700 with additional plots provided in the Supplementary Fig. S9.

- **Drone stability is not discussed. The Authors used a commercial drone platform, which is not capable of compensating for high wind conditions. In Lieber et al., 2021, the Authors mention that “There were no static reference points within the camera’s field of view so that whole-field contamination from the relative motion of the UAV cannot be removed”. However, this is not the case for the O2 experiment, where the turbine (or part of it) could have been regarded as static during the 2 minutes-long hovers.**

Based on Fairley et al. 2022 who assessed drone stability using a similar sized drone and wind speed range (up to 5.3 ms^{-1}), frame to frame movement was converted to error in ms^{-1} , giving a mean value of 0.07 ms^{-1} with an average geo-referencing error of 2.1 meters. For

our flow speeds (order 2 ms^{-1}), this equates to a contamination of turbulence intensity of order 0.035 – this has now been added to the Methods in L714-717. Please also refer to our Summary Table (Supplementary Table S1) where we now provide in-flight wind readings at altitude which show that wind conditions were benign (averaging $6.53 \pm 2.00 \text{ ms}^{-1}$) on the days the fine-scale transects (and associated drone hovers) were performed.

Further, the O2 platform is not static but is anchored using a dynamic mooring system, which means that it does move over time scales of minutes and seconds. We have now included a metric for the O2 platform movement as part of the above-mentioned Summary Table (Supplementary Table S1) where it can be seen that the variance in the movement is of order several meters - therefore it cannot be treated as a static reference point.

Fairley, I. et al. Drone-based large-scale particle image velocimetry applied to tidal stream energy resource assessment. *Renew Energy* 196, 839–855 (2022).

- **PIV only provides a velocity characterization of the sea surface. I believe the Authors should have better clarified and discussed this.**

Thank you, we refer to these measures as surface measurements throughout and we have now also included a discussion point on this in L484. In short, given our findings on vertical shear variability in the cross-stream, it is rather unlikely that PIV-derived surface currents at this site could be reliably translated to currents at depth as suggested elsewhere. However, aerial imagery is useful in highlighting regions of flow variability that require further measurement campaigns.

- **Standard deviations of the PIV velocity fields are not provided.**

Although the SD of the velocity field is not provided, we provide the spatial distributions of both the velocity magnitude and turbulence intensity; the latter reflecting SD, as it is calculated using the variance divided by the mean. We therefore feel it is unnecessary to provide the SDs separately.

Suggested improvements

I suggest the Authors to take into account information on the power generation performance of the turbine. Such data could help in truly assessing whether the O2 is performing well or is negatively affected by the island wakes. PIV data should be stabilized before processing.

Our study was not aimed to assess power performance of the turbine. Instead, our measures provide a baseline for the non-operational status and characterise the highly variable inflow region as well as the variable wake propagation which was still measurable despite the turbine not operating. Additionally, obtaining power performance data from tidal energy developers is not easy as such data is commercially sensitive– particularly in cases where turbines are prone to functional difficulties for prolonged periods.

Regarding drone stability, for our flow speeds in the order 2 ms^{-1} , a contamination of turbulence intensity of order 0.035 has now been added to the Methods in L714-717.

Clarity and context

Generally, I had difficulties in identifying the mean streamwise and cross-stream direction in Figures. I think such pieces of information should be properly included in all Figures.

Thank you for this feedback. To aid clarity we have added the terms “streamwise” and “cross-stream” to this figure and caption (Figure 1C). Throughout we have orientated our Y-axis with the cross-stream direction.

Figure 3 is of poor quality.

We have now improved the clarity of this figure, also increasing the size of the vectors and reducing the density of the vectors. We have also improved the resolution of the figure.

I found it difficult to identify the O2 in Figure 6.

Please note, this figure has now been moved to the Supplementary Information (Fig. S5) but we have added clarity to the caption to indicate when the O2 was passed (between the upstream and the first downstream line, ‘D1’).

Visualization of the ebb and flood tides is also not easy. For instance, this sentence: “The ebb and flood tide upstream distance to the O2 is 103 m and 108 m, respectively, while the ebb and flood tide downstream distance to the O2 is 86 m and 189 m, respectively.” would be clearer through a Figure sketch.

We agree that our previous explanation was unclear and have therefore added annotations to the figure as suggested to clarify the distances. Please also note that we have added shaded errors to show variability in these measurements.

References

I think the Authors should consider similar experimental studies that make use of aerial drone surveys for non-invasive flow characterization.

We have originally referred to Fariley et al. in the Introduction and have now referred back to this study in the Methods (L714) and Discussion (L484) as well. We have not referenced studies that did not provide quantitative measures directly from the drone imagery. However, in the Introduction we also highlight other surface flow measurement techniques as obtained from radar and satellites, such as:

Lopez, G., Bennis, A., Barbin, Y. & Sentchev, A. Surface currents in the Alderney Race from high-frequency radar measurements and three-dimensional modelling. *Phil.Trans.R. Soc. A* 387, 20190494 (2020).

Marmorino, G. Investigation of Turbulent Tidal Flow in a Coral Reef Channel Using Multi-Look WorldView-2 Satellite Imagery. *Remote Sens (Basel)* 14, (2022).

Fairley, I. et al. Drone-based large-scale particle image velocimetry applied to tidal stream energy resource assessment. *Renew Energy* 196, 839–855 (2022).

Reviewer #4 (Remarks to the Author):

Review of the manuscript: “Troubled waters: revealing sheared turbulent flows and wake dynamics of a floating tidal turbine” by Lieber et al. submitted to Nature Communications.

Overview:

This manuscript presents the results of a resource characterisation at one of the berths at the EMEC site in Scotland where an idled floating tidal stream turbine is deployed, using vessel-mounted ADCP and drone imagery to measure the velocity field. Results capture the three dimensional nature of the tidal flow at this location which is highly influenced by the local bathymetry and nearby headland, which induced flow curvature and large levels of shear. Such findings will be of use for the tidal turbine developer to characterise loading when device is in operation.

We thank the reviewer for the consideration of our manuscript.

General comments:

Whilst the analysis of the time-averaged velocity field from ADCP and drone is good, there are not enough insights into the turbulence characteristics that are developed during the different tidal phases measured. Unfortunately, the turbine is in parked position which does not generate a significant wake throughout the water column as the results evidence. Moreover, the use of free-surface velocity measurements through drone imagery is very recent as is the need for further understanding and/or correlation of such measurements with the velocities through the water column, which the authors did not analyse. In many instances the authors link ADCP signal to bubbles as a signature of turbulence, but this seems quite speculative, e.g. the idled turbine blades would not undergo cavitation and thus they will not generate air bubbles but fluctuations to the velocity field.

We thank the reviewer for their recognition of the novelty of our measurement techniques and we draw attention to our expanded Methods, Results and Discussion sections where we have expanded upon our analysis of turbulence (specifically turbulence intensity and length scales in both the inflow as well as the wake regions of the O2, and adjacent regions, such as the shear line and the freestream). Please refer to updated Figures 5F and 8B, the added section in the Methods (L711 -714) and the accompanying figure in the Supplementary Information (Fig. S10).

We have now also included a discussion point on the transferability of surface currents to depth in L484. In short, given our findings on vertical shear variability in the cross-stream, it is rather unlikely that PIV-derived surface currents at this site could be reliably translated to currents at depth as suggested elsewhere.

Small bubbles generated by surface wave breaking and entrainment of air at the water surface by turbulent features have a very low rise-velocity (order mm/s) and so are readily advected vertically downwards when vertical velocities exceed this. Hence they have been widely used in physical oceanography to map the distribution of near-surface physical flow features.

We have updated the Introduction (L114-L119) to explain this phenomenon and included some helpful example references, e.g.:

Nimmo-Smith, W. A. M., Thorpe, S. A. & Graham, A. Surface effects of bottom-generated turbulence in a shallow tidal sea. *Nature* 400, 251–254 (1999).

Thorpe, S. A., Green, J. A. M., Simpson, J. H., Osborn, T. R. & Nimmo Smith, W. A. M. Boils and turbulence in a weakly stratified shallow tidal sea. *J Phys Oceanogr* 38, 1711–1730 (2008).

We have also included references on how active acoustics are used in physical oceanography to map physical processes (L114), including turbulence and microstructure, gas seeps, or mixed layer depth.

Another key fluid-mechanics aspect that has received little attention is the wave field, acknowledging this would be difficult but aiming for a publication in Nature Communications would require such extensive analysis. Authors claim that during two ebb cycles the results differ notably. Thus, the question is what drove those changes, where waves very different in period, direction or height?

We have now included a full Summary table of the environmental conditions covering our data collection campaign. This comprehensive summary table (Supplementary Table S1) provides information on mean tidal velocities for the duration of each transect (as measured by the O2 ADCPs) and concurrently measured environmental data (e.g. wind and waves) as well as information on O2 platform heading and displacement, drone flight times and in-flight wind conditions. While not considered here, as waves were negligible during the small-scale transects, the inclusion of high waves would likely further influence wake propagation and recovery (L523). We also refer to waves later in the Discussion (L495), stating that platform motion under the combined effects of currents and waves can further amplify rotor load fluctuations potentially leading to damaging effects as found here:

Arcos, F. Z. De, Vogel, C. R. & Willden, R. H. J. A numerical study on the hydrodynamics of a floating tidal rotor under the combined effects of currents and waves. Elsevier Preprint, (2023).

However, there is no clear difference in wind speed, direction and wave height between the two ebb cycle transects and hence we do not have a clear explanation for the observed

variability and it may therefore be associated with flow variability at larger-scales than our present measurements capture.

Regarding wake propagation, we have also expanded our Discussion on the variation in cross-stream location of the wake. The cross-stream movement (in meters) of the O2 platform (Supplementary Table S1) is now referred to in the Discussion (L518) and we also draw attention to new findings from the offshore floating wind literature which suggest that side-to-side motion of a floating offshore wind platform can be a novel origin for the onset of wind wake meandering (L515):

Li, Z., Dong, G., & Yang, X. (2022). Onset of wake meandering for a floating offshore wind turbine under side-to-side motion. *Journal of Fluid Mechanics*, 934. <https://doi.org/10.1017/jfm.2021.1147>

The manuscript has several flaws regarding the description of the flow dynamics:

♣ In figure 5 you present “divergence” field, but you have not defined what that parameter is and how is calculated.

We have now clarified how all turbulence parameters are calculated in our Methods in L706 onwards. Specifically for divergence, we used the standard MATLAB function *divergence* (L707).

♣ Figure 9 B. This plots the time evolution of the velocity series in absolute magnitude whilst authors indicate these are velocity fluctuation, which is an erroneous concept.

We agree that there was a lack of clarity between the figure 9B and its caption that we have now rectified and we have also modified the corresponding Results section, referring to ‘fluctuations in velocity’ in L409-L413. Please also note, we have modified this Figure (now Figure 8) to show additional analyses on turbulence intensity and length scales in the O2 wake and adjacent regions.

♣ Authors refer many times to “Shear line”, but this is not a concept used in fluid mechanics. Perhaps you aimed to refer to the shear layer but this is unclear.

We have reviewed and clarified our use of the term “shear line”, see L55-60 in the Introduction as we acknowledge that this type and scale of oceanographic flow feature may be unfamiliar. Specifically, we have added the following:

“For instance, fast currents past headlands and islands can generate back-eddies (local flow reversals) and leeward wakes bounded by regions of strong horizontal shear (cross-stream gradient in streamwise velocity) hereafter referred to as “shear lines”^{5 6}. Vortical structures, upwelling (surface divergence) and associated downwelling are characteristic of shear lines⁷ and their often kilometre-scale streamwise extent and cross-stream location will change with underlying flow velocities, as well as wind and wave action.

5. Neill, S. P. & Elliott, A. J. Observations and simulations of an unsteady island wake in the Firth of Forth, Scotland. *Ocean Dyn* 54, 324–332 (2004).
6. Wolanski, E., Asaeda, T., Tanaka, A. & Deleersnijder, E. Three-dimensional island wakes in the field, laboratory experiments and numerical models. *Cont Shelf Res* 16, 1437–1452 (1996).
7. Horwitz, R. & Hay, A. E. Turbulence dissipation rates from horizontal velocity profiles at mid-depth in fast tidal flows. *Renewable and Sustainable Energy Reviews* 114, 283–296 (2017).

Further analysis of the PIV field would be required, e.g. authors refer to those shear lines but without quantifying Reynolds stresses, time-averaged vorticity, etc.

Thank you for this comment. We have now extended our analysis of the PIV data to include estimates of turbulence length scale for three different regions during the inflow hover upstream (Supplementary Fig. S3), over the O2 (see updated Figure 5 and Supplementary Fig. S4) as well as downstream of the O2 (Supplementary Fig. S6 and S7). The three different regions that length scales are calculated for include the shear line, the area covered by the O2 and the freestream, using cross-stream averaging (~20 m).

We refer to this in the Results (L299):

“Turbulence intensity (Fig. 5F) increases from 10% in the freestream to more than 30% towards Eday with an associated increase in turbulence length scales (L_u/D) from 0.26 to more than 0.5. The increase in turbulence intensity and length scales in the shear line is due to the prevalence of intermittent, intense turbulent coherent structures.”

We also refer to our new analysis and results in the Discussion (L482).

Finally, authors claim that IEC standards do not consider deploying two measuring devices on the lateral sides of the turbine. Please review the standards because they do present this very clearly in Figure 4 of the IEC 62600-200 2013.

We thank the reviewer for this comment and have now outlined both deployment configurations in our Discussion, the ‘in-line’ preferred configuration as well as the ‘adjacent’ configuration. However, we have now extended our Discussion to argue that neither method would capture the immediate inflow with platform excursion extending 40 m in both the cross-stream and alongstream directions – these measurements are now available in the Summary Table (Supplementary Table S1) and an additional Figure in the Supplementary Information ((Supplementary Fig. S8).

Please see our changes in the Discussion (L497):

“The variability in onset conditions combined with extensive platform excursion (exceeding 40 m in both the cross- and along stream directions (Supplementary Fig. S8) also has implications for power curve testing. Power curve standards following the International

Electrotechnical Commission Technical Standard 62600-200⁶³ currently recommend two independently located ADCPs deployed up- and downstream of the device ('in-line', preferred option) or else 'adjacent' to the device. This approach is designed for mono-rotor, seabed mounted (i.e. static) devices, which is understandable given their prevalence to date. We show that this approach is incapable of capturing the spatially varying flow conditions across a dual rotor floating platform (cross-stream extent of 45 m) within the bounds of the varying device position."

Recommendation:

The reviewer considers that the manuscript does not have a rigorous analysis of the fluid dynamics of the flow at this site, instead it provides a good resource characterisation in terms of mean velocity distribution and time series. Unfortunately, the turbine is not operating, reducing the impact of this study in the community. The recommendation is to reject this paper.

Hopefully the comments provided here are useful for the authors to improve this paper and resubmit it to another journal.

Thank you for your consideration. We have modified the entire manuscript to better express the aim of this contribution, namely that our work provides a bridge between simulations and reality that is fundamental to allow the wider adoption of this (and similar) technology and analysis approaches to measure real-world flow interactions with floating infrastructure, which includes their impact on the physical environment. The latter is not only important for realised energy capacity but also for environmental and ecological considerations (energy-environment nexus) – we have now expanded on this highly topical theme in the Discussion (in L539 onwards, and L549 onwards). We have extensively modified the manuscript to better highlight this and also emphasize our objective on providing a more holistic, oceanographic lens on flow characterisation and interactions in the abstract (L24) and the Introduction (L134), as opposed to providing a power performance assessment.

Our baseline data are highly valuable for a number of reasons as they will: 1) inform the developer in its current location, 2) help EMEC to re-consider berth locations, 3) inform site characterization approaches at the nearby planned tidal array site (Westray Firth; refer to L538) and further afield and 4) increase the understanding of complex, natural tidal stream sites by providing data for modelling data sets that have yet to be validated (e.g. the RealTide project). Finally, there is significant value in understanding the flow dynamics in the absence of power generation, so that they can be separated from those with power generation and inform floating structure design.

We have substantially advanced conventional and novel instrument-specific analysis approaches, including ADCP-related wake tracing and PIV-derived turbulence intensity and length scales (please refer to comments above on newly added analyses) as well as velocity deficits. Our combined measurement approach delivers on surface and water column complexities (rather than mean velocity distributions) which have yet to be considered in simplified numerical simulations.

Therefore, we believe that this is a highly valuable contribution, with methods that can be extended to other offshore platforms, including floating wind (please refer to changes in the Discussion in L539 onwards).

This is the first study to characterises the inflow and wake of a utility scale, multi-rotor floating platform. The method is highly repeatable (regardless of site), which can help to better understand turbine performance that ultimately informs future device design as devices scale up and encounter potentially larger loading challenges as a result of spatially variable flow (refer to L489 onwards).

Minor comments:

Please provide page number and line numbers as it is good practice that eases reviewers' task.

Yes, we have done this now and apologies for the inadvertent omission during the first submission.

♣ Abstract: “utility-scale floating...” need to indicate that the turbine was not in operation during the measurements.

We have now amended this in the modified abstract (L18):

“Here we use aerial drones and acoustic profiling transects to quantify the site- and scale-dependent complexities of actual turbulent flows around an idled, utility-scale floating tidal turbine (20 m rotor diameter, D).”

♣ Abstract: “sufficiently high to provide... and downstream velocity deficit”. what are the key elements that you have unveiled of such flows?

We have now amended this in the modified abstract (L20):

“The combined spatial resolution of our baseline measurements is sufficiently high to quantify sheared, turbulent inflow conditions (reversed shear profiles, turbulence intensity >20%, and turbulence length scales > 0.5D). We also detect downstream velocity deficits (approaching 20% at 4D in its idled configuration) and trace the far-wake propagation downstream using acoustic scattering techniques in excess of 30D.”

♣ Introduction, first sentence: OE is capable of partly contributing to the decarbonisation but not to absorb all of it. Needs update.

This has now been modified (L44).

♣ Remove “underwater” as it is intrinsic of tidal turbines.

We keep this to be inclusive of an intended broad readership.

♣ **“shear lines”, what do you mean by shear lines? is this referring to the developed flow?**

Please see our comment above, we have explained this in L55-L60 of the Introduction.

♣ **“spread laterally”, flow would spread normal to the coastline rather than laterally.**

We have amended our terminology throughout the manuscript and provide explanation in the Introduction, now referring only to it as “cross-stream”. The orientation relative to the local coastline (embayment, headlands) is not directly relevant.

♣ **“km-scale spatial extent of shear lines”. still unclear what this is.**

This has been clarified see in L55-L60.

♣ **“horizontal gradient”, do you mean in the transverse direction? horizontal involves x and y.**

This has been clarified throughout by the appropriate use of “cross-stream and along-stream”.

♣ **Boils. Boils are only one type of coherent flow structure depending on the bathymetry features.**

We have amended the wording at L63.

♣ **“floating multi-rotor turbines”. Floating turbines can also be with a single rotor**

That’s correct and we have amended this.

♣ **Introduction, second page. The standard actually recommends that two ADCPs are deployed on the sides of the turbines to capture this spatial variability, which has even been discussed in previous studies looking at tidal turbine loading.**

Please see our changes in the Discussion (L497):

“The variability in onset conditions combined with extensive platform excursion (exceeding 40 m in both the cross- and along stream directions (Supplementary Fig. S8) also has implications for power curve testing. Power curve standards following the International Electrotechnical Commission Technical Standard 62600-200⁶³ currently recommend two independently located ADCPs deployed up- and downstream of the device (‘in-line’, preferred option) or else ‘adjacent’ to the device. This approach is designed for mono-rotor, seabed mounted (i.e. static) devices, which is understandable given their prevalence to

date. We show that this approach is incapable of capturing the spatially varying flow conditions across a dual rotor floating platform (cross-stream extent of 45 m) within the bounds of the varying device position.”

♣ **“It is still unknown... various spatial scales”. It is known that turbulence intensity and lengthscale will drive the wake recovery, they just need to be characterised at the site. This statement needs to be corrected.**

We have added parameters to discuss this (turbulence intensity and length scales). Please also refer to the amended discussion to address this (L520):

“...enhanced recovery likely associated with turbulent coherent structures and associated increases in TI and L_v/D in the shear line.”

♣ **Introduction, page 3, last paragraph. Need to highlight that the device was not in operation.**

We have included this now in the Introduction (L139) as well as throughout all other sections (Abstract Results, Discussion, Methods).

♣ **Figure 1. It would be helpful to include flow directions of ebb and flood tides.**

We have now modified the figure to include this information.

♣ **Results, page 6. “and vortices”. What kind of vortices and how did you determine these?**

We added clarification on this in Line 180 (‘vortical structures’).

♣ **Page 11. “divergence distributions”. Can you define this quantity?**

We have now called these fields as they describe the spatial distributions of this parameter (L303).

♣ **Page 13. “Notably...”. would this be attributed to wind-induced shear or waves? have you measured the wave conditions at the site?**

We have now included a Summary table of all environmental conditions and have added a comment on this point in our discussion. We have now included a full Summary table of the environmental conditions covering our data collection campaign. This comprehensive summary table (Supplementary Table S1) provides information on mean tidal velocities for the duration of each transect (as measured by the O2 ADCPs) and concurrently measured environmental data (e.g. wind and waves) as well as information on O2 platform heading and displacement, drone flight times and in-flight wind conditions. There is no clear

difference in wind speed, direction and wave height between the two ebb cycle transects and hence we do not have a clear explanation for the observed variability and it may therefore be associated with flow variability at larger-scales than our present measurements capture.

♣ Page 16. “the wake’s intensity and area”. This is not clear. What do you mean by wake area and how is this quantified?

We have expanded on our Methods to provide better clarity on how these were calculated. Detail on this can be found in the methods in L675 onwards and in the Supplementary Fig. S5. Briefly, retaining only surface-connected scattering (bubble entrainment) in the water column, the cross-sectional area (spread in both the vertical and cross-stream, in m^2) was calculated. We’ve changed the word ‘intensity’ throughout to the more appropriate term ‘backscattering strength’.

♣ Page 16. “of flow speed and downstream distance”, and more parameters such as TI or lengthscales now only speed here.

Yes, we recognise that wake recovery is influenced by TI and length scales and have expanded on this in the Discussion (L521). Please note, we have also modified the section mentioned in the comment to:

“Overall, the turbine wake forms immediately downstream of the turbine (see drone image in Fig. 1D) and can be characterised by shallow, intense bubble entrainment at the first downstream line (~100 m downstream or 5D). The bubbles then start to disperse in the cross-stream and vertically (as deep as 20 m) as they are advected downstream.”

♣ Page 16. “The bubbles then start to disperse laterally...”. This read very speculative. It is very hard to characterise the flow based on these "bubbles".

Please see earlier comment on surface-entrained bubbles and their wide-scale use as oceanographic tracers.

Why did you not look into the analysis of turbulence intensities in their vertical components? what about vertical velocity distribution? Bubbles are an idealisation as they would only be introduced by the blades when cavitating and as the device is not operating they are not in the flow.

Due to the nature of our mobile transects, the vertical velocity is highly variable and also subject to vessel heave due to surface waves.

♣ Page 18. “velocity fluctuations of the U and V components over time”. These are not fluctuations but instantaneous variation.

Thank you, we have now amended this. When assessing the variation of the U and V velocity components over time as done in Figure 8, we now refer to “fluctuations in velocity”.

♣ Page 18. “In the shear line area... through the sample area”. Again, this is speculative because if this is a shear layer there would be vortices induced due to lateral shear that travel intermittently downstream.

We have now reviewed and amended this section in L407.

♣ Page 18, “more isotropic compared”. More isotropic based on what criterion? To establish this you would need to look into the three components of turbulence intensity and is not the case.

Amended in L414 to: “of similar amplitude in both horizontal components”.

♣ Page 18. “The negative velocity deficit... leading to vertical blockage effects”. this are likely to be due to bathymetry effects as an idled turbine will not have a large thrust that can block the flow. However, changes in bathymetry (both roughness and slope) can contribute to this, as shown in your figure 2.

We agree with the Reviewer and have now lost this statement all together as we agree that it was too speculative.

♣ Page 20. “tank experiments.”. Do you mean water tanks?

Yes, including flumes, but we have now re-structure the Discussion and have lost this particular statement.

♣ Page 20. “there are no studies to date... shear during the inflow”. This is the very similar to studying a device with a given yaw angle, as it encounters a given shear across the rotor.

Yes, there are similarities and we thank the reviewer for this comment. As noted above, we have now modified the Discussion and this particular phrase is no longer in the manuscript. Please refer to our revised section in L458-476.

♣ Page 20. “in the cross-stream ‘meandering’ is... idealised wake modelling.”. This is going to be confusing to the readers. Wake meandering occurs only if the turbine is operating as the rotor rotates, it induces wake meandering downstream. However, here the turbine is not operating, thus there is no wake meandering as such. Flow can be oscillating laterally as it goes through a porous body. This needs to be amended.

We have originally used the term “meandering” to refer to the variation in cross-stream (Y-dimension) location as the wake is advected downstream, best visualised in our wake summary plot (now Figure 7). However, we appreciate the reviewer’s concern that this might be confusing in light of the traditional use of the term “meandering”. We have amended the manuscript throughout and now discuss the variation as the mean “cross-stream” location of the wake in the Y-dimension which shows large-scale cross-stream

variation (± 50 m in the mean cross-stream location during a single transect of order 30 min duration), please refer to L513.

♣ Page 21. “Current power curve standards... both tidal states”. See previous comment. Figure 5 of the IEC 62600:200 show how to locate two ADCPs.

Please see our changes in the Discussion (L497):

“The variability in onset conditions combined with extensive platform excursion (exceeding 40 m in both the cross- and along stream directions (Supplementary Fig. S8) also has implications for power curve testing. Power curve standards following the International Electrotechnical Commission Technical Standard 62600-20063 currently recommend two independently located ADCPs deployed up- and downstream of the device (‘in-line’, preferred option) or else ‘adjacent’ to the device. This approach is designed for mono-rotor, seabed mounted (i.e. static) devices, which is understandable given their prevalence to date. We show that this approach is incapable of capturing the spatially varying flow conditions across a dual rotor floating platform (cross-stream extent of 45 m) within the bounds of the varying device position.”

♣ Page 21. “There is currently a lack of measurements... turbine-turbine interaction from being developed”. That is right. More needs to be done here.

We concur and we believe at this stage (prior to arrays and the built-out of turbines), there is significant value in understanding the flow dynamics in the absence of power generation, so that they can be separated from those with power generation and inform floating structure design. We thank the reviewer for their helpful comments that again highlight the importance of our work demonstrating the real-world complexities of tidal flow environments.

REVIEWER COMMENTS

Reviewer #2 (Remarks to the Author):

The authors have addressed and clarified a good part of my comments. Here are observations:

Overall, the presentation (figures, context) is not that easy to follow. Each author has his/her own way, but some adjustments may help readers. For instance, Fig. 8 is not self-evident/clear by a quick look. A reader can easily understand it if, say, adding legends to its lines.

I would remind that work has a unique value for modeling multiphysics ocean flows, especially if more details are added in this regard. Multiphysics flows (not merely multiscale) are, for instance, those at near-field (vortices and mixing at turbine) and those in the far field (e.g., ocean currents). Although both are fluid flows, they exhibit different patterns in physics and have to be modeled by different models/equations. On the term "multiphysics", here are some refs:

- a. Tang, H.S., Kraatz, S., Wu, X.G., Cheng, W.L., Qu, K., Polly, J., 2013. Coupling of shallow water and circulation models for prediction of multiphysics coastal flows: method, implementation, and experiment. *Ocean Eng.*
- b. Candy, A.S., 2017. An implicit wetting and drying approach for non-hydrostatic baroclinic flows in high aspect ratio domains, *Advances in Water Resources*

Reviewer #3 (Remarks to the Author):

In this second round of revisions, I have taken into consideration the Authors' responses to the major points raised in the first round:

1. Lack of data on the power generated by the turbine.

I agree with the Authors that such data may be difficult to obtain, however I am still convinced that these pieces of information would have put the contribution in a better context and would have broadened the impact of the paper. While performing PIV around a utility-scale turbine is new, extracting turbulence metrics from PIV is not. Therefore, using the experiment to assess the turbine performance would have been beneficial to the article.

2. Assessment of drone vibrations during surveys

The Authors refer to Fairley et al., 2022 to state that frame-by-frame motion resulted in negligible errors in PIV-derived velocity measurements. However, Fairley et al., 2022 conducted experiments in another location and, most importantly, using two different aerial platforms (an M210 and a Phantom 4 Pro 2.0). In particular, drone stability was assessed for the M210 drone. I think it is conceptually wrong to refer to experiments executed with other platforms. I underline that drone stability during hovers is a key parameter since the horizontal accuracy of these systems can typically vary from ± 0.3 m to ± 1.5 m.

3. Novelty of the paper

As mentioned above, extracting turbulence metrics from PIV measurements is not new. Conducting PIV in challenging ocean settings is new but the experimental component of the paper, or at least its description, presents several methodological flaws. Therefore, I do not believe that the paper aligns to the journal aim to represent important advances of significance in the field.

4. PIV processing

While the Authors have provided a few additional information on PIV processing, they refer to Lieber et al., 2021 for the rest of the details. However, the drone platforms used in the experiments (as well as the cameras) are different (DJI Phantom 3 vs Phantom 3 Advanced). Further, the altitude at which experiments were conducted also changed. I think that a preliminary parametric analysis should have been undertaken before analyzing the videos to identify (and

demonstrate) the optimal PIV settings.

In conclusion, I agree that the manuscript may emphasize the importance of experiments towards ocean flow characterization and that it proposes drone-based PIV surveys as a viable technique in this respect. However, I also believe that the paper methodology should have been more robust to support the use of such an approach in similar settings.

Reviewer #5 (Remarks to the Author):

Dear Editor/Authors,

Firstly, thank you for giving me the opportunity to provide a specialist review for this work.

Unfortunately, although I have expertise in turbulence, coherent structures, tidal energy characterisation, boil characterisation from image processing and ADCP's, I have never used PIV techniques and so am lacking in the expertise you require. Having only the abstract when I accepted, and not the questions themselves; I'm sorry if this means a wasted effort for you. However, I have answered your questions inline below where I am able and have also extensively been through the manuscript and added comments and suggestions.

My overarching view is that this work is very much required for this field, the experimental plan is robust and the results interesting, the inflow characterisation of this study is an important analysis and result, but I do not feel that this study as is, is ready for publication. (also; if published, I feel it would be more appropriate in a journal such as JTECH for the novel use of known experimental techniques, or a tidal energy/site specific journal). If published, then I feel that 'wake dynamics' should be changed to 'idle wake dynamics' throughout.

My reasoning is:

- 1) The wake dynamics that are characterised are that of a non-active turbine. I fail to see how these dynamics, although interesting from a science perspective, are useful to the tidal energy industry. Moreover, the dynamics that are characterised would likely be obfuscated should the turbine be spinning. I do feel that these results have merit, but more as a case study for background hydrography, rather than high impact.
- 2) I do not believe that all the dynamics that are characterised are resultant O2 wake; I believe some coherent structure dynamics is being folded into the study.
 - a. This, in itself, is an extremely interesting result and could actually prove fruitful in its own right.
- 3) I have concerns about the methodology of the acquisition of the ADCP data.

The reasoning for these statements are detailed in length inline in the 'article file' and 'supp info' attached in comments and markup.

I am more than happy to share my work on surface characterization and coherent structures (CS) if requested: nacas@bas.ac.uk.

(Answers to specific editorial questions regarding responses to Reviewer #1's previous comments:)

1.) Do you feel that the revised manuscript's coverage of literature about turbulence measurements at tidal energy sites is sufficient to provide adequate context of the work to readers?

I feel yes, for this journal type and space available.

2.) Is the manuscript's description and documentation of flow at varying tidal states clear and presented in sufficient detail?

With the supp material yes, but tidal state means would be advantageous. Notwithstanding the concerns I have on ADCP averaging time and description of what is wake and CS and separation of such.

3.) Reviewer #1 previously commented that the use of 2 minutes of PIV data is insufficient to obtain robust turbulence statistics and that at least 5 minutes of data are required. The authors have responded by revising their manuscript to indicate that there needed to be compromises in the measured data between overall flight time and being able to capture various sites (upstream and downstream hovers) under the assumption of Taylor's frozen field. Additionally, they indicate that TI and velocity magnitude measurements converge over time in the freestream but do not fully converge in the shear line region due to the presence of large intermittent, turbulent coherent structures. Could you please comment on this revision, and indicate whether this sufficiently addresses the reviewer's concern?

Noted and I feel the response is adequate.

4.) Does the revised manuscript sufficiently describe how the PIV data is processed, and how divergence and vorticity are calculated?

Sorry, PIV is not my expertise.

5.) For the ADCP processing, is the gridding sufficiently described?

Yes, but a bit buried (I'd done the calc's manually before I found the info) and I'm not sure on the processing validity in such flows.

6.) Is the description of the shear line impingement in the revised manuscript clear?

Yes.

7.) Are the reduced wake velocities and velocity deficits in the revised manuscript's figures clearly interpretable?

Yes.

8.) Is the error of turbulence intensity estimation with PIV clearly represented in the revised manuscript?

Sorry, PIV is not my expertise. As a layman to this field I do feel they have given an error in frame to frame referencing, but they have computed this with mean flow field speed – this will not be the speed of the CS or wake dynamics they are measuring but rather the speed that these signatures will be advected... I feel this should be detailed more.

9.) Do you consider the revised manuscript's discussion section's comparison with other studies on submerged turbines and floating turbines to be sufficient?

I feel that the discussion section is thorough and compares well with other work. The wake dynamics section should be highlighted earlier in the MS; for relevance. I'm not sure all the statements are robust though, again – comments inline.

Dr N S Lucas

Reviewer #6 (Remarks to the Author):

I have carefully read the paper “Troubled waters: Revealing sheared turbulent flows and wake dynamics of a floating tidal turbine platform” focusing on the questions below for which I provide my opinion.

Questions related to Reviewer #1's remaining concerns:

1.) Does the revised manuscript sufficiently describe how the PIV data is processed, and how divergence and vorticity are calculated?

The paper does not sufficiently describe how the PIV data is processed. In fact, PIV stands for Particle Image Velocimetry and nowhere in the text the authors provide any information to what type of particles were used by the authors, and what is their effect on the velocity determination, as this technique measures the velocity of particles as a proxy of the velocity of the fluid $u_{\text{particles}} = u_{\text{fluid}}$. The choice of particles is therefore important to assess how good is the data.

The authors reference a previous paper of their co-authorship (Lieber et al., 2021) but even there nothing is mentioned about the type of particles used.

They claim to use in this paper the same methodology and even the same parameters as they used before (Lieber et al. 2021), but no justification is presented for, for example, the use of a correlation coefficient of 60%. This is for me an unusual parameter, as in PIV one of the quality data selection parameters is the Signal to Noise ratio. Typical minimum values of SNR are $\text{SNR} > 1.2$.

The density of particles is crucial for PIV calculations and nowhere is stated if a proper density of particles was used.

PIV software packages usually include multi-step and window distortion algorithms to better adjust the processing to the flow gradients and improve spatial resolution. Nowhere in the text the authors describe what was their software (this contrasts with the amount of information given about the ADCP measurements, lines 613-622). In my opinion, the authors should clarify if they use a commercial software or an in-house developed PIV and eventually how does their solution compares with existing solutions. After, a clarification can be made regarding the authors claim: “with a resolution of 3.7 cm per pixel” (lines 701-702) and the depicted velocity field of Figure 5, for example, where only 18 vectors/ ~ 80 m are depicted corresponding to a resolution 1 vector every 18m.

No reference is made to peak-locking and the authors should clarify if the PIV data had any significant peak-locking.

Another question I have concerning the data processing is the fact that images were acquired at 30Hz (that is $\text{dt} = 0.033$, line 690), but in line 703 it is stated that: “instantaneous velocity fields are acquired at 0.25 s interval” (that is $f = 4$ Hz). This should be clarified, if the authors used a time averaged of about 8 images to have a so called instantaneous value every 0.25 s, or resampled the data set.

In lines 714-719 the authors provide an estimation of errors due to the stability of the drone. The authors essentially use the estimations provided by Fairley et al. (2022). However, the authors should better explain how these values lead to the error of 3.5% in the turbulence intensity.

When the authors estimate that the geolocation error is 2.1 m, does this mean that the location of the vectors in the map (e.g. Figure 5E) is affected by the same error?

Using the Matlab's divergence and curl functions to determine the divergence and vorticity is not a problem, the problem in my opinion is the quality of the input data (velocity field) which I can't assess based on the information provided by the authors.

Fairley, I. et al. Drone-based large-scale particle image velocimetry applied to tidal stream energy resource assessment. *Renew Energy* 196, 839–855 (2022)

Lieber L, Langrock R, Nimmo-Smith WAM. (2021) A bird's-eye view on turbulence: seabird foraging associations with evolving surface flow features. *Proc. R. Soc. B*, <https://doi.org/10.1098/rspb.2021.0592>.

2.) Is the error of turbulence intensity estimation with PIV clearly represented in the revised manuscript?

First, I would recommend the authors to rewrite the equation on line 709 as

$$TI = \sqrt{\overline{(u'^2)}} / U$$

As u' is normally used as turbulent fluctuation. Secondly, I would suggest the authors to define what mean velocity, U , was used and how it was computed, namely if it's a spatial (in the field of view of the PIV measurements) and time averaged velocity, if it's only a time averaged velocity and how it was computed.

In Figure S9 it would be useful to represent all the data series used for the spatial average to see their dispersion and better assess the validity of the convergence of both mean values and turbulence intensity. Ideally, this convergence study should have been made for each interrogation window of the PIV

I would recommend the authors the paper by Hitching and Lewis (1991) that provides a way to determine the number of data points necessary to give statistical convergence in fluid measurements. These authors estimate about 3500 the number of samples needed. Also, the paper by Mycek et al. (2014) provides valuable information about convergence of time series in single turbine measurements.

Here it is important to clarify what was the data acquisition frequency (if 30 Hz if 4 Hz) as stated in the previous comment, because if the acquisition frequency was 30 Hz the authors would have 3900 images (30*130), but if 4 Hz were used the number of samples 520 (4*130) is clearly not enough to ensure a convergence of the mean results). I would suggest representing figure S9 indicating in the upper horizontal axis the number of samples.

Hitching, E., & Lewis, A. W. (1999). Sampling techniques for accurate measurement of Reynolds stresses using laser Doppler anemometry. *Flow Measurement and Instrumentation*, 10, 241–247.

Mycek, P., Gaurier, B., Germain, G., Pinon, G. & Rivoalen, E. (2014). Experimental study of the turbulence intensity effects on maritime current turbines behaviour. Part I. *Renewable Energy*, 66, pp 729-746.

3.) Do you consider the revised manuscript's discussion section's comparison with other studies on submerged turbines and floating turbines to be sufficient?

In the Discussion section, the authors mainly point out the merits of their monitoring approach regarding other previous publications in tidal flows, in that sense the performance of submerged of floating turbines are not the main focus.

Questions related to Reviewer #4's remaining concerns:

1.) Reviewer #4 previously commented that the original manuscript's link of the observed ADCP signal to bubbles as a signal of turbulence seems speculative. The authors have responded by revising their manuscript to further explain the observed phenomenon, indicating that this type of observation has been used widely in physical oceanography to map distributions of near-surface physical flow. Do you feel that this has been sufficiently discussed in the revised manuscript, and that the authors' connection between bubbles observed in the ADCP signal and turbulence is correctly represented?

It is a fact that water bubbles are good acoustic tracers, but ADCP's also react to suspended sediments. The ADCP uses the Doppler effect to measure the velocity of tracers dispersed in the water column. These tracers can be suspended sediment, air bubbles, organic matter. Turbulence measurements require fast sampling techniques. According to the RDI's ADCP manual, the instrument has a sampling frequency of 2Hz (ping frequency) which limits the turbulence measurements to events with $f < 1\text{Hz}$ (Nyquist-Shannon criterion). The information regarding ADCP acquisition frequency, and its limitations should be clear stated.

Greene, A.D., Hendricks, P.J., Gregg, M.C. (2015). Using an ADCP to Estimate Turbulent Kinetic Energy Dissipation Rate in Sheltered Coastal Waters, Journal of Atmospheric and Oceanic Technology Journal of Atmospheric and Oceanic Technology, 32,2 DOI: <https://doi.org/10.1175/JTECH-D-13-00207.1>

2.) Reviewer #4 previously commented that the wave field received little attention in the original manuscript and asked what drove the changes in the results between the two ebb cycles observed. The authors revised their manuscript to provide full environmental conditions during their measurements and indicated that, since there is no clear difference in wind speed, direction and wave height between the two ebb cycle transects, they do not have a clear explanation for the observed variability and that it could be associated with flow variability at larger scales than their measurements capture. Could you please comment on this and indicate whether these concerns are sufficiently addressed in the revised manuscript?

The authors included a table with a complete description of the field conditions which is positive. However, in their response they point out to line 523 where nothing is mentioned about wave effects.

The authors' hypothesis regarding flow variability at larger scales is plausible and it should be clearly stated in the manuscript.

3.) Could you please comment on the technical accuracy of the revised manuscript's consideration of wake propagation in the results and discussion?

The fact that the turbine is idle limits the true assessment of the turbine wake. However, it provides a sort of “base case” that may be useful to establish the impact of such structures.

The authors rightfully mention regarding the importance of assessing the wake dynamics the need to determine the energy dissipation (line 91). However, the authors do not develop this issue.

Thiébaud M., Filipot J.-F., Maisondieu Ch., Damblans G., Duarte R., Droniou E. & Guillou S., (2020) Assessing the turbulent kinetic energy budget in an energetic tidal flow from measurements of coupled ADCPs *Phil. Trans. R. Soc. A*.3782019049620190496

The wake analysis provided by the authors rely on the velocity deficit and on the estimation of the wake area both dependent on a threshold criterion (0.75 m/s). How universal is this criterion?

Given that the authors have ADCP transects, wouldn't it be possible to characterize the wake at each transect section as a cut into the wake cone, and therefore allowing for a 3D representation of the turbine effect downstream of it?

4.) Reviewer #4 previously commented that the original manuscript's consideration of "fluctuations in velocity" in the observed velocity series was erroneous; the authors responded by revising their results to clarify this point. Do you feel that the revised manuscript has sufficiently addressed this point?

Fluctuations are defined as the difference between the instantaneous velocity and the average velocity. The authors emphasis on fluctuations can be found in lines 409-414 with reference to Figure 8C. Their and it is essentially correct, however the manuscript could be improved if: i) the turbulence intensity of each time series was referred on the lines 409-414 and that in Figure 8 C would represent the fluctuations (also, or instead, of the instantaneous values).

5.) Is the use of the term "shear lines" clearly described and discussed in the revised manuscript?

No. The term “shear line” is not well defined in the text not even supported by the references used by the authors (Horwitz and Hay, 2017, Neil and Elliot, 2004, Wolanski et al. 1996). Nowhere in those texts the expression shear line is used. I agree with Reviewer 4 and perhaps the concept of “shear layer” fits better here.

Horwitz, R. & Hay, A. E. (2017) Turbulence dissipation rates from horizontal velocity profiles at mid depth in fast tidal flows. *Renewable and Sustainable Energy Reviews* 114, 283–296.

Neill, S. P. & Elliott, A. J. (2004) Observations and simulations of an unsteady island wake in the Firth of Forth, Scotland. *Ocean Dyn* 54, 324–332.

Wolanski, E., Asaeda, T., Tanaka, A. & Deleersnijder, E. Three-dimensional island wakes in the field, laboratory experiments and numerical models. *Cont Shelf Res* 16, 1437–1452 (1996).

6.) Could you please comment on the comprehensiveness of the PIV field analysis presented in the revised manuscript? Do you feel that there is further analysis or discussion required beyond what is provided in the revised manuscript?

The PIV topic must be improved, and the questions raised in my previous comments should be addressed.

7.) Do you agree with the revised manuscript's assertion that IEC 62600-200's recommendations for 'in-line' or 'adjacent' ADCP placement is incapable of capturing the spatially varying flow conditions across a dual rotor floating platform such as the platform considered in this manuscript?

At the moment I have no data to validate or invalidate this sentence.

8.) Do you feel that the flow and vorticity characteristics considered in this manuscript have been sufficiently and accurately represented/discussed?

No. They use the PIV data regarding which I have raised some issues. Furthermore, the spatial of the PIV grid, which is not defined, prevents me to comment further on this issue. The vorticity is the curl of the velocity field. It is a vector whose component perpendicular to the plane xy is given by

$$(dv/dx-du/dy)$$

It is important to have a good spatial resolution to calculate the derivatives, and from the paper it is not possible to assess what was the spatial resolution of the PIV.

Furthermore, I am not convinced about the Taylor macroscale representation, namely on Figures 5F, S3F, S4F, S6F, S7F. Each interrogation window of PIV will have associated a time series, from which it is possible to determine a length scale using the same method as described by the authors, and thus have a length scale map (similar to Figure 5 D), which then can be spatial averaged and displayed as a line like the turbulence intensity in Figure 5F.

The authors use the Taylor frozen-turbulence hypothesis to convert time into space, but since they used PIV they have spatial distributed data, so they could do the correlation directly by using the definition.

Some other minor remarks

Figure 1 c): Indicates the mean location of the O2 turbine. I would suggest using an error-bar like representation that showed the maximum displacements of the O2 turbine. Or if not possible indicate in the legend what were the extremes of the measured O2 location.

Line 188: I would suggest where $D=20m$ is the rotor diameter.

Figure 3: I would suggest placing the red vector used for scalar always in the same relative position.

It would be helpful to the reader to have an indication on the Figure of the geographic referencing points, namely Eday (y positive direction).

It is difficult to understand the role of the streamlines.

Line 222: correct "re $1m^{-1}$ " to "re $1m^{-1}$ "

Line 223: To clarify I guess when referring to $(X/Y = 0)$ the authors mean $(X, Y) = (0,0)$?

Lines 223, 634: The authors should clarify what they mean by “gridded at 1D” It is my assumption that the authors built a grid with 1D spacing (with $D=20\text{m}$ the rotor diameter)

Figure 6A: The PIV hover sections looks like they are located at the bottom instead of the surface.

Line 363: Why were 0.75 m/s chosen as velocity threshold?

Line 369: Why were 1.5 m/s chosen as velocity threshold?

Lacks information about PIV method. See above.

Reference 39 is not a proper PIV reference, I would suggest the following:

Raffel M, Willert CE, Wereley ST, Kompenhans J. (2007) Particle image velocimetry: a practical guide. New York: NY: Springer.

Adrian, R.J., Westerweel, J. (2010) Particle Image Velocimetry. Cambridge University Press.

The reference 17. Is incomplete.

Responses to Reviewers

Below is a point-by-point response to all four reviewers. Reviewer comments are highlighted in bold using indentation with responses provided below. We refer to Line numbers in the new ('clean') manuscript version. We also show all changes in an additional manuscript text file with track changes.

Reviewer #2 (Remarks to the Author):

The authors have addressed and clarified a good part of my comments. Here are observations:

Overall, the presentation (figures, context) is not that easy to follow. Each author has his/her own way, but some adjustments may help readers. For instance, Fig. 8 is not self-evident/clear by a quick look. A reader can easily understand it if, say, adding legends to its lines.

I would remind that work has a unique value for modeling multiphysics ocean flows, especially if more details are added in this regard. Multiphysics flows (not merely multiscale) are, for instance, those at near-field (vortices and mixing at turbine) and those in the far field (e.g., ocean currents). Although both are fluid flows, they exhibit different patterns in physics and have to be modeled by different models/equations. On the term "multiphysics", here are some refs:

a. Tang, H.S., Kraatz, S., Wu, X.G., Cheng, W.L., Qu, K., Polly, J., 2013. Coupling of shallow water and circulation models for prediction of multiphysics coastal flows: method, implementation, and experiment. Ocean Eng.

b. Candy, A.S., 2017. An implicit wetting and drying approach for non-hydrostatic baroclinic flows in high aspect ratio domains, Advances in Water Resources

We thank the reviewer for their thoughtful consideration of our manuscript and positive evaluation. We have now revised Figure 8 for easier interpretation which also includes the mapping of u' and v' (velocity fluctuations) rather than U and V . We have also made some changes to other figures throughout to improve clarity.

We are delighted to hear that our data has relevance for addressing multiphysics using geophysical fluid dynamics models and thank the reviewer for the above references. We will make our data available for exactly these purposes and look forward to seeing how our *in situ* data can be used to calibrate or validate fluid dynamic models in this region. For instance, we've added the overall excursion of the O2 in Figure 1F, improved labelling in Figure 3, and emphasized the PIV hovers in Figure 6 – one of the key figures for this manuscript. Please also note the revised version of Figure 5 demonstrating a more appropriate method for estimating turbulence length scales, detailed in the Supplementary Information under Methods Section S2, page 27.

Reviewer #3 (Remarks to the Author):

In this second round of revisions, I have taken into consideration the Authors' responses to the major points raised in the first round:

1. Lack of data on the power generated by the turbine.

I agree with the Authors that such data may be difficult to obtain, however I am still convinced that these pieces of information would have put the contribution in a better context and would have broadened the impact of the paper. While performing PIV around a utility-scale turbine is new, extracting turbulence metrics from PIV is not. Therefore, using the experiment to assess the turbine performance would have been beneficial to the article.

We acknowledge the desirability of such data, and our study originally aimed to collect data during the operational and the idled state. Unfortunately, due to technical challenges beyond our control, the turbine was unable to generate power during that period. However, to date, floating tidal turbines are still challenged by the complex flow interactions in energetic tidal streams. Our study provides the first assessments of a utility-scale floating tidal turbine, with its immediate inflow region, wake and interactions with local flow structures characterised. To date this has never been done using large-scale PIV techniques around real-world devices. The inflow assessments will still allow valuable information for future power performance evaluation regardless of the turbine operating during this study. Further, it has become evident that assessing flow interactions with an idled structure still provides 1) valuable insight into the trajectory of the resulting wake which is still highly informative for additional turbine placement in the area or in similarly complex environments and 2), provides engineering insight on thrust differences between the rotors as well as loading during downtime and maintenance (associated with the life of the devices and economies of volume).

Secondly, this study will inform environmental assessments (hydrodynamic interactions and potential impacts on the natural environment) for consenting floating platforms. With space for marine infrastructure becoming increasingly limited in coastal seas, there is now an increased focus on the ecologically sustainable development of marine energy structures, as evidenced by initiatives such as NERC's ECOFLOW funding call with focus on floating offshore wind and related infrastructure-induced mixing of seasonally stratified shelf seas. Similarly for tidal streams, there are concerns about the physical footprint of tidal energy devices, especially in light of the build-out of turbines at scale, such as planned arrays north of our study site, off North Wales (Anglesey) or Canada (Bay of Fundy). This aspect is especially relevant for floating turbines given the substantial alteration to the natural environment through their near-surface location and associated wake. Their co-existence with other users of the sea (fisheries, and also including marine fauna in light of the biodiversity on top of the energy crises) will thus play a pivotal role in future site management, also informing cumulative impacts (see references below). There is a growing emphasis on evaluating these effects during all operational states which will also include substantial periods where the device is idled (downtime/maintenance etc.). Demonstrating how to best collect baseline data is also crucial to design future environmental impact assessments.

In summary, beyond providing engineering insight and siting of devices, our work also informs the sustainable development and roll-out of ocean energy. As the development and placement of coastal and offshore infrastructure proliferates in an increasingly crowded (squeezed) seascape, infrastructure-induced mixing and other secondary effects will play a pivotal role during the consenting processes to balance energy needs with nature recovery amidst the biodiversity crisis. Our study remains forward-thinking and relevant in addressing and communicating these considerations.

Hutchinson, Z. L., Lieber, L., Miller, R. G., & Williamson, B. J. (2022). Environmental impacts of tidal and wave energy converters. In T. M. Letcher (Ed.), *Comprehensive renewable energy* (2nd ed., Vol. 8, pp. 258-290). Elsevier. <https://doi.org/10.1016/B978-0-12-819727-1.00115-1>

Dorrell, R. M. et al. Anthropogenic Mixing in Seasonally Stratified Shelf Seas by Offshore Wind Farm Infrastructure. *Frontiers in Marine Science* vol. 9 Preprint at <https://doi.org/10.3389/fmars.2022.830927> (2022)

Christiansen, N., Carpenter, J. R., Daewel, U., Suzuki, N., & Schrum, C. (2023). The large-scale impact of anthropogenic mixing by offshore wind turbine foundations in the shallow North Sea. *Frontiers in Marine Science*, 10. <https://doi.org/10.3389/fmars.2023.1178330>

2. Assessment of drone vibrations during surveys

The Authors refer to Fairley et al., 2022 to state that frame-by-frame motion resulted in negligible errors in PIV-derived velocity measurements. However, Fairley et al., 2022 conducted experiments in another location and, most importantly, using two different aerial platforms (an M210 and a Phantom 4 Pro 2.0). In particular, drone stability was assessed for the M210 drone. I think it is conceptually wrong to refer to experiments executed with other platforms. I underline that drone stability during hovers is a key parameter since the horizontal accuracy of these systems can typically vary from ± 0.3 m to ± 1.5 m.

We thank the reviewer for their comment on drone stability and we agree that it was too simplified to refer to another study to fully address this. We have now flown the study-specific drone over a static surface (beach scene) in strong and gusty winds beyond the conditions encountered during the study to assess the drone's stability. We have added a new section to our Supplementary Information (Methods Section S3, page 28) detailing the outcome of this and amended our manuscript at Line 745-747 accordingly. We have also noted that in the Fairley et al 2022 paper, the authors only report on the mean wind speed, without including the gusts. However, following our assessment of detailed in-flight winds, we can now show that the gusts had the greatest impact on drone stability.

3. Novelty of the paper

As mentioned above, extracting turbulence metrics from PIV measurements is not new. Conducting PIV in challenging ocean settings is new but the experimental component of the paper, or at least its description, presents several methodological flaws. Therefore, I do not believe that the paper aligns to the journal aim to represent important advances of significance in the field.

We would like to clarify that, traditionally, characteristics of turbulent flow have primarily been assessed using laboratory Particle Image Velocimetry (PIV) applications in controlled environments. However, in recent years, efforts have been made to extend the capabilities of lab-based PIV to field investigations of turbulent rivers, most commonly referred to as 'large-scale PIV (LSPIV)' approaches. LSPIV has proven to be a valuable tool for measuring surface velocity in various fluvial systems, although its application in ocean settings remains rare. To facilitate cross-comparisons with previous studies, including Fairley et al. (2022) who also use the term 'LSPIV', we now refer to our PIV approach as 'LSPIV' in the Results in Line 281 and in the Methods section in Line 733 but use the term 'PIV' interchangeably throughout the manuscript.

Please also refer to our new, detailed section in the Supplementary Information (Methods Section S1, page 26).

In summary, the application of LSPIV in turbulent ocean settings, such as tidal streams, is exceedingly rare and has primarily been trialled for cross-comparison studies, as demonstrated by Fairly et al. (2022). To our knowledge, the use of drone-based LSPIV has not been conducted in the vicinity of a real-world tidal turbine to characterise inflow and wake dynamics, marking our study as a significant advancement in the field.

We'd also like to refer to a PIV paper (Hong et al., 2014) which prompted our submission to Nature Communications. The authors report on large-scale PIV measurements in the atmospheric boundary layer behind a fully instrumented wind turbine using natural snowflakes as tracer particles during a snowstorm. This example underscores that the novelty and advancements in a field are often shaped not only by the technique itself but rather by the context in which it is applied.

Hong, J., Toloui, M., Chamorro, L. P., Guala, M., Howard, K., Riley, S., Tucker, J., & Sotiropoulos, F. (2014). Natural snowfall reveals large-scale flow structures in the wake of a 2.5-MW wind turbine. *Nature Communications*, 5(May). <https://doi.org/10.1038/ncomms5216>

Additionally, in response to the reviewer's prompt, we have assessed drone stability now detailed in the Supplementary Information (Methods Section S3, page 28) and associated measurement errors under high wind conditions, rather than relying on a third-party experiment. We have further revised our turbulence length scales (Supplementary Information Methods Section S2, page 27) using the methodology suggested by Reviewer 6 during this round of revisions and we'd like to refer the reviewer's attention to our Methods in Line 740-743 and associated revised plots in Figure 5 and in the Supplementary Information. Our methodologies are thus not flawed, and we'd like to sincerely express our gratitude to the reviewer for prompting this additional investigation.

4. PIV processing

While the Authors have provided a few additional information on PIV processing, they refer to Lieber et al., 2021 for the rest of the details. However, the drone platforms used in the experiments(as well as the cameras) are different (DJI Phantom 3 vs Phantom 3 Advanced). Further, the altitude at which experiments were conducted also changed. I think that a preliminary parametric analysis

should have been undertaken before analyzing the videos to identify (and demonstrate) the optimal PIV settings.

In conclusion, I agree that the manuscript may emphasize the importance of experiments towards ocean flow characterization and that it proposes drone-based PIV surveys as a viable technique in this respect. However, I also believe that the paper methodology should have been more robust to support the use of such an approach in similar settings.

We appreciate the reviewer's comment and have now added additional information on the large-scale PIV processing in the Supplementary Information Methods Section S1, page 26.

The drone platform was indeed the same and we have clarified this in the text, however, the altitude was set to 100 m in the previous Lieber et al. 2021 study to avoid any potential disturbance to foraging birds under investigation.

Throughout our PIV analysis we have always taken a simple and conservative approach and so the only tuneable parameters are the window size and the correlation threshold. For the former, we opted to use a single size rather than iteratively reprocess using decreasing window sizes which is only sensible when in controlled laboratory settings. For the latter, we adopted the same correlation threshold as we used in a more sheltered environment where sometimes regions of completely smooth water would not return a reliable velocity measurement. However, in this current study, all our correlation coefficients were higher than this threshold due to the increased surface roughness and associated natural tracers.

Reviewer #5 (Remarks to the Author):

Dear Editor/Authors,

Firstly, thank you for giving me the opportunity to provide a specialist review for this work.

Unfortunately, although I have expertise in turbulence, coherent structures, tidal energy characterisation, boil characterisation from image processing and ADCP's, I have never used PIV techniques and so am lacking in the expertise you require. Having only the abstract when I accepted, and not the questions themselves; I'm sorry if this means a wasted effort for you. However, I have answered your questions inline below where I am able and have also extensively been through the manuscript and added comments and suggestions.

We thank the reviewer for their thoughts and comments - We have diligently endeavoured, to the best of our abilities, to address all the feedback provided across the various files.

My overarching view is that this work is very much required for this field, the experimental plan is robust and the results interesting, the inflow characterisation of this study is an important analysis and result, but I do not feel that this study as is, is ready for publication. (also; if published, I feel it would be more appropriate in a journal such as JTECH for the novel use of known experimental techniques,

or a tidal energy/site specific journal). If published, then I feel that ‘wake dynamics’ should be changed to ‘idle wake dynamics’ throughout.

We sincerely appreciate the reviewer's support, particularly regarding the study's robustness. In response to the specific feedback on the ‘idled status’, we have not only incorporated it directly into the abstract but have now also adjusted the manuscript's title to reflect that the turbine was in an idled state. The revised title now reads: "Troubled Waters: Revealing Sheared Turbulent Flows and Wake Dynamics of an Idled Floating Tidal Turbine." While we respect the reviewer's perspective, it is essential to recognize that even in the absence of power generation (non-operational state), the platform’s wake signature remains substantial. Consequently, the insights provided in this study are crucial for informing the placement of additional turbines, subsequent array planning, and environmental consenting processes, particularly in spatially constrained tidal streams with ecologically sensitive marine fauna. Moreover, the larger-scale characterisation of the flows provides new insight into the complex and yet poorly resolved flows across the Fall of Warness, particularly in relation to existing EMEC berth locations, a lesson for other geographically constrained tidal stream sites. Finally, it provides previously unknown fine-scale insight regarding the O2’s lateral positioning of their individual rotors informing essential engineering parameters such as thrust and loading.

My reasoning is:

1) The wake dynamics that are characterised are that of a non-active turbine. I fail to see how these dynamics, although interesting from a science perspective, are useful to the tidal energy industry. Moreover, the dynamics that are characterised would likely be obfuscated should the turbine be spinning. I do feel that these results have merit, but more as a case study for background hydrography, rather than high impact.

We greatly appreciate the reviewer's perspective. It's worth noting that the physical alterations to the natural environment and methods for assessing flow dynamics (inflow/wake) remain pertinent regardless of turbine operation. While the concern raised may be more applicable to submerged tidal turbines, it's crucial to recognize that floating structures, situated in regions of strongest flows, still exhibit substantial wake signatures even during idled conditions. Through extensive internal discussions involving the tidal energy developer and site operator, the wake analysis and fine-scale flow characterisation presented in this study offer invaluable insights to the industry. Particularly noteworthy is the utility of our low-cost approach for informing turbine placement decisions and addressing persistent technical challenges.

Further, turbine developers need to understand the flow characterisation around the turbine, and the resulting floating structure stability during periods that the turbine is not generating power when it is waiting for maintenance intervention, as this will impact the loading and therefore life of the device.

Developers may also need to understand the stability of the platform to design devices that use the same floating structure but different rotors/power trains for different sites. For

example, shallower sites may use 9 m rotors instead of 10 m rotors, with this serial platform design being key to achieving economies of volume to bring down cost of energy.

As new arrays are planned for deployment in similar locations, our innovative combination of mobile ADCP surveys and drone-based Particle Image Velocimetry (PIV) represents a cost-effective and pioneering method for industry-wide flow characterization. Furthermore, it's anticipated that with turbines in operation, the wake effects observed herein are not likely obfuscated, but rather magnified in all dimensions.

**2) I do not believe that all the dynamics that are characterised are resultant O2 wake; I believe some coherent structure dynamics is being folded into the study.
a. This, in itself, is an extremely interesting result and could actually prove fruitful in its own right.**

We appreciate the reviewer's observation. Throughout the manuscript, we have provided comprehensive details on coherent structures. It's important to note, as the reviewer rightly points out, that coherent structures are not only found in the downstream wake region of the O2 but also in the freestream, such as characterised during the inflow. Additionally, they are observed around the island wakes, spatially independent of the O2 wake influence.

**3) I have concerns about the methodology of the acquisition of the ADCP data. The reasoning for these statements are detailed in length inline in the 'article file' and 'supp info' attached in comments and markup.
I am more than happy to share my work on surface characterization and coherent structures (CS) if requested: nacas@bas.ac.uk.**

We have extracted the specific reviewer comments found in the main article file, the Suppl Info file and the previous review document and respond to each in line below which should remove any concerns.

L 101-103: also boil characterisation from fixed cameras, oblique to plan measurements, quantify size and propagation. Happy to provide details if needed.

We thank the reviewer for their suggestions. The use of fixed cameras and subsequent rectification (oblique to plan measurements) is not as novel as the other examples used, but we have now added this methodology to address this comment, please refer to Line 101 of the Introduction regarding image orthorectification.

L 170: you have covered a great extent for this survey, not easy, but it is not the full tidal cycle, particularly the broad scale transects, so this wording is misleading

We have changed this to 'throughout our tidal measurements' in Line 167.

L 190-191: Unfortunately, especially considered with the supplementary information, I don't feel you can say this is simply wake. The same is apparent on the alternate side of the channel (circled in B), and in similar extents on flow reversal (circled in supp info). I believe this phenomena is due to the wake, but obfuscated with coherent structures bringing with them sediments from the sea bed. See:

Salim, S. et al. (2017) ‘The influence of turbulent bursting on sediment resuspension under unidirectional currents’, *Earth Surface Dynamics*, 5(3), pp. 399–415. doi: 10.5194/esurf-5-399-2017

or redistribution:

Nimmo Smith, W. A. M., Thorpe, S. A. and Graham, A. (1999) ‘Surface effects of bottom-generated turbulence in a shallow tidal sea’, *Nature*, 400(6741), pp. 251–254. doi: 10.1038/22295.

This is my view - but I don't feel the results herein are robust enough to distill the phenomena down to simply wake dynamics.

Here (Line 187) we simply state that there is an indication for a wake signature, but that the wake dynamics are better captured in the fine-scale surveys. We appreciate the reviewer's perspective, but the signature circled in B by the reviewer is in fact the shear line from Muckle Green Holm, which is dominated by upwelling and associated downwelling, thus coherent structures, with surface-connected bubbles and bubble entrainment visible. Thus, the dominant features here are large bubble plumes corresponding to crossing the shear line. Additional scattering from e.g. suspended sediment is observed elsewhere in the water column (near the bed), but the predominant backscatter we refer to is surface-connected and does not originate from the seabed. The fate of a bubble entrained at the shear line is dependent on the bubble's size and strength of downwelling, with the high intensity scattering returns (-40 dB) being characteristic for bubbles. As briefly explained in the introduction, bubbles are injected into the ocean by breaking waves (including small capillary waves at the edges of coherent structures) and in the presence of downwelling, these bubbles can be subducted to depths greater than those associated with the breaking wave event alone – hence we see relatively deep bubble intrusion when crossing the shear line. The transport of sediment near the seabed by resuspension and advection occurs when shear stresses on the seabed rise to sufficiently high levels (overcoming their negative buoyancy) and this is also observed throughout most transects (with stronger backscattering returns at the source, i.e. near the bed rather than towards the surface), but is very much a second-order effect and not the focus of this study.

We have now added a comprehensive figure to the Supplementary Information (Fig. S10) where we outline some of the dominant scattering sources (bubbles and sediment) and also explain the impacts of excess attenuation and multiple scattering visible below the O2 wake and its bubble plume which appears to have a “tail” of echoes extending below the actual area of the wake, thus distorting the apparent shape and intensity of the wake. Multiple scattering can occur when the density of scatterers (bubbles) is high, and signals scattered from one bubble are re-scattered from neighbouring bubbles (bouncing of echoes). In this case, the apparent wake seems to occupy the full water column, however de-noising and thresholding of the data to intensities consistent with adjacent portions of the wake was sufficient to isolate and trace the wake as per Figure S5 and Figure 6 & 7 of the main manuscript.

To better interpret such phenomena, we suggest the reviewer to refer to the references used in the introduction or refer to the below reference where a shear line is also visualised using ADCP backscatter:

Johnston, D. W., & Read, A. J. (2007). Flow-field observations of a tidally driven island wake used by marine mammals in the Bay of Fundy, Canada. *Fisheries Oceanography*, 16(5), 422–435.
<https://doi.org/10.1111/j.1365-2419.2007.00444.x>

Please also note that the above paper presents ADCP-derived flows and backscatter data collected across a shear line in Figure 4 of the paper, highly similar to our backscatter data across the Muckle Green Holm shear line (note, to the left, there is also sediment re-suspension near the bed for comparison).

We appreciate the reviewer's perspective; however we are the author of one of the papers highlighted by the reviewer above and our combined experience in working in high-flow tidal environments, and as observed by many other authors, highlight that suspended sediment will have a stronger backscattering strength nears its source i.e. the seabed. As noted above, there are some signatures of suspended sediments near the bed, although for the most part in this location, the bed comprises rock/boulders/gravel and so does not have the significant sources of estuarine muds encountered in some other tidal channels, such as e.g. the Menai Strait.

L 203: why only a proxy for surface connected turbulence. I see the link with bubbles you've made, but backscatter will also pick up sediments, marine life etc. As per comment above - these could be brought from the seabed rather than the surface. It's not conclusive - so best to keep it ore general.

Please refer to our comments above for more detail. The dominant source of the scattering is surface-connected, while we agree, to a much lesser extent, there is also sediment suspension visible. However, for the fine-scale transects, we have also extracted the EK80 echosounder-derived backscatter following thresholding and here (Fig. S5) you see that we have isolated the surface-connected scattering (bubbles), and this is also the case for Figure 6 in the main article and these signatures are the focus of our study.

L 210: why not rectify this image and quantify directly these sizes, measurements etc?

We deliberately included this oblique image to present a recognisable overview of the site and the intensity and dimension of the shear lines. We are aware that you can rectify oblique imagery but at this distance and flight altitude there would be insufficient ground resolution to accurately undertake any of the suggested measurements.

L 218/ Figure 3A: this looks uncannily like a von Karman Vortex street! This signal could be removed from the CS signature on the sheer line and quantification of each made? Would all this detail not be lost if the turbine was on. Thus this is only quantifying the idle structure.. in itself a valuable bit of information - but not as the paper is being directed. Vectors in Figure 3: too small - expand this scale please.

We appreciate the comment on the vector size and have increased the clarity of the vector scale in Figure 3. Regarding the von Kármán vortex street – we have identified a classic case of this in a previous paper concerning flows going past a monopile:

Lieber, L., Nimmo-Smith, W.A.M., Waggitt, J.J. et al. Localised anthropogenic wake generates a predictable foraging hotspot for top predators. *Commun Biol* 2, 123 (2019).

<https://doi.org/10.1038/s42003-019-0364-z>

However, here we have originally referred to the displacement of the wake in the Y-location as ‘meandering’, but this has caused confusion (see previous review). However, it is not a von Kármán vortex street and also has to do with the gridding of the data between ADCP lines.

L 260-263: time gap between these measurements - how much has the tidal range changed in this time?

There is no time gap, we have just shown the difference between a central region of the transect and the rest of the data.

L 280-282: clear statement somewhere as to when the O2 would be operating; what stage of the cycle - which is important to study? or does it work on all tidal states?

The minimum velocities for a tidal turbine to operate vary highly between devices, but no, tidal turbines do not operate in every tidal state as they need a minimum current for the turbines to be operating. In the case of the O2, the cut-in current speed is 1 m/s. We have incorporated this statement in the Methods in Line 601.

L 286: plan view... quantify.. I won't keep repeating. Sorry

Again, this particular shot is oblique for a reason, please see above.

L 291: instantaneous... maybe actually a time frequency/rep rate of camera would be more realistic

Changed to: ‘Incorporating all of the instantaneous (4 Hz) velocity fields over a two-minute hover,...’ in Line 289.

L 232-235: the idle o2 far-wake...

This is clarified in the next sentence and the wake is still deriving from the O2 platform.

L 327: yes, very much so, and useful to see how idle platforms will still interfere with CS/wake/sediment transfer. but if non-idle I don't believe these noticeable and quantified wake signals would persist, or rather - they would be obfuscated.

We have already responded to this comment above. In essence, it is paramount to quantify the wake even in the idled state and the wake signature will not be obfuscated but exaggerated.

L 477: these PIV techniques could be validated if you rectify your oblique images and quantify the boil sizes.. This would be a worthy endeavour!

We have already commented on this above. Briefly, boils sizes can be quantified directly from our nadir drone imagery, with more accuracy than using oblique and then rectified imagery, however this was not the aim of this paper and would provide only limited additional value to the velocity fields (and associated direct measurements of turbulent fluctuations and divergence) that we already include.

L 510-511: I agree with this whole heartedly, the issue this gives is I'm not sure this is nature worthy, it's first insight yes, but not headline

We appreciate the reviewer's perspective, but in fact it is critical to be able to trace a wake using low-cost tools – similar approaches are currently being investigated for quantifying the wake signatures of offshore floating wind structures (which are 'idle' too but still have an impact on water column mixing).

L 511-513: you provide insight into the wake of a floating platform, not really of turbine wake, and the evolution would not be intensified as such, but likely completely different..?’

We have clarified our meaning by replacing the word “evolution” with “signatures” as it is these that will be intensified.

L 540- 542: this paragraph sets a scene for why the idle configuration is useful... this should be reported earlier and followed to make the wake dynamics more viable

We thank the reviewer for this comment and have modified the associated section in the introduction accordingly in Line 88-90. Please note, we also make reference to this in the abstract: ‘...there is also uncertainty in how devices change the natural environment.’.... & ‘...energy-environment nexus....’.

L 591: 5-7 knots on all transects? FS and BS?? changes averaging time of bins.. see below. Happy to be wrong here, as the results seem sound, but I cannot find this information in the paper or supp mat. Back of Envelope Calculation: if each transect is ~25 mins.. each excursion is 400 m line: $6 \times 400 = 2,400$ Plus connecting edges: $200 + 4 \times 100 = 600$. Total distance = $2400 + 600 = 3000$ m. 25 mins = $25 \times 60 = 1,500$ seconds. Speed = $3000 / 1500 = 2$ m/s . One excursion = 400 m, so time for that excursion = $400 / 2 = 200$ s. $200 / 60 = 3.3333$ mins!!! Sanity check: using knots: 5-7 knots in methods = 3 m/s - so faster!!! $400 / 3 = 133.3333 / 60 = 2.2222$ mins!!

Each bin in plot S2 appears to be bins of size, hmmm different... $50 / 3 = 16.6667$. $50 / 4.5 = 11.1111$. $50 / 4 = 12.5$. $50 / 5.5 = 9.0909$. $(17 + 11 + 13 + 9) / 4 = 12.5$. So let's say each bin about 12 m. Sanity check: $12 \times 4 \times 50 = 2,400$ m excursion.. 12m @ 2 m/s gives bin average of $12 / 2 = 6$ s!!! $12 / 3 = 4$ s!!!! Surely a 6s or 4s averaging time is not enough to capture mean flow dynamics: see: Marian, M., Kim, J. and Kim, D. (2021) ‘Impact of the sampling duration on the uncertainty of averaged velocity measurements with acoustic instruments’, Hydrological Processes, 35(4). doi: 10.1002/hyp.14125.

This appears to be related to the comment below on ensemble intervals – please refer to our response there.

L 607: waves when 2 on beaufort???

This statement related to the winds and associated drone surveys – please refer to our summary table for associated mean wind speeds on the 15th and 16th.

L 611: is this period important for turbine output and thus needs to be included or can be missed - clarity would help.

In an ideal world, any tidal period would provide important information for the turbine developer. We have tried our best to cover the tidal cycle within practical constraints.

L 614: waves.. see above... calc stokes..

This appears in relation to the depth of the ADCP mounting pole that is addressed below.

L 615: Sorry - just spotted this.. I'm not convinced that 4.22 s per bin/ensemble is enough to capture flow characterisation in such energetic environments. That's only about 8 pings at 2Hz..

Yes, if we were dealing with a static ADCP, these comments would hold true – however, the clearly stated purpose of our observations was to undertake moving vessel ADCP measurements where there is a trade-off between collecting data at any single location versus covering the survey area within a short enough time frame that mean flow is not changing due to the tides. There will be ensemble-to-ensemble variability due to passage through large-scale coherent structures and this is demonstrated by the standard deviation envelopes in Figure 9.

L 623-625: The profiling range of a 600 is 70m? are the velocity readings dubious, or is there a correlation coefficient/error velocity error across beams from CS?

No, the profiling range of an RDI 600 kHz Workhorse is not 70m, it has a nominal range of about 50m in mode 1 broadband. Yes, error velocities will increase as the beams spread with increasing range. Our focus is on the near-surface velocities and what the reviewer refers to could be a topic of a different investigation which is outside the scope of this study.

L 716: mean flow speed, not the speed of the measured quantities?

We have corrected this to: 'For our typical flow speeds'.. in Line 747.

Supplementary Information:

Fig. S1: as reviewer 1 comments - highlighted - could these transects at similar tidal states not be meant to get a better idea of background flow across tidal range rather than transect.

This has been resolved already and we provided detail in the last revision.

Fig S1 C: is this clearly surface turbulence away from o2 that is on boil/shear line? Also on upstream side of 02, with translation from bed?

Yes, correct. The dominant features here are large bubble plumes (surface connected) corresponding to crossing the shear line. Additional scattering from e.g. suspended sediment

is observed elsewhere in the water column (near the bed), e.g. up- and downstream of the O2.

Fig. S1 E: surely this is convincing.. poss the wake is reducing this 'coherency'?

Again, this is the shear line arising from the ebb flow past Muckle Green Holm and has nothing to do with the wake signature of the O2 which is well over 500 m away from it (in the cross stream direction).

Fig. S1 H: no boils here now, but boils & wake now on opposite head point. is this the propogation downstream of teh boil signature as you see in your highlighted plots

Correct, the shear line forms in the wake of the island (Muckle Green Holm), thus the signature is not there during the flood tide. What you can see in the far distance (upstream of the O2) is the wake of the derelict Hydro platform as well as bubble entrainment from turbulence structures in the free-stream. Please refer to the fine-scale plots for the wake signature of the idled O2 as well as figure S6 in the Supp. Info.

Fig. S1: seems to be enough for 'time means' e.g slack mean, ebb accelerating mean etc..?

We appreciate the reviewer's perspective, but believe our chosen and more straight-forward presentation of the data that captures the reality of the flow variability, rather than the presentation of 'time means', is more useful.

Fig. S1 caption: depth avearaged?

Thank you, that has been added to the caption.

Fig. S2 caption: not magenta

Thank you, that has been edited in the caption.

Fig. S5: are flow structures here extending from the bed? thus you have a surface signature from the wake (seen in image-processed Sv) that is interacting with structures already there in this tidal state?

Yes, there certainly are less strong signatures arising from the bed, which are most likely the suspension of sediment given the scattering response (sediment re-suspension scatter less compared to bubbles) and increasing intensity closer to the source (seabed). However, the scattering directly underneath the wake is an artefact (please note the line-like shadowing of the wake above) and due to excess attenuation and multiple scattering – please refer to our response above and the new Figure S10 with its comprehensive caption in the Supplementary Information. This multiple scattering has been removed during the de-noising and thresholding. Here, our focus was the isolation of the true wake signature (through its bubble entrainment) for tracing, while sediment resuspension near the bed was filtered out for this purpose as it is very much a secondary effect.

Fig. S5 caption: correlation coefficient? error velocity?

This is EK80 echosounder data and there is no associated velocity data.

Fig. S5 caption: why denoise? how - what are you removing - what wrong with original? see supposition above about signature from bed - that is being removed by this denoising.. what is your rationale for needing to do this - is this data not real?

Please refer to our response above.

Fig. S5 caption: yes! but there seems to be more.

Yes, we agree that there is e.g. sediment suspension but we do not focus on this in our analysis of the wake, please see our more detailed responses above.

Fig. S5 caption: what is the area here? aural extent of transect? - why not the same? Unsure

It's the cross-sectional area (spread in m^2) of the isolated wake area shown in C of the same plot.

Fig. S6 caption: black circle? I see on A and B but not on C or D

This has been corrected.

Fig. S7: how is TI so low in region of clear turbulence in the wake..?

Please note, we have now revised these plots throughout. Please also refer to Line 561 in the Discussion of the main article and please also refer to our updated plots of turbulence length scales in response to Reviewer 6.

Fig. S10: autocorrelation is not a method that can discern large CS easily. this study uses the first zero crossing point, thus larger scale correlations could be missed.

Prompted by a comment from Reviewer 6, we have now updated our method to estimate L_u so that it now uses a spatial correlation approach as detailed in the Suppl Information Methods Section S2, page 27.

Comments on previous round of reviews:

Wake: can we say what it is.. when it looks like so many.. how relevant if we care about the wake when 'active'? do we care about the wake when 'inactive'? would the current wake not erode or as reviewer says interact with to form new structure

Yes, the wake resulting from the inactive state is still important for environmental consenting and engineering aspects concerned with loading etc. Please refer to our more detailed response at the start.

ADCP processing: no info on mode of measurement that I can find - affects statistics. I'm not sure 1.35 m is enough to not be affected by the vessel - have you

quantified this? Ships are generally 10-15 m, but the scale is quite different. Would be nice to know you've done a BoE calc for it.

We have clarified this in our Methods Line 640. Although this is not normally necessary for moving vessel applications (other modes are used in static ADCP deployments).

Yes, the survey poles were specifically designed to match the vessel hull configuration and we have validated the depth of the ADCP transducer as well as the depth of EK80 in the field at that site prior to data collection – both were submerged sufficiently deep to return good quality data at the survey speeds. If they had not been deep enough to prevent bubbles from obscuring the transducer, there would have been data drop-outs.

(Answers to specific editorial questions regarding responses to Reviewer #1's previous comments:)

1.) Do you feel that the revised manuscript's coverage of literature about turbulence measurements at tidal energy sites is sufficient to provide adequate context of the work to readers?

→ I feel yes, for this journal type and space available.

OK.

2.) Is the manuscript's description and documentation of flow at varying tidal states clear and presented in sufficient detail?

→ With the supp material yes, but tidal state means would be advantageous. Notwithstanding the concerns I have on ADCP averaging time and description of what is wake and CS and separation of such.

We are uncertain about the specific interpretation of this comment. However, in Table S1, the summary table included in the supplementary materials, we indeed present tidal mean velocities for each transect line along with the corresponding tidal states, indicated as, for instance, "Ebb Accelerating." Additionally, in all figures, we consistently reference the given tidal state and the mean flow velocity measured by the O2 Acoustic Doppler Current Profilers (ADCPs), which served as the source for background/reference tidal velocities throughout.

3.) Reviewer #1 previously commented that the use of 2 minutes of PIV data is insufficient to obtain robust turbulence statistics and that at least 5 minutes of data are required. The authors have responded by revising their manuscript to indicate that there needed to be compromises in the measured data between overall flight time and being able to capture various sites (upstream and downstream hovers) under the assumption of Taylor's frozen field. Additionally, they indicate that TI and velocity magnitude measurements converge over time in the freestream but do not fully converge in the shear line region due to the presence of large intermittent, turbulent coherent structures. Could you please

comment on this revision, and indicate whether this sufficiently addresses the reviewer's concern?

→ Noted and I feel the response is adequate.

OK.

4.) Does the revised manuscript sufficiently describe how the PIV data is processed, and how divergence and vorticity are calculated?

→ Sorry, PIV is not my expertise.

OK. Please note, we have added more detail to the PIV processing in the Supplementary Information (Methods Section S1, page 26) and have provided more detailed responses to Reviewer 6.

5.) For the ADCP processing, is the gridding sufficiently described?

→ Yes, but a bit buried (I'd done the calc's manually before I found the info) and I'm not sure on the processing validity in such flows.

These details are clearly stated in Line 659-662 of the Methods section.

6.) Is the description of the shear line impingement in the revised manuscript clear?

→ Yes.

OK.

7.) Are the reduced wake velocities and velocity deficits in the revised manuscript's figures clearly interpretable?

→ Yes.

OK.

8.) Is the error of turbulence intensity estimation with PIV clearly represented in the revised manuscript?

→ Sorry, PIV is not my expertise. As a layman to this field I do feel they have given an error in frame to frame referencing, but they have computed this with mean flow field speed – this will not be the speed of the CS or wake dynamics they are measuring but rather the speed that these signatures will be advected... I feel this should be detailed more.

Yes, the turbulence intensity is the root mean square of the velocity fluctuations divided by the mean flow speed. In our calculations we use the location-specific mean, which we have now clarified at Line 740. By definition this will capture the variations in the mean flow in the wake and outside of it (including the passage of any CS). In addition, we have expanded our review of potential errors arising from drone stability (Suppl Info Methods Section S3, page 28).

9.) Do you consider the revised manuscript's discussion section's comparison with other studies on submerged turbines and floating turbines to be sufficient?

→ I feel that the discussion section is thorough and compares well with other work. The wake dynamics section should be highlighted earlier in the MS; for relevance. I'm not sure all the statements are robust though, again – comments inline.

Dr N S Lucas

OK. We have modified the paragraph on the wake dynamics in the introduction to better link with this section's discussion.

Reviewer #6 (Remarks to the Author):

I have carefully read the paper “Troubled waters: Revealing sheared turbulent flows and wake dynamics of a floating tidal turbine platform” focusing on the questions bellow for which I provide my opinion.

Questions related to Reviewer #1's remaining concerns:

1.) Does the revised manuscript sufficiently describe how the PIV data is processed, and how divergence and vorticity are calculated?

→ The paper does not sufficiently describe how the PIV data is processed. In fact, PIV stands for Particle Image Velocimetry and nowhere in the text the authors provide any information to what type of particles were used by the authors, and what is their effect on the velocity determination, as this technique measures the velocity of particles as a proxy of the velocity of the fluid $u_{particles} = u_{fluid}$. The choice of particles is therefore important to assess how good is the data.

While lab-based PIV relies on seeding particles for flow visualization and velocity measurements in controlled settings, the term ‘Large-scale particle image velocimetry (LSPIV)’ is often referred to when using PIV approaches in natural environments such as rivers, estuaries, or oceans. LSPIV utilizes natural features or tracers for image-based or remote sensing-based velocity measurements which makes LSPIV well-suited for studying large-scale flows (order of meters) in natural water bodies.

Typically (see References listed below), although there are exceptions (as seen below in Strelnikova et al., 2020), LSPIV doesn't require seeding the flow with particles. Instead, it relies on natural features or tracers present in the fluid, such as surface waves, boils, debris, foam lines (as seen below in Lieber et al., 2022), or other high-contrast features.

In our study, the 'particles' we track are thus natural tracer features. To prevent confusion and to align with the below review by Muste et al. 2008 (which we have now added as a reference to Line 101 & 734), and more recently, Fairley et al. (2022), who also employ the

term 'LSPIV' in coastal ocean settings, we now refer to our PIV approach as 'LSPIV' in the Results and Methods but use the term 'PIV' interchangeably throughout the manuscript.

In summary, LSPIV has proven to be a valid and cost-effective image-based tool for measuring surface velocity using natural tracers in various fluvial systems, particularly when applied in the field rather than controlled laboratory environments. To-date, LSPIV remains predominantly applied in riverine environments (except for Fairley et al. & Lieber et al.). In addition to the Fairley et al. 2022 reference cited in the manuscript, further studies utilizing LSPIV approaches mostly in natural river settings (including drone-based LSPIV) are provided below in chronological order:

1. Muste, M., I. Fujita, and A. Hauet (2008), Large-scale particle image velocimetry for measurements in riverine environments, *Water Resour. Res.*, 44, W00D19, <https://doi.org/10.1029/2008WR006950>
2. Lewis, Q. W., and B. L. Rhoads (2015), Resolving two-dimensional flow structure in rivers using large-scale particle image velocimetry: An example from a stream confluence, *Water Resour. Res.*, 51, 7977–7994, <https://doi.org/10.1002/2015WR017783>
3. Tauro, F., Petroselli, A., Arcangeletti, E., (2016). Assessment of drone-based surface flow observations. *Hydrol. Process.* 30 (7), 1114–1130. <https://doi.org/10.1002/hyp.10698>
4. Tong Jin, Qian Liao (2019) Application of large scale PIV in river surface turbulence measurements and water depth estimation. *Flow Measurement and Instrumentation.* <https://doi.org/10.1016/j.flowmeasinst.2019.03.001>.
5. Zhu, X., & Lipeme Kouyi, G. (2019). An analysis of LSPIV-based surface velocity measurement techniques for stormwater detention basin management. *Water Resources Research*, 55, 888–903. <https://doi.org/10.1029/2018WR023813>
6. Strelnikova D, Paulus G, Käfer S, Anders K-H, Mayr P, Mader H, Scherling U, Schneeberger R. (2020) Drone-Based Optical Measurements of Heterogeneous Surface Velocity Fields around Fish Passages at Hydropower Dams. *Remote Sensing.* 12(3):384. <https://doi.org/10.3390/rs12030384>
7. Lieber, L., Füchtencordsjürgen, C., Hilder, R.L., Revering, P.J., Siekmann, I., Langrock, R. and Nimmo-Smith, W.A.M. (2022), Selective foraging behavior of seabirds in small-scale slicks. *Limnol. Oceanogr. Lett.*, 8: 286-294. <https://doi.org/10.1002/lo2.10289>

We have now clarified this in the manuscript from the start and edited the Methods section accordingly, providing a full account within a new section in the Supplementary Information (Methods Section S1, page 26).

There need to be a sufficient amounts of natural tracer features to allow good quality data and this is determined by the correlation threshold whereby smooth water will give very low correlation, whereas textured water will give higher correlation values. The PIV methodology is essentially using the cross-correlation between small regions of successive images. The Signal to Noise ratio metric that the reviewer addresses below is used in lab-based PIV and presents a measure of the relative size of multiple correlation peaks where distinct tracer particles are used – this is a less useful metric for where features have a more continuous variation across the window, such as when using PIV in natural environments.

The authors reference a previous paper of their co-authorship (Lieber et al., 2021) but even there nothing is mentioned about the type of particles used.

Please see our response above for more detail. The PIV approach uses natural feature tracers rather than particles. Generally, flows are only seeded with particles in controlled environments (laboratory/flumes etc). While there have been attempts to seed more

constrained natural river flows with particles, using e.g. 'ecofoam', practical considerations, such as the quantity of seeds needed, the non-constrained channel flow of our tidal stream site, as well as ecological concerns make this approach unfeasible. Given the site's ecological sensitivity with the diverse seabird and marine life populations in the area, the potential biological and ecological effects of particle releases remain unknown and are thus deemed impractical and unfeasible for such natural sites.

Other examples of using naturally occurring tracers for PIV analysis include Nimmo-Smith et al. 2003 (and related papers) using submersible PIV to quantify turbulence in the coastal environment (using suspended particles) and Hong et al. 2014 using PIV on snowflakes to study the wake of wind turbines:

Nimmo-Smith, W. A. M., J. Katz, and T. R. Osborn, 2005: On the Structure of Turbulence in the Bottom Boundary Layer of the Coastal Ocean. *J. Phys. Oceanogr.*, 35, 72–93, <https://doi.org/10.1175/JPO-2673.1>.

Hong, J., Toloui, M., Chamorro, L. P., Guala, M., Howard, K., Riley, S., Tucker, J., & Sotiropoulos, F. (2014). Natural snowfall reveals large-scale flow structures in the wake of a 2.5-MW wind turbine. *Nature Communications*, 5(May). <https://doi.org/10.1038/ncomms5216>

They claim to use in this paper the same methodology and even the same parameters as they used before (Lieber et al. 2021), but no justification is presented for, for example, the use of a correlation coefficient of 60%. This is for me an unusual parameter, as in PIV one of the quality data selection parameters is the Signal to Noise ratio. Typical minimum values of SNR are $SNR > 1.2$.

Please see our response above and Supplementary Information (Methods Section S1, page 26) for more detail. Rather than using the Signal to Noise ratio metric commonly applied in lab-based PIV, we use a direct correlation threshold when measuring velocities using natural features as compared to seeded flows. This is a well-established method, and although we use custom-code, it is also a common approach when using the MATLAB toolbox PIVlab, please see the references below:

1. Raffel M, Willert CE, Wereley ST, Kompenhans J. 2007 Particle image velocimetry: a practical guide. New York: Springer Press. 79–92
2. Thielicke W, Sonntag R (2021) Particle Image Velocimetry for MATLAB: Accuracy and enhanced algorithms in PIVlab. Journal of Open Research Software, 9: 12. DOI: <https://doi.org/10.5334/jors.334>
3. Fairley, I., King, N., McIlvenny, J., Lewis, M., Neill, S., Williamson, B. J., Masters, I., & Reeve, D. E. (2024). Intercomparison of surface velocimetry techniques for drone-based marine current characterization. *Estuarine, Coastal and Shelf Science*, 299. <https://doi.org/10.1016/j.ecss.2024.108682>

The density of particles is crucial for PIV calculations and nowhere is stated if a proper density of particles was used.

Please see our responses above for more detail - this is only true for seeded flows. Otherwise, it is in the density of discernible features that provide the correlation – so that when there are no features (smooth water) then the correlation coefficient falls below the threshold set (see above). We have included a comment on this in our new Supplementary Information section.

PIV software packages usually include multi-step and window distortion algorithms to better adjust the processing to the flow gradients and improve spatial resolution. Nowhere in the text the authors describe what was their software (this contrasts with the amount of information given about the ADCP measurements, lines 613-622). In my opinion, the authors should clarify if they use a commercial software or an in-house developed PIV and eventually how does their solution compares with existing solutions. After, a clarification can be made regarding the authors claim: “with a resolution of 3.7 cm per pixel” (lines 701-702) and the depicted velocity field of Figure 5, for example, where only 18 vectors/~80 m are depicted corresponding to a resolution 1 vector every 18m.

We sincerely apologise for the confusion we have caused by referring to our previously published (open-access) paper and the Methods section therein (Lieber et al. 2021). Briefly, we have not used any third-party software, but custom code and we have now detailed the methods in the Supplementary Information (Methods Section S1, page 26) including detailing the choices made to ensure robust outputs over applying more complex algorithms better suited to controlled laboratory data. We’ve now included the vector spacing within the Methods section at Line 732. We’d like to further point out that we have sub-sampled the vectors within Figure 5 (and similar figures) for clarity in presentation of the data and we have now specified this step in the corresponding caption (Fig. 5).

No reference is made to peak-locking and the authors should clarify if the PIV data had any significant peak-locking.

This is when you use low seeding density of under-resolved small particles such that there is ambiguity in their actual sub-pixel location – as above, we are not using seeding particles and our method does resolve sub-pixel locations in correlation peaks as described in the new section in the Supplementary Information (Methods Section S1, page 26).

Another question I have concerning the data processing is the fact that images were acquired at 30Hz (that is $dt = 0.033$, line 690), but in line 703 it is stated that: “instantaneous velocity fields are acquired at 0.25 s interval” (that is $f = 4$ Hz). This should be clarified, if the authors used a time averaged of about 8 images to have a so called instantaneous value every 0.25 s, or resampled the data set.

This was detailed in our cited reference but we have now expanded on this in the Supplementary Information (Methods Section S1, page 26). The instantaneous velocities are derived from 4 consecutive video frames (using median filtering) and so cover a period of time of duration 0.1 seconds. This time scale would be significant if undertaking micro-scale measurements within a laboratory setting, but are essentially “instantaneous” when considering the time and space scales (order metres and seconds) within our field observations.

In lines 714-719 the authors provide an estimation of errors due to the stability of the drone. The authors essentially use the estimations provided by Fairley et al. (2022). However, the authors should better explain how these values lead to the error of 3.5% in the turbulence intensity.

We thank the reviewer for their comment on drone stability. Prompted by another reviewer as well, we have now assessed drone stability: We have flown the study-specific drone over a static surface (beach scene) in strong and gusty winds beyond the conditions encountered during the study to assess the drone's stability. We have added a new section to our Supplementary Information (Methods Section S3, page 28) detailing the outcome of this and amended our manuscript at Line 745-748 accordingly. We have also noted that in the Fairley et al 2022 paper, the authors only report on the mean wind speed, without including the gusts. However, following our assessment with a detailed in-flight wind assessment, we can now show that the gusts had the greatest impact on drone stability. We have now explained how we have derived a very conservative Turbulence Intensity threshold using the outcomes of that additional assessment.

When the authors estimate that the geolocation error is 2.1 m, does this mean that the location of the vectors in the map (e.g. Figure 5E) is affected by the same error?

Yes, this drone-intrinsic error translates to the location of the vectors and will lead to a small degree of spatial smoothing of time-averaged statistics (e.g. Figure 5C) in addition to that intrinsic to the 50% overlap between adjacent PIV processing windows. We now report the variance in the position of the drone during each hover as part of our expanded assessment in the Supplementary Information.

Using the Matlab's divergence and curl functions to determine the divergence and vorticity is not a problem, the problem in my opinion is the quality of the input data (velocity field) which I can't assess based on the information provided by the authors.

Fairley, I. et al. Drone-based large-scale particle image velocimetry applied to tidal stream energy resource assessment. *Renew Energy* 196, 839–855 (2022)

Lieber L, Langrock R, Nimmo-Smith WAM. (2021) A bird's-eye view on turbulence: seabird foraging associations with evolving surface flow features. *Proc. R. Soc. B*, <https://doi.org/10.1098/rspb.2021.0592>.

We have now expanded on this in the Supplementary Information (Methods Section S1, page 26).

2.) Is the error of turbulence intensity estimation with PIV clearly represented in the revised manuscript?

→ First, I would recommend the authors to rewrite the equation on line 709 as

$$TI = \sqrt{\overline{(u')^2}} / U$$

As u' is normally used as turbulent fluctuation. Secondly, I would suggest the authors to define what mean velocity, U , was used and how it was computed, namely If it's a spatial (in the field of view of the PIV measurements) and time averaged velocity, if it's only a time averaged velocity and how it was computed.

We thank the reviewer for their suggestion. We have originally stated part of the equation in the text but have now edited the equation as suggested and clarified how the means were computed (location specific time mean). Please refer to Line 739-740.

In Figure S9 it would be useful to represent all the data series used for the spatial average to see their dispersion and better assess the validity of the convergence of both mean values and turbulence intensity. Ideally, this convergence study should have been made for each interrogation window of the PIV

Thank you – we have now added additional information to updated Figure S11 in the Supplementary Information to show the individual data series that have contributed to the spatial average.

I would recommend the authors the paper by Hitching and Lewis (1991) that provides a way to determine the number of data points necessary to give statistical convergence in fluid measurements. These authors estimate about 3500 the number of samples needed. Also, the paper by Mycek et al. (2014) provides valuable information about convergence of time series in single turbine measurements.

Thank you for providing these references.

Here it is important to clarify what was the data acquisition frequency (if 30 Hz if 4 Hz) as stated in the previous comment, because if the acquisition frequency was 30 Hz the authors would have 3900 images (30*130), but if 4 Hz were used the number of samples 520 (4*130) is clearly not enough to ensure a convergence of the mean results). I would suggest representing figure S9 indicating in the upper horizontal axis the number of samples.

We have now clarified the sampling rate and hence the number of samples included in the spatial average of temporally-calculated statistics (updated Figure S11), opting to provide the number of samples within the figure caption.

Hitching, E., & Lewis, A. W. (1999). Sampling techniques for accurate measurement of Reynolds stresses using laser Doppler anemometry. *Flow Measurement and Instrumentation*, 10, 241–247.

Mycek, P., Gaurier, B., Germain, G., Pinon, G. & Rivoalen, E. (2014). Experimental study of the turbulence intensity effects on maritime current turbines behaviour. Part I. *Renewable Energy*, 66, pp 729-746.

3.) Do you consider the revised manuscript's discussion section's comparison with other studies on submerged turbines and floating turbines to be sufficient?

→ In the Discussion section, the authors mainly point out the merits of their monitoring approach regarding other previous publications in tidal flows, in that sense the performance of submerged of floating turbines are not the main focus.

Thank you, we have now also added limitations to our study and design to help steer future research in this area. These are located in Lines 484-487; Line 498-502; Line 540-547; and Line 561-564.

Questions related to Reviewer #4's remaining concerns:

1.) Reviewer #4 previously commented that the original manuscript's link of the observed ADCP signal to bubbles as a signal of turbulence seems speculative. The authors have responded by revising their manuscript to further explain the observed phenomenon, indicating that this type of observation has been used widely in physical oceanography to map distributions of near-surface physical flow. Do you feel that this has been sufficiently discussed in the revised manuscript, and that the authors' connection between bubbles observed in the ADCP signal and turbulence is correctly represented?

→ It is a fact that water bubbles are good acoustic tracers, but ADCP's also react to suspended sediments. The ADCP uses the Doppler effect to measure the velocity of tracers dispersed in the water column. These tracers can be suspended sediment, air bubbles, organic matter. Turbulence measurements require fast sampling techniques. According to the RDI's ADCP manual, the instrument has a sampling frequency of 2Hz (ping frequency) which limits the turbulence measurements to events with $f < 1$ Hz (Nyquist-Shannon criterion). The information regarding ADCP acquisition frequency, and its limitations should be clearly stated.

Greene, A.D., Hendricks, P.J., Gregg, M.C. (2015). Using an ADCP to Estimate Turbulent Kinetic Energy Dissipation Rate in Sheltered Coastal Waters, Journal of Atmospheric and Oceanic Technology Journal of Atmospheric and Oceanic Technology, 32,2 DOI: <https://doi.org/10.1175/JTECH-D-13-00207.1>

We thank the reviewer for their input and fully agree that ADCPs need a higher sampling frequency (order of 4 Hz) and ideally be stationary (e.g. bed-mounted/on a mooring) to fully resolve turbulence. We have already revised these sections in the previous version and largely refer to it as 'surface-connected acoustic scattering', 'bubble entrainment', or as 'bubble entrainment by macro-turbulence'. At no point are we attempting to directly measure the turbulent velocity fluctuations directly from the ADCP.

2.) Reviewer #4 previously commented that the wave field received little attention in the original manuscript and asked what drove the changes in the results between the two ebb cycles observed. The authors revised their manuscript to provide full environmental conditions during their measurements and indicated that, since there is no clear difference in wind speed, direction and wave height between the two ebb cycle transects, they do not have a clear explanation for the observed variability and that it could be associated with flow variability at larger scales than their measurements capture. Could you please comment on this and indicate whether these concerns are sufficiently addressed in the revised manuscript?

→ The authors included a table with a complete description of the field conditions which is positive. However, in their response they point out to line 523 where nothing is mentioned about wave effects. The authors' hypothesis regarding flow

variability at larger scales is plausible and it should be clearly stated in the manuscript.

Thank you, we have now included a clear statement to this effect in the Results section at Line 334-336.

3.) Could you please comment on the technical accuracy of the revised manuscript's consideration of wake propagation in the results and discussion?

→ The fact that the turbine is idle limits the true assessment of the turbine wake. However, it provides a sort of “base case” that may be useful to establish the impact of such structures. The authors rightfully mention regarding the importance of assessing the wake dynamics the need to determine the energy dissipation (line 91). However, the authors do not develop this issue.

We thank the reviewer for their support. We agree that we are investigating the wake of an idled rather than an operating turbine, however, as outlined previously, these baseline characterisations are paramount for array planning, consenting and provide important engineering insight on thrust and loading. Please note that we have now edited the title of this manuscript to reflect the idled state. While we do not develop energy dissipation further, we believe that would be a worthwhile investigation when the turbine is operating. We also thank the reviewer for their reference below and we have made a reference to it in the introduction in Line 90 to direct the reader.

Thiébaud M., Filipot J.-F., Maisondieu Ch., Damblans G., Duarte R., Droniou E. & Guillou S., (2020) Assessing the turbulent kinetic energy budget in an energetic tidal flow from measurements of coupled ADCPs Phil. Trans. R. Soc. A.3782019049620190496

The wake analysis provided by the authors rely on the velocity deficit and on the estimation of the wake area both dependent on a threshold criterion (0.75 m/s). How universal is this criterion?

Rather than this being a universal criterion, we used this value following assessment of our data for defining the tidal states (also summarised in the Supplementary summary table) based on mean velocities as measured by the O2 ADCPs in line with some further interpretation of our individual ADCP transect lines. Peak currents exceeded 3 m/s and the threshold for slack water was set to ≤ 0.5 m/s, with the tidal currents only starting to set in above this value. Transect 6 on 15/04 was the closest transect in relation to slack with a mean tidal velocity of -0.78m/s – we have hence used 0.75 m/s to define the threshold for e.g. plotting the wake in Fig. 7.

Given that the authors have ADCP transects, wouldn't it be possible to characterize the wake at each transect section as a cut into the wake cone, and therefore allowing for a 3D representation of the turbine effect downstream of it?

Yes – this is shown in Figure 6 for one transect, and then summarised for multiple wakes in Figure 9.

4.) Reviewer #4 previously commented that the original manuscript's consideration of "fluctuations in velocity" in the observed velocity series was erroneous; the authors responded by revising their results to clarify this point. Do you feel that the revised manuscript has sufficiently addressed this point?

→ Fluctuations are defined as the difference between the instantaneous velocity and the average velocity. The authors emphasis on fluctuations can be found in lines 409-414 with reference to Figure 8C. Their and it is essentially correct, however the manuscript could be improved if: i) the turbulence intensity of each time series was referred on the lines 409-414 and that in Figure 8 C would represent the fluctuations (also, or instead, of the instantaneous values).

We thank the reviewer for their helpful suggestions. We have now re-plotted Figure 8 to show u' and v' (velocity fluctuations) as suggested. We have also modified the accompanying text in Line 412-425.

5.) Is the use of the term "shear lines" clearly described and discussed in the revised manuscript?

No. The term "shear line" is not well defined in the text not even supported by the references used by the authors (Horwitz and Hay, 2017, Neil and Elliot, 2004, Wolanski et al. 1996). Nowhere in those texts the expression shear line is used. I agree with Reviewer 4 and perhaps the concept of "shear layer" fits better here.

Horwitz, R. & Hay, A. E. (2017) Turbulence dissipation rates from horizontal velocity profiles at mid depth in fast tidal flows. *Renewable and Sustainable Energy Reviews* 114, 283–296.

We have now added the term 'shear layer' to this paragraph, see Line 55. However, we would like to direct the reviewer to the below oceanographic paper clearly making use of the term 'shear line', the region at the boundary or interface of leeward island/headland wakes and non-wake waters, which we have described as waters "bounded by regions of strong horizontal shear (cross-stream gradient in streamwise velocity)". We have now replaced the previous references with this key reference:

1. Johnston, D. W., & Read, A. J. (2007). Flow-field observations of a tidally driven island wake used by marine mammals in the Bay of Fundy, Canada. *Fisheries Oceanography*, 16(5), 422–435. <https://doi.org/10.1111/j.1365-2419.2007.00444.x>

Please also note that the above paper presents ADCP-derived flows and backscatter data collected across a shear line in Figure 4 of the paper, highly similar to our backscatter data across the Muckle Green Holm shear line, for instance.

Further, Coutis & Middleton 2002 provided another oceanographic description of this using the following wording: "Results indicated that the shear layers separating free stream and wake currents were relatively thin. This created localised regions of strong flow convergence and divergence along the **shear lines** resulting in downwelling and upwelling, respectively..."

2. Coutis, P. F., & Middleton, J. H. (2002). The physical and biological impact of a small island wake in the deep ocean. In *Deep-Sea Research I* (Vol. 49).

There are other papers in the literature to demonstrate this, however, among physical oceanographers, the term “shear line” is most commonly used rather than the term “shear layer”.

We have previously used references where the concept of a shear line was well described or characterised. For instance, this was the case in the original reference number 7 (Horwitz & Hay 2017) which was associated with our following text in the original Line 57: ‘Vortical structures, upwelling (surface divergence) and associated downwelling are characteristic of shear lines...⁷.’

7. Horwitz, R. & Hay, A. E. Turbulence dissipation rates from horizontal velocity profiles at mid-depth in fast tidal flows. *Renewable and Sustainable Energy Reviews* 114, 283–296 (2017).

Associated with Figure 19 in Horwitz & Hay 2017 (below), they describe the eddy shedding as follows: “...eddies are being generated at positions of strong lateral shear in the tidal flow”. Therefore, this was in line with our expression of horizontal (=lateral) shear. While Wolanski & Hamner 1988 describe these as ‘separation streamlines’, similar to Neill & Elliot 2004 who describe this as: “Eddies form in the lee of islands because of flow separation occurring where streamlines break away from the boundary layer, carrying high vorticity fluid into the interior of the flow (Signell and Geyer 1991).”

[REDACTED]

LEFT: Horwitz & Hay 2017 Figure 19; RIGHT: Wolanski & Hamner 1988 Figure 2

Neill, S. P. & Elliott, A. J. (2004) Observations and simulations of an unsteady island wake in the Firth of Forth, Scotland. *Ocean Dyn* 54, 324–332.

Wolanski, E., Asaeda, T., Tanaka, A. & Deleersnijder, E. Three-dimensional island wakes in the field, laboratory experiments and numerical models. *Cont Shelf Res* 16, 1437–1452 (1996).

6.) Could you please comment on the comprehensiveness of the PIV field analysis presented in the revised manuscript? Do you feel that there is further analysis or discussion required beyond what is provided in the revised manuscript?

→ The PIV topic must be improved, and the questions raised in my previous comments should be addressed.

Editorial Note: Panels above redacted where no permission to publish could be obtained.

We thank the reviewer for their comments regarding the PIV analysis – as stated above, we have now added a more detailed section in the Supplementary Information (Methods Section S1, page 26).

7.) Do you agree with the revised manuscript's assertion that IEC 62600-200's recommendations for 'in-line' or 'adjacent' ADCP placement is incapable of capturing the spatially varying flow conditions across a dual rotor floating platform such as the platform considered in this manuscript?

→ At the moment I have no data to validate or invalidate this sentence.

We thank the reviewer for their comment – one of our co-authors has access to this document and we have carefully described the various configurations in context with our study's results.

8.) Do you feel that the flow and vorticity characteristics considered in this manuscript have been sufficiently and accurately represented/discussed?

→ No. They use the PIV data regarding which I have raised some issues. Furthermore, the spatial of the PIV grid, which is not defined, prevents me to comment further on this issue. The vorticity is the curl of the velocity field. It is a vector whose component perpendicular to the plane xy is given by $(dv/dx-du/dy)$.

It is important to have a good spatial resolution to calculate the derivatives, and from the paper it is not possible to assess what was the spatial resolution of the PIV.

As stated above, we have now added the missing detail on the velocity vector spacing in our Methods sections. Yes, good spatial resolution is required but in proportion to the relevant scales within the flow. We are using the spatial derivatives only to qualitatively show the variability in the flow in different regions, rather than undertaking a quantitative assessment of these parameters.

Furthermore, I am not convinced about the Taylor macroscale representation, namely on Figures 5F, S3F, S4F, S6F, S7F. Each interrogation window of PIV will have associated a time series, from which it is possible to determine a length scale using the same method as described by the authors, and thus have a length scale map (similar to Figure 5 D), which then can be spatial averaged and displayed as a line like the turbulence intensity in Figure 5F.

The authors use the Taylor frozen-turbulence hypothesis to convert time into space, but since they used PIV they have spatial distributed data, so they could do the correlation directly by using the definition.

Thank you for this helpful suggestion – we have now updated our analysis so that we calculate the correlation curves directly from the spatial data. We have updated all our Figures and incorporated an intercomparison between methods of calculating the Turbulence length scales in our Supplementary information (Methods Section S2, page 27). We have also included a comment on the difference observed by using each method.

Some other minor remarks

Figure 1 c): Indicates the mean location of the O2 turbine. I would suggest using an error-bar like representation that showed the maximum displacements of the O2 turbine. Or if not possible indicate in the legend what were the extremes of the measured O2 location.

We show the O2 excursion in Figure S8 in the Supplementary Information, however, we have now added this also to Figure 1 (Fig. 1F) in the main manuscript.

Line 188: I would suggest where $D=20\text{m}$ is the rotor diameter.

We have corrected this.

Figure 3: I would suggest placing the red vector used for scalar always in the same relative position. It would be helpful to the reader to have an indication on the Figure of the geographic referencing points, namely Eday (y positive direction).

Thank you, we have placed the red vector in the top right corner of each plot and included the Eday reference direction in the plots and in the caption (y positive direction).

It is difficult to understand the role of the streamlines.

The streamlines indicate the general behaviour of the flow.

Line 222: correct “re 1m^{-1} ” to “re 1m^{-1} ”

We have corrected this.

Line 223: To clarify I guess when referring to $(X/Y = 0)$ the authors mean $(X, Y) = (0,0)$?

Yes, we have edited this accordingly in the caption of Figure 3.

Lines 223, 634: The authors should clarify what they mean by “gridded at $1D$ ” It is my assumption that the authors built a grid with $1D$ spacing (with $D=20\text{m}$ the rotor diameter)

Yes, that is meant by this.

Figure 6A: The PIV hover sections looks like they are located at the bottom instead of the surface.

Thank you for this suggestion. On closer inspection we agree with the reviewer and have added thin white lines around the PIV hover section and believe the figure has improved in clarity.

Line 363: Why were 0.75 m/s chosen as velocity threshold?

This has been clarified above and we have now added an explanation for this in the text in Line 372.

Line 369: Why were 1.5 m/s chosen as velocity threshold?

This is based on the data - the wake only doubles in extent at flow velocities > 1.5 m/s - therefore, this is not a threshold but an observation.

Lacks information about PIV method. See above.

Yes, this has now been amended: Supplementary Information (Methods Section S1, page 26).

Reference 39 is not a proper PIV reference, I would suggest the following:

Raffel M, Willert CE, Wereley ST, Kompenhans J. (2007) Particle image velocimetry: a practical guide. New York: NY: Springer.

Adrian, R.J., Westerweel, J. (2010) Particle Image Velocimetry. Cambridge University Press.

The reference 17. Is incomplete.

Thank you, in our original reference 39 (Lieber et al 2021) which we refer to for our PIV-methodology (open-access publication), we have made use of the Raffel reference and have now again added it into our more detailed methods sections in the Supplementary Information. Apologies, we corrected the reference.

REVIEWER COMMENTS

Reviewer #2 (Remarks to the Author):

In this round, I observed minimal revision addressing the comments raised in the previous round, particularly concerning presentation and multiphysics, which are directly relevant to the main concerns highlighted in the initial review. Below are some suggestions for improvement:

1. Regarding the discussion of innovation in Line 126-127, it would be beneficial to incorporate supporting evidence or references and provide further elaboration. Notably, there was a measurement conducted for the flow past tidal turbines (fix-bottom) in New York's East River.
2. While most results are presented via figures showing contours, these visuals, albeit varied, may not effectively convey the associated values. One approach to enhance clarity is to supplement the contours with contour lines, labeling numerical values on these lines. Alternatively, introducing tables or x-y plots could better illustrate the data.
3. Since this study pertains to a specific site, the findings may have broader applicability if underlying patterns or rules of the data can be extracted. One method to achieve this is through dimensional analysis, presenting the data in the form of dimensionless numbers. I recommend that the authors explore this avenue and revise their work accordingly.
4. In the "Discussion" section, it would be valuable to include a discussion on the multiscale (multiphysics?) aspects of the flows. For instance, exploring how the wake changes in size, direction, etc., in response to ocean currents upstream with figures/tables.

Reviewer #3 (Remarks to the Author):

I appreciate the work done by the Authors in clarifying several methodological aspects in the new version of the paper.

I have the following minor comments:

- Methods Section S1:

1. How many clean velocity fields were computed and used in the paper?
2. What does median 3x3x3 median filter mean?

- Methods Section S3:

1. Please quantify weak vs strong wind gusts
2. Please include maximum and minimum variations in drone positions and altitude during wind gusts.

Reviewer #6 (Remarks to the Author):

The authors did a very good work in responding to the different queries and they provided additional material and clarifications that improved the clarity and quality of the paper. However, I still have some minor remarks that were either not clarified or still inconsistent, namely:

i) Regarding LSPIV/PIV

In lab flows, particles are usually seeded in the flows in order to create patterns. Particles must have certain properties, namely that their velocity is a good approximation of the flow velocity. Either patterns created by particles or by any other features such as surface waves, boils, debris, foam lines must be good proxies for the flow velocity.

Both Particle Image Velocimetry or Large Scale Particle Image Velocimetry both rely on cross-correlation, and cross-correlation can be referred as a pattern tracking mathematical operation.

Nevertheless, either in PIV or LSPIV the determination of the cross-correlation peak location is crucial for velocity determination. Except in very ideal cases, applying cross-correlation function to two time-consecutive flow images divided into interrogation windows, will result in a spatial (pixel space) distribution of peaks and valleys. Signal to noise ratio is usually defined as the ratio between the highest and second highest peak so it can also be calculated for the authors cross-correlation method. This information complements the threshold criterion used by the authors therefore I disagree with the authors' response: "The Signal to Noise ratio metric that the reviewer addresses below is used in lab-based PIV and presents a measure of the relative size of multiple correlation peaks where distinct tracer particles are used – this is a less useful metric for where features have a more continuous variation across the window, such as when using PIV in natural environments."

I also disagree with the authors regarding their answer: "including detailing the choices made to ensure robust outputs over applying more complex algorithms better suited to controlled laboratory data." those complex algorithms were developed to improve the cross-correlation results independently of controlled lab images or not.

These disagreements don't invalidate the authors' work and are here stated just for the purpose of scientific discussion.

In my previous review I wrote: "No reference is made to peak-locking and the authors should clarify if the PIV data had any significant peak-locking." The authors replied:

"This is when you use low seeding density of under-resolved small particles such that there is ambiguity in their actual sub-pixel location – as above, we are not using seeding particles and our method does resolve sub-pixel locations in correlation peaks as described in the new in section in the Supplementary Information (Methods Section S1, page 26)."

Subpixel resolution is applied to determine the location of the selected peak of the cross-correlation function. When particles/feature are under-resolved ($d \leq 2 \text{ pix}$), the correlation function peak location tends to be in integer numbers of pixels.

Given the novelty of applying LSPIV to coastal areas and to strength the authors' results I would suggest to the authors to add in the supplementary material document a figure with the histogram of the measured cross-correlation displacements for a given time instant to assess if the natural features used to determine the correlation don't lead to peak-lock.

ii) Shear line

In the rebuttal the authors wrote: "'shear line', the region at the boundary or interface (...)" ; if, by definition, a region has an area, then a line can't be a region, because the area of a line is zero. An example of this incoherence is stated on line 414: "In the shear line area".

In Figure 4B the authors drew a diagonal shear line, and probably had they measured other sections parallel to that one, they would obtain a family of shear lines that would form a surface. In that sense it would result that a shear line is just the intersection of the shear surface with the velocity field measuring plane provided that the velocity gradient obeys to a given condition. What is the velocity gradient needed to define a shear line is perhaps the question that needs to be answered, as otherwise remains a concept more qualitative than quantitative.

As the authors stated, and provided references included in the text, the term is often used in oceanography, in a manner not much different from the authors' own usage, therefore, although disagreeing, I accept the authors' expression provided that:

a) the incoherences such as line 414, 419-420 where the authors write: "In the shear line area" are corrected.

b) in order to make a better liaison between the text (e.g. lines 300 and 303) and figures (e.g. Figure 5 E F G H) please indicate the shear lines on the figures. Same for Figures 8 A and B

c) in the supplementary material add the shear lines in Figures S4, S7 and S8

In the rebuttal the authors wrote about the use of divergence and vorticity: "We are using the spatial derivatives only to qualitatively show the variability in the flow in different regions, rather than undertaking a quantitative assessment of these parameters." However, in the manuscript the authors stated (lines 286-287): "(dominated by vortices and boils, quantified by their vorticity and divergence, respectively, in Figs 5G & 5H)." By using "quantified" the authors contradict the idea stated in their rebuttal. I suggest the authors to clarify that the divergence and vorticity results are more qualitative than quantitative.

Minor Revisions

Line 131: "replace 20 m rotor diameter, D" by "D=20 m rotor diameter" to be uniform with line 188.

Figure 3B: please clarify in the caption which velocity was used to determine the streamlines, because it gets confusing when observing the vectors corresponding to the difference between local and reference velocities.

References 18, 20, 26, 42, 43, 45, 50, 55, 68, 77, 78 and 87 should list all the authors and not "et al."

Supplementary material review

I was probably not clear enough when I wrote: "Another question I have concerning the data processing is the fact that images were acquired at 30Hz (that is $dt = 0.033$, line 690), but in line 703 it is stated that: "instantaneous velocity fields are acquired at 0.25 s interval" (that is $f = 4$ Hz). This should be clarified, if the authors used a time averaged of about 8 images to have a so-called instantaneous value every 0.25 s, or resampled the data set."

My question was related to the acquisition frequency, and to know how the authors obtained a 0.25 s interval when the acquisition frequency is 0.033 s, which would result in 7.5 images. Is my interpretation that at each 8 frames (0.233 s), the authors choose 4 frames to perform their analysis? Perhaps it would be better to define the resample interval in terms of the main acquisition period ($t = 0.033$ s) Please check if any of the hypotheses listed in the figure I've attached corresponds to your procedure. If not, please clarify.

In the caption of Figures S3, S4, S7 and S8 indicate which method was used to determine the turbulence length scales (Taylor's frozen turbulence hypothesis or definition).

In Methods section S2 (page 27) instead of quoting equation 8 of ref 11, the authors should write down the equation they used.

Figure S12:

For better comparison, horizontal scales should be the same.

In the caption (Figure S12) add "respectively" after "S, O2 and FS indicate shear-line, O2 and free stream cross-stream location".

Figure S13:

Caption: Uniformize notation $L_u(U \Delta t)$ similar to the used in $R_{\{u,x\}}(\overline{U} \Delta t)$

I thank the authors the effort to assess the uncertainty associated with the drone stability. The authors presented a value for the central section of the field of view, they should add if this is the maximum value expected, or if in other parts of the field-of-view the uncertainty is higher. To better put things into perspective the uncertainty associated with the drone stability should be compared with the other sources of uncertainty, such as the uncertainty associated with the cross-correlation peak location among others.

Responses to Reviewers

Below is a point-by-point response to all four reviewers. Reviewer comments are highlighted in bold using indentation with responses provided below. We refer to Line numbers in the new ('clean') manuscript version. We also show all changes in an additional manuscript text file with track changes.

Reviewer #2 (Remarks to the Author):

In this round, I observed minimal revision addressing the comments raised in the previous round, particularly concerning presentation and multiphysics, which are directly relevant to the main concerns highlighted in the initial review. Below are some suggestions for improvement:

We would like to express our gratitude to the reviewer for their valuable comments. We have taken into account the reviewer's previous *observations* and have improved most figures in the manuscript as well as in the Supplementary Information for clarity. We also appreciated the earlier comment regarding multiphysics. While our measurements did not permit modelling of near-field (millimetre scales vortices) or larger-scale flow fields (kilometre-scales) beyond the study site, we have incorporated references addressing both aspects. Additionally, we will make our data available for future studies on multi-scale/multi-physics as we do support the efforts towards multiscale, multiphysics *modelling* of coastal ocean processes. However, these modelling aspects are beyond the scope of our study which was set out to capture *real-world measurements and complex flow behaviours* of regional dynamics across the site, the immediate inflow to the rotors as well as the far-wake propagation.

1. Regarding the discussion of innovation in Line 126-127, it would be beneficial to incorporate supporting evidence or references and provide further elaboration. Notably, there was a measurement conducted for the flow past tidal turbines (fix-bottom) in New York's East River.

We have now revised the section on innovation to reflect that the novelty of our measurements comes from the combination of low-cost survey methodologies and analytical approaches:

L125: "Here we set out to develop new combined survey methodologies and analytical approaches for floating platforms that can capture all three points raised above..."

2. While most results are presented via figures showing contours, these visuals, albeit varied, may not effectively convey the associated values. One approach to enhance clarity is to supplement the contours with contour lines, labeling numerical values on these lines. Alternatively, introducing tables or x-y plots could better illustrate the data.

We have provided numerical values where appropriate, such as in Table format in the Supplementary Information. Other figures contain colour scales. Rather than presenting absolute values, the intention of these figures is to provide flow visualisations to reflect flow field patterns resulting from measurements in time and space.

3. Since this study pertains to a specific site, the findings may have broader applicability if underlying patterns or rules of the data can be extracted. One method to achieve this is through dimensional analysis, presenting the data in the form of dimensionless numbers. I recommend that the authors explore this avenue and revise their work accordingly.

We have already adopted this approach where most relevant, specifically in the presentation of the turbulence intensities, velocity deficits and turbulence length scales (e.g. Figures 8 & 9).

4. In the "Discussion" section, it would be valuable to include a discussion on the multiscale (multiphysics?) aspects of the flows. For instance, exploring how the wake changes in size, direction, etc., in response to ocean currents upstream with figures/tables

We would like to thank the reviewer for their insightful comment. We do provide upstream/inflow data and the resulting far-wake dynamics/propagation in several figures (e.g. Figures 2,3 6). Additionally, we have noted the difference in wake propagation between the two ebb flow transects, suggesting that it may be associated with flow variability at larger scales, given the absence of a clear difference in wind speed, direction, or wave height between the two transects (L335-337). While a full analysis of this phenomenon is beyond the scope of our current paper, we agree that it would make an interesting study for future research. We have also now included this aspect as part of the discussion (L501-502).

Reviewer #3 (Remarks to the Author):

I appreciate the work done by the Authors in clarifying several methodological aspects in the new version of the paper.

We thank the reviewer for their comments and helpful suggestions throughout the review process. We have clarified their remaining minor comments below.

I have the following minor comments:

- Methods Section S1:

1. How many clean velocity fields were computed and used in the paper?

Each 2-min hover results in about 500 clean velocity fields and this has now been added to the Methods Section S1 (page 26).

2. What does median 3x3x3 median filter mean?

We have clarified this by the following in the Methods Section S1 (page 26):

“... , a $3 \times 3 \times 3$ median filter (filtering in both spatial dimensions and time) was applied...”

- Methods Section S3:

1. Please quantify weak vs strong wind gusts

We have added more detail for clarity in the Methods Section S3 (page 30):

“The horizontal and vertical stability of the drone is indicated by the standard deviation of the position and altitude as well as their maximum horizontal and vertical displacements during the hover (see Table S2). The yaw stability of the drone is indicated by the standard deviation in heading. Throughout the surveys when the gusts were weak (Std of wind < 0.4 m/s), the variations in drone position, altitude and yaw are much smaller than under the strong gusts (Std of wind > 2.5 m/s) of the subsequent test flight, highlighting the importance of including gusts in these drone stability assessments.”

2. Please include maximum and minimum variations in drone positions and altitude during wind gusts.

We thank the reviewer for the suggestion and have added additional values as new columns to Table S2. Drone altitude now includes the minimum, mean and maximum and we also included more detail on heading, and position data, the latter including the standard deviation in Horizontal Displacement and the Maximum Horizontal Displacement (m).

Reviewer #6 (Remarks to the Author):

The authors did a very good work in responding to the different queries and they provided additional material and clarifications that improved the clarity and quality of the paper. However, I still have some minor remarks that were either not clarified or still inconsistent, namely:

i) Regarding LSPIV/PIV

In lab flows, particles are usually seeded in the flows in order to create patterns. Particles must have certain properties, namely that their velocity is a good approximation of the flow velocity. Either patterns created by particles or by any other features such as surface waves, boils, debris, foam lines must be good proxies for the flow velocity.

Both Particle Image Velocimetry or Large Scale Particle Image Velocimetry both rely on cross-correlation, and cross-correlation can be referred as a pattern tracking mathematical operation.

Nevertheless, either in PIV or LSPIV the determination of the cross-correlation peak location is crucial for velocity determination. Except in very ideal cases, applying cross-correlation function to two time-consecutive flow images divided into interrogation windows, will result in a spatial (pixel space) distribution of peaks and valleys. Signal to noise ratio is usually defined as the ratio between the highest and second highest peak so it can also be calculated for the authors cross-correlation method. This information complements the threshold criterion used by the authors therefore I disagree with the authors' response: "The Signal to Noise ratio metric that the reviewer addresses below is used in lab-based PIV and presents a measure of the relative size of multiple correlation peaks where distinct tracer particles are used – this is a less useful metric for where features have a more continuous variation across the window, such as when using PIV in natural environments."

I also disagree with the authors regarding their answer: "including detailing the choices made to ensure robust outputs over applying more complex algorithms better suited to controlled laboratory data." those complex algorithms were developed to improve the cross-correlation results independently of controlled lab images or not.

These disagreements don't invalidate the authors' work and are here stated just for the purpose of scientific discussion.

We thank the reviewer for their insightful and really helpful suggestions. We have found this exchange to be very valuable and enjoyed our scientific discussions and the outcome. Below, we outline how we have addressed all remaining comments.

Regarding Signal to Noise ratio, we have now included a plot to show randomly selected spatial (pixel space) distributions of correlation peaks and valleys as suggested. Please refer to our new Figure S12 and associated text in the Supplementary Information (page 27). As can be seen, the distributions are predominantly unimodal, making a simple correlation threshold quality metric more suitable than metrics derived from multi-modal distributions (i.e. most distributions contain only a single peak).

In my previous review I wrote: "No reference is made to peak-locking and the authors should clarify if the PIV data had any significant peak-locking." The authors replied:

"This is when you use low seeding density of under-resolved small particles such that there is ambiguity in their actual sub-pixel location – as above, we are not using seeding particles and our method does resolve sub-pixel locations in correlation peaks as described in the new in section in the Supplementary Information (Methods Section S1, page 26)."

Subpixel resolution is applied to determine the location of the selected peak of the cross-correlation function. When particles/feature are under-resolved ($d \leq 2$ pix), the correlation function peak location tends to be in integer numbers of pixels.

Given the novelty of applying LSPIV to coastal areas and to strength the authors' results I would suggest to the authors to add in the supplementary material document a figure with the histogram of the measured cross-correlation displacements for a given time instant to assess if the natural features used to determine the correlation don't lead to peak-lock.

We thank the reviewer for this helpful suggestion. We have now included histograms of LSPIV-derived displacement data (pixels) for an undisturbed flow field upstream of the O2 for all vectors throughout a 2-min hover and for 10 random time instants within the same hover demonstrating that the combination of window size, pixel resolution and scales of natural features used to determine the correlation do not lead to peak-locking. The distributions are continuous with no strong bias to integer pixel displacement values. Please refer to our new Figure S13 in the Supplementary Information and associated commentary (page 28).

ii) Shear line

In the rebuttal the authors wrote: "‘shear line’, the region at the boundary or interface (...)"; if, by definition, a region has an area, then a line can't be a region, because the area of a line is zero. An example of this incoherence is stated on line 414: **"In the shear line area"**.

In Figure 4B the authors drew a diagonal shear line, and probably had they measured other sections parallel to that one, they would obtain a family of shear lines that would form a surface. In that sense it would result that a shear line is just the intersection of the shear surface with the velocity field measuring plane provided that the velocity gradient obeys to a given condition. What is the velocity gradient needed to define a shear line is perhaps the question that needs to be answered, as otherwise remains a concept more qualitative than quantitative.

As the authors stated, and provided references included in the text, the term is often used in oceanography, in a manner not much different from the authors' own usage, therefore, although disagreeing, I accept the authors' expression provided that:

a) the incoherences such as line 414, 419-420 where the authors write: "In the shear line area" are corrected.

We appreciate the reviewer's views on this and agree with the suggestions. We have now corrected these sections, stating 'near the shear line' throughout the manuscript, rather than refereeing to it as an 'area'.

b) in order to make a better liaison between the text (e.g. lines 300 and 303) and figures (e.g. Figure 5 E F G H) please indicate the shear lines on the figures. Same for Figures 8 A and B

c) in the supplementary material add the shear lines in Figures S4, S7 and S8

As per our definition in the introduction (L55-56), "*strong horizontal shear (cross-stream gradient in streamwise velocity, or 'shear layer')*" hereafter referred to as "*shear lines*", we believe that these figures do not require indications/lines to delineate the 'shear line' as they effectively demonstrate the cross-stream gradient in streamwise velocity. Specifically, in Figure 5C, the cross-stream gradient is clearly visualised. Subsequent Figures 5E-H further illustrate this with the spatial and temporal mean horizontal velocity magnitude, turbulence intensity (TI), and regions of vorticity and divergence depicting the same frame and field of view.

In the rebuttal the authors wrote about the use of divergence and vorticity: "We are using the spatial derivatives only to qualitatively show the variability in the flow in different regions, rather than undertaking a quantitative assessment of these parameters." However, in the manuscript the authors stated (lines 286-287): "(dominated by vortices and boils, quantified by their vorticity and divergence, respectively, in Figs 5G & 5H)." By using "quantified" the authors contradict the idea stated in their rebuttal. I suggest the authors to clarify that the divergence and vorticity results are more qualitative than quantitative.

We agree with this comment and have now edited this to the following:

L287: "... (dominated by vortices and boils, defined and visualised by their vorticity and divergence, respectively, in Figs 5G & 5H)."

Minor Revisions

Line 131: "replace 20 m rotor diameter, D" by "D=20 m rotor diameter" to be uniform with line 188.

Done.

Figure 3B: please clarify in the caption which velocity was used to determine the streamlines, because it gets confusing when observing the vectors corresponding to the difference between local and reference velocities.

Done.

References 18, 20, 26, 42, 43, 45, 50, 55, 68, 77, 78 and 87 should list all the authors and not "et al."

We have used the journal's citation style which abbreviates long author lists as such, but we are happy to change this shall it be requested from the editorial team.

Supplementary material review

I was probably not clear enough when I wrote: "Another question I have concerning the data processing is the fact that images were acquired at 30Hz (that is $dt = 0.033$, line 690), but in line 703 it is stated that: "instantaneous velocity fields are acquired at 0.25 s interval" (that is $f = 4$ Hz). This should be clarified, if the authors used a time averaged of about 8 images to have a so-called instantaneous value every 0.25 s, or resampled the data set."

My question was related to the acquisition frequency, and to know how the authors obtained a 0.25 s interval when the acquisition frequency is 0.033 s, which would result in 7.5 images. Is my interpretation that at each 8 frames (0.233 s), the authors choose 4 frames to perform their analysis? Perhaps it would be better to define the resample interval in terms of the main acquisition period ($t = 0.033$ s) Please check if any of the hypotheses listed in the figure I've attached corresponds to your procedure. If not, please clarify.

Thank you for your careful consideration of this point. You are correct that the interval should be precisely defined based on an integer number of frames (consistent with your "Hypothesis 2"). We have therefore amended our Methods in the main manuscript (L33) as "extracted at 0.266 s (8 frames) intervals" and also in the more detailed Methods Section S1 in the Supplementary Information (page 26 onwards).

In the caption of Figures S3, S4, S7 and S8 indicate which method was used to determine the turbulence length scales (Taylor's frozen turbulence hypothesis or definition).

Thank you – as set out in the main manuscript Methods, we have use the spatial method in presenting turbulence length scales. We have clarified this in the figure captions.

In Methods section S2 (page 27) instead of quoting equation 8 of ref 11, the authors should write down the equation they used.

Done.

Figure S12:

For better comparison, horizontal scales should be the same.

Done.

In the caption (Figure S12) add "respectively" after "S, O2 and FS indicate shear-line, O2 and free stream cross-stream location".

Done.

Figure S13:

Caption: Uniformize notation $L_u(U \Delta t)$ similar to the used in

$R_{\{u,x\}}(\overline{U} \Delta t)$

Done.

I thank the authors the effort to assess the uncertainty associated with the drone stability. The authors presented a value for the central section of the field of view, they should add if this is the maximum value expected, or if in other parts of the field-of-view the uncertainty is higher. To better put things into perspective the uncertainty associated with the drone stability should be compared with the other sources of uncertainty, such as the uncertainty associated with the cross-correlation peak location among others.

Thank you for this helpful suggestion. We have now included additional information and commentary in Methods Section S3 in the Supplementary Information (page 30), specifically the standard deviation in the heading that when combined with the vertical movements of the drone slightly increases (from 0.149 to 0.195 ms⁻¹) the RMS velocity fluctuation at the peripheries of the sample area. Given the substantially larger movements of the drone under the very gusty test conditions in comparison to the smaller movements of the drone during the main data collection period, we feel it reasonable to still provide a “conservative sensitivity threshold” for TI of 0.075. We have demonstrated (see above) that our LSPIV data are not impacted by cross-correlation peak locking.

REVIEWERS' COMMENTS

Reviewer #6 (Remarks to the Author):

I thank the authors for their answers to my queries. It was a good discussion and from it I believe the paper was improved in clarity and quality.

I have no further questions.